# Heterogeneous Matrix Factorization: When Features Differ by Datasets

**Naichen Shi**                                                        *naichen.shi@northwestern.edu*
*Department of Industrial Engineering and Management Sciences*
*Department of Mechanical Engineering*
*Northwestern University*

**Salar Fattahi**                                                      *fattahi@umich.edu*
*Industrial and Operations Engineering Department*
*University of Michigan*

**Raed Al Kontar**                                                     *alkontar@umich.edu*
*Industrial and Operations Engineering Department*
*University of Michigan*

**Reviewed on OpenReview:** *https: // openreview. net/ forum? id= 1BUBOI3Obx*

## Abstract

In many statistical applications, data are collected from related but heterogeneous sources. These sources share some commonalities while containing idiosyncratic characteristics. One of the most fundamental challenges in such scenarios is to recover the shared and source-specific factors at scale. Despite the existence of a few heuristic approaches, a scalable algorithm with theoretical guarantees has yet to be established.

In this paper, we tackle the problem by proposing a method called **H**eterogeneous **M**atrix **F**actorization to separate the shared and unique factors for a class of problems. HMF maintains the orthogonality between the shared and unique factors by leveraging an invariance property in the objective. The algorithm is easy to implement and intrinsically distributed. On the theoretical side, we show that for the square error loss, HMF will converge into the optimal solutions, which are close to the ground truth.

HMF can be integrated into auto-encoders to learn nonlinear feature mappings. Through a variety of case studies, we show how HMF can be applied to video segmentation, time-series feature extraction, and recommender systems.

## 1 Introduction

In the Internet of Things (IoT), data are frequently gathered at edge devices such as mobile phones or sensors, which often operate under different conditions such as temperature, pressure, or vibration (Kontar et al., 2021). This is an example of data collection from a diverse range of related sources. As a result, data exhibit both shared features, which represent common knowledge, and unique features, associated with source-specific individual characteristics.

A central challenge in analyzing the data patterns among heterogeneous sources is to decouple the shared and unique features from the observations. A natural approach to handle it is to use mutually orthogonal vectors to model the shared and unique features (Lock et al., 2013; Yang & Michailidis, 2016; Zhou et al., 2015; Sagonas et al., 2017; Gaynanova & Li, 2019; Park & Lock, 2020; Liang et al., 2023). Along this line, JIVE (Lock et al., 2013) uses an alternating minimization algorithm to optimize the joint and individual factors. While JIVE can find meaningful patterns in some genetic applications, the orthogonality is not fully respected in the algorithm. Resultingly, the separation is not ideal, and the estimates of the shared

components have limited fidelity (Zhou et al., 2015). Several works attempt to improve JIVE by analyzing the singular vectors of observation matrices (Zhou et al., 2015), considering more complicated structures (Park & Lock, 2020), and many more. However, many of these algorithms focus on specific statistical problems and employ heuristic algorithms that do not have a strong convergence guarantee. Distributing these algorithms is also an arduous task.

Very recently, personalized PCA (perPCA) (Shi & Kontar, 2024) has emerged as a promising method for distributedly recovering shared and unique features with strong convergence guarantees. Despite its strong performance on an array of applications, perPCA is limited to handling symmetric covariance matrices and cannot be directly extended to asymmetric and incomplete observation settings, where only a small subset of data is available.

Hence, there is a need for an extensible algorithm to separate shared and unique factors with a provable convergence guarantee. In this paper, we propose a framework called **H**eterogeneous **M**atrix **F**actorization (`HMF`) to capture the shared and unique factors under the orthogonality constraint. Our formulation is based on nonconvex Matrix Factorization (MF), a widely used technique in data analytics for identifying low-rank structures within high-dimensional matrices. Such low-rank structures can capture underlying physical processes or latent features that are highly informative for understanding patterns in high-dimensional observations (Wright & Ma, 2022), thus are extremely suitable to represent the shared and unique features among data. Also, the matrix factorization formulation can be extended to an auto-encoder model, which employs the representation power of neural networks.

More specifically, we consider a group of $N$ observation matrices $\{\mathbf{M}_{(i)}\}_{i=1}^N$ from $N$ related but heterogeneous sources. We model the common information in these matrices as low-rank components whose columns span the same subspace. Accordingly, the unique signals are modeled by low-rank components with source-specific column subspaces. The common and unique subspaces are orthogonal to represent the inductive bias that the shared and unique features are independent. In nonlinear models, we similarly use shared and source-specific nonlinear embeddings to model common and unique data patterns.

To learn the shared and unique features, `HMF` exploits an invariance property of the objective to handle the constraints by introducing two correction steps. The application of the invariance property is one of the key distinctions between `HMF` and previous works, as it allows `HMF` to maintain orthogonality between shared and unique factors without changing the objective. With the correction step, `HMF` is proved to converge linearly to an optimal solution under the squared error (SE) loss with suitable stepsize and initial optimality gap. We also provide an upper bound on the statistical error between the updates and the ground truth.

We use multiple numerical experiments to demonstrate the applicability of `HMF`. The case studies on video segmentation, temporal signal analysis, and movie recommendation illustrate how the separation of shared and unique factors arises naturally across diverse applications.

## 2 Related Work

**Matrix Factorization** MF has been applied to a diverse range of fields, including image processing (Lee & Seung, 1999), time series analysis (Yu et al., 2016), and many others, making it one of the most popular methods in data analytics. Numerous works analyze the theoretical properties of first-order algorithms that solve the (asymmetric) matrix factorization problem $\min_{\mathbf{U},\mathbf{V}} \left\| \mathbf{M} - \mathbf{U}\mathbf{V}^T \right\|_F^2$ or its variants (Li et al., 2018a; Ye & Du, 2021; Sun & Luo, 2016; Park et al., 2017; Tu et al., 2016). Among them, Sun & Luo (2016) analyzes the local landscape of the optimization problem and establishes the local linear convergence of a series of first-order algorithms. Park et al. (2017); Ge et al. (2017) study the global geometry of the optimization problem. Tu et al. (2016) propose the Rectangular Procrustes Flow algorithm that is proved to converge linearly into the ground truth under proper initialization and a balancing regularization of the form $\left\| \mathbf{U}^T\mathbf{U} - \mathbf{V}^T\mathbf{V} \right\|_F^2$. Despite the abundance of literature on standard matrix factorization and matrix completion, these works do not consider the case where data contain heterogeneous trends.

**Distributed matrix factorization** Recent development of edge computation has fueled a trend to move matrix factorization to the edge. Distributed matrix factorization (DMF) (Gemulla et al., 2011) exploits

the distributed gradient descent to factorize large matrices. Chai et al. (2021) proposes a cryptographic framework where multiple clients use their local data to collaboratively factorize a matrix without leaking private information to the server. These works use one set of feature matrices $\mathbf{U}$ and $\mathbf{V}$ to fit data from all clients, and hence, they do not account for source-by-source feature differences either.

**Personalized modeling** As discussed, there are a few methods attempting to find the joint and individual components by heuristic algorithms. Besides the works discussed in the introduction section, some works leverage additional structural assumptions in data. Among them, SLIDE (Gaynanova & Li, 2019) analyzes the group sparsity structure. iNMF (Yang & Michailidis, 2016) adds nonnegativity constraints to the recovered matrices. RJIVE (Sagonas et al., 2017) aims to cope with the gross noise in the observations. Recent work in multi-omics data integration has produced methods that share the global-plus-local decomposition goal. MOFA+ (Argelaguet et al., 2020) extends multi-omics factor analysis to multi-group settings using a hierarchical Bayesian model with variational EM inference. BIDIFAC (Park & Lock, 2020) and its extension BIDIFAC+ (Lock et al., 2022) decompose multi-source data into a bi-dimensional joint component and source-specific components via nuclear-norm penalization and alternating SVD updates. AJIVE (Feng et al., 2018) determines the joint subspace through the principal angles between block-specific signal spaces. ProJIVE (Murden et al., 2026) casts the same decomposition as a probabilistic model with EM-based inference. None of these methods provides convergence guarantees for a first-order algorithm under the masked observation setting. `HMF` addresses both limitations simultaneously. It improves upon the previous work by providing a flexible formulation for matrix factorization and proposing a first-order algorithm equipped with a convergence guarantee. A comprehensive comparison between `HMF` and exemplary previous works is outlined in Table 1.

Table 1: Comparison of different methods. $d$ stands for whether a method can be *d*istributed, $o$ stands for whether the method ensures the *o*rthogonality between the shared and unique components, $c$ stands for whether it has a *c*onvergence guarantee, $i$ stands for whether it can natively handle *i*ncomplete observations without explicit imputations.

| Algorithm | d | o | c | i |
|---|---|---|---|---|
| DMF (Gemulla et al., 2011) | ✓ | ✗ | ✓ | ✗ |
| JIVE (Lock et al., 2013) | ✗ | ✗ | ✗ | ✗ |
| COBE (Zhou et al., 2015) | ✗ | ✗ | ✗ | ✗ |
| RJIVE (Sagonas et al., 2017) | ✗ | ✓ | ✗ | ✗ |
| BIDIFAC (Park & Lock, 2020) | ✗ | ✗ | ✗ | ✗ |
| perPCA (Shi & Kontar, 2024) | ✓ | ✓ | ✓ | ✗ |
| perDL (Liang et al., 2023) | ✓ | ✗ | ✓ | ✗ |
| `HMF` (ours) | ✓ | ✓ | ✓ | ✓ |

## 3 Model

We consider the setting where $N \in \mathbb{N}^+$ noisy observation matrices $\mathbf{M}_{(1)}, \mathbf{M}_{(2)}, \cdots, \mathbf{M}_{(N)}$ are collected from $N$ different but related sources. These matrices $\mathbf{M}_{(i)} \in \mathbb{R}^{n_1 \times n_{2,(i)}}$ are assumed to have the same number of rows $n_1$.

### 3.1 Linear model

The natural way to model the commonality and uniqueness among the matrices is to assume that each matrix is driven by $r_1$ shared factors and $r_{2,(i)}$ unique factors and also contaminated by noise. More specifically, we consider the generation model for matrix $\mathbf{M}_{(i)}$ is defined as

$$\mathbf{M}_{(i)} = \mathbf{U}^{\star}_g \mathbf{V}^{\star T}_{(i),g} + \mathbf{U}^{\star}_{(i),l} \mathbf{V}^{\star T}_{(i),l} + \mathbf{E}^{\star}_{(i)} \tag{1}$$

where $\mathbf{U}^{\star}_g \in \mathbb{R}^{n_1 \times r_1}$, $\mathbf{V}^{\star}_{(i),g} \in \mathbb{R}^{n_{2,(i)} \times r_1}$, $\mathbf{U}^{\star}_{(i),l} \in \mathbb{R}^{n_1 \times r_{2,(i)}}$, $\mathbf{V}^{\star}_{(i),l} \in \mathbb{R}^{n_{2,(i)} \times r_{2,(i)}}$, $\mathbf{E}^{\star}_{(i)} \in \mathbb{R}^{n_1 \times n_{2,(i)}}$. We use $\star$ to denote the ground truth. In the above model, $r_1$ is the rank of the global (shared) feature matrices,

while $r_{2,(i)}$ is the rank of local (unique) feature matrices from source $i$. The matrix $\mathbf{U}^\star{}_g\mathbf{V}^\star{}_{(i),g}^T$ models the shared low-rank components of the observation matrix, as the column space is the same across different sources. The matrix $\mathbf{U}^\star{}_{(i),l}\mathbf{V}^\star{}_{(i),l}^T$ models the unique low-rank part. The rationale behind the low-rank matrix factorization is to use a small number of latent vectors to explain the observations. Thus the ranks $r_1$ and $r_{2,(i)}$ are often much smaller than matrix dimensions. $\mathbf{E}^\star{}_{(i)}$ models the noise from source $i$.

In matrix factorization problems, the representations $\mathbf{U}^\star{}_g$ and $\mathbf{U}^\star{}_{(i),l}$ often correspond to latent data features. For better interpretability, it is often desirable to have the underlying features disentangled so that each feature can vary independently of others (Higgins et al., 2017). Under this rationale, we consider the model where shared and unique factors are orthogonal,

$$\mathbf{U}^\star{}_g^T\mathbf{U}^\star{}_{(i),l} = 0, \ \forall i \in [N] \tag{2}$$

We use $[N]$ to denote the set $\{1, 2, \cdots, N\}$. The orthogonality of features implies that the shared and unique features span different subspaces, thus describing different patterns in the observation. The orthogonality condition is consistent with literature (Lock et al., 2013; Zhou et al., 2015). Beyond interpretability, orthogonality is essential for identifiability: without a constraint separating the shared subspace from each source-specific subspace, the same reconstruction can be represented by shifting components between the global and local terms. Explicitly enforcing orthogonality mitigates the ambiguity. Ablation studies in Section 6 verify the effectiveness of the orthogonality modeling numerically.

Under the data generation model equation 1 and equation 2, our goal is to find $\mathbf{U}_g, \{\mathbf{V}_{(i),g}, \mathbf{U}_{(i),l}, \mathbf{V}_{(i),l}\}$ from observations $\{\mathbf{M}_{(i)}\}$. To achieve this goal, we propose a generic constrained optimization problem.

$$\min_{\mathbf{x}} \sum_{i=1}^N \tilde{f}_i(\mathbf{U}_g, \mathbf{V}_{(i),g}, \mathbf{U}_{(i),l}, \mathbf{V}_{(i),l}) \quad \text{such that } \mathbf{U}_g^T\mathbf{U}_{(i),l} = 0, \ \forall i \in [N] \tag{3}$$

where $\mathbf{x} = \left(\mathbf{U}_g, \{\mathbf{U}_{(i),l}, \mathbf{V}_{(i),g}, \mathbf{V}_{(i),l}\}_{i=1}^N\right)$ collects the decision variables and $\tilde{f}_i$ is a regularized empirical risk consisting of two parts.

$$
\begin{aligned}
&\tilde{f}_i(\mathbf{U}_g, \mathbf{V}_{(i),g}, \mathbf{U}_{(i),l}, \mathbf{V}_{(i),l}) \\
&= \underbrace{\ell\left(\mathbf{M}_{(i)}, \mathbf{U}_g\mathbf{V}_{(i),g}^T + \mathbf{U}_{(i),l}\mathbf{V}_{(i),l}^T\right)}_{f_i} + \underbrace{\frac{\beta}{2}\left\|\mathbf{U}_g^T\mathbf{U}_g - \mathbf{I}\right\|_F^2 + \frac{\beta}{2}\left\|\mathbf{U}_{(i),l}^T\mathbf{U}_{(i),l} - \mathbf{I}\right\|_F^2}_{g_i}
\end{aligned} \tag{4}
$$

We will explain them in tern. Term $f_i$ is the empirical risk between the sum of shared and unique signals and the observation matrix $\mathbf{M}_{(i)}$. $\ell$ is a loss metric that can incorporate a wide range of applications. For example, in matrix factorization, the squared error loss is $\ell^{se}\left(\mathbf{M}_{(i)}, \hat{\mathbf{M}}_{(i)}\right) = \frac{1}{2}\left\|\mathbf{M}_{(i)} - \hat{\mathbf{M}}_{(i)}\right\|_F^2$, i.e., the square error between the observed and reconstructed matrices. In matrix completion, only a limited number of entries are observed. We use $\Omega_{(i)}$ to denote the set of indices of the observed entries, $\mathcal{P}_{\Omega_{(i)}}(\cdot)$ to denote the projection

$$\left[\mathcal{P}_{\Omega_{(i)}}(\mathbf{M})\right]_{j,k} = \begin{cases} [\mathbf{M}]_{j,k} & \text{if entry } (j,k) \in \Omega_{(i)} \\ 0 & \text{otherwise.} \end{cases}$$

for a general matrix $\mathbf{M}$. Then the projected square error loss $\ell^{pse}$ is defined as $\ell^{pse}\left(\mathbf{M}_{(i)}, \hat{\mathbf{M}}_{(i)}\right) = \frac{1}{2}\left\|\mathcal{P}_{\Omega_{(i)}}\left(\mathbf{M}_{(i)} - \hat{\mathbf{M}}_{(i)}\right)\right\|_F^2$. There are also numerous other loss metrics $\ell$, such as Huber loss used in robust matrix factorization (Wang & Fan, 2022) or cross-entropy loss used in community detection (Yang & Leskovec, 2013).

The constraints $\mathbf{U}_g^T\mathbf{U}_{(i),l} = 0$ reflect our prior belief about the data generation process. Since we consider the model where the true shared and unique features are orthogonal equation 2, it is natural to encode the orthogonality into constraints. Explicitly enforcing the orthogonality constraints is one of the major differences between our work and the previous works including JIVE. As we will see in numerical examples, such constraints can encourage shared and unique features to capture more distinct patterns.

Term $g_i$ is a regularization term that pushes the $\mathbf{U}$ matrices to be orthonormal. A well-known theoretical issue in matrix factorization is that objectives like the term $f_i$ are homogeneous with respect to the variables, which in turn leads to the non-smoothness of the loss function. Regularization terms are added to ensure the smoothness of the loss. Several works on matrix completion (e.g. (Park et al., 2017; Tu et al., 2016; Fattahi & Sojoudi, 2020)) consider balancing regularization terms of the form $\left\| \mathbf{U}_g^T \mathbf{U}_g - \mathbf{V}_{(i),g}^T \mathbf{V}_{(i),g} \right\|_F^2$ to encourage column factors $\mathbf{U}_g$ and row factors $\mathbf{V}_{(i),g}$ to have similar singular values. Under a similar rationale, we leverage the regularization terms in $g_i$ to prevent too radical $\mathbf{U}_g$ and $\mathbf{U}_{(i),l}$'s, as such regularizations are easier to work with in the presence of the constraints $\mathbf{U}_g^T \mathbf{U}_{(i),l} = 0$. Here, the parameter $\beta$ controls the strength of the regularization.

Since the terms $f_i$ and $g_i$ are both nonconvex, and the feasible set corresponding to the constraint $\mathbf{U}_g^T \mathbf{U}_{(i),l} = 0$ is also nonconvex, the problem in equation 3 is nonconvex. In the next section, we will propose a gradient-based and distributed algorithm to solve the nonconvex problem.

## 3.2 Extension to auto-encoders

The model equation 1 is based on linear embeddings of $\mathbf{U}$ and $\mathbf{V}$. As a natural extension, we use auto-encoders (Goodfellow et al., 2016) to find the nonlinear embeddings in data. For notational simplicity, if some entries of a matrix $\mathbf{M}$ are missing, we use $\mathbf{M}^{obs}$ to denote the padded matrix $\mathbf{M}$ whose unobserved entries are simply replaced with 0. For auto-encoders, the feature matrix $\mathbf{U}$ is replaced by an encoder network $\mathtt{Net}_{\mathbf{U}} : \mathbb{R}^d \to \mathbb{R}^r$ that maps a column of matrix $\mathbf{M}_{(i)}^{obs}$ into an embedding vector with dimension $r$, and the coefficient matrix $\mathbf{V}$ is replaced by a decoder neural network $\mathtt{Net}_{\mathbf{V}} : \mathbb{R}^r \to \mathbb{R}^d$ that maps an embedding vector with dimension $r$ to the reconstructed observation vector in $\mathbb{R}^d$. We use $\mathbf{U}$ and $\mathbf{V}$ to denote trainable weights that parametrize the neural networks. To accommodate the heterogeneity of features from $N$ different $\mathbf{M}_{(i)}^{obs}$, we also introduce a shared encoder $\mathtt{Net}_{\mathbf{U}_g}$ and $N$ unique encoders $\{\mathtt{Net}_{\mathbf{U}_{(i),l}}\}$ to generate the embeddings. All encoders have the same architecture but are parametrized by different weights. In analogy to equation 3, we thus propose the following objective for the nonlinear version of $\mathtt{HMF}$,

$$\min_{\mathbf{U}_g, \{\mathbf{U}_{(i),l}\}, \mathbf{V}} \sum_{i=1}^{N} \left[ \ell \Big( \mathbf{M}_{(i)}, \mathtt{Net}_{\mathbf{V}} \Big( \mathtt{Net}_{\mathbf{U}_g} \Big( \mathbf{M}_{(i)}^{obs} \Big) + \mathtt{Net}_{\mathbf{U}_{(i),l}} \Big( \mathbf{M}_{(i)}^{obs} \Big) \Big) \Big) + \mathcal{R} \Big( \mathbf{U}_g, \mathbf{U}_{(i),l}, \mathbf{V} \Big) \right]$$

$$\text{such that, } \mathcal{F}(\mathtt{Net}_{\mathbf{U}_g}([\mathbf{M}_{(i)}^{obs}]_{:,j}))^T \mathcal{F}(\mathtt{Net}_{\mathbf{U}_{(i),l}}([\mathbf{M}_{(i)}^{obs}]_{:,j})) = 0, \quad \forall j \in [n_{(i)}], \quad i \in [N] \tag{5}$$

In equation 5, we use a compact notation $\mathtt{Net}_{\mathbf{U}_g}(\mathbf{M}_{(i)}^{obs})$ to denote the application of $\mathtt{Net}_{\mathbf{U}_g}$ on each column of $\mathbf{M}_{(i)}^{obs}$. $\ell$ is still a loss metric. $\mathcal{R}\left(\mathbf{U}_g, \mathbf{U}_{(i),l}, \mathbf{V}\right)$ is a regularization term that prevents the over-fitting of the neural network. In the constraint, $\mathcal{F}(\cdot)$ is the folding operator that folds an $r$-dimensional vector into $k$ vectors, each with dimension $r/k$. The constraint $\mathcal{F}(\mathtt{Net}_{\mathbf{U}_g}([\mathbf{M}_{(i)}^{obs}]_{:,j}))^T \mathcal{F}(\mathtt{Net}_{\mathbf{U}_{(i),l}}([\mathbf{M}_{(i)}^{obs}]_{:,j})) = 0$ is a "group orthogonalization condition" that requires the $k$ folded vectors generated by the shared encoders to be orthogonal to $k$ folded vectors generated by the unique encoders for each column of $\mathbf{M}_{(i)}^{obs}$. Such group orthogonalization constraint is a natural extension of the linear constraint in equation 3, and encourages $\mathtt{Net}_{\mathbf{U}_g}$ and $\mathtt{Net}_{\mathbf{U}_{(i),l}}$ to encode different embeddings.

We call this formulation heterogeneous auto-encoders ($\mathtt{HAE}$).

## 4 Algorithm

This section will develop the heterogeneous matrix factorization ($\mathtt{HMF}$) algorithm and its extension to optimize the heterogeneous auto-encoder equation 5.

### 4.1 Heterogeneous Matrix Factorization

We use $\tilde{f}$ to denote the summation of $\tilde{f}_i$'s over different sources $i$,

$$\tilde{f}(\mathbf{U}_g, \{\mathbf{V}_{(i),g}, \mathbf{U}_{(i),l}, \mathbf{V}_{(i),l}\}) = \sum_{i=1}^{N} \tilde{f}_i \tag{6}$$

The objective $\tilde{f}$ is differentiable. Thus to optimize equation 3, one can calculate the gradient of $\tilde{f}_i$ with respect to its variables as,

$$\begin{cases} \nabla_{\mathbf{U}_g} \tilde{f}_i = \ell' \mathbf{V}_{(i),g} + 2\beta \mathbf{U}_g \left(\mathbf{U}_g^T \mathbf{U}_g - \mathbf{I}\right) \\ \nabla_{\mathbf{V}_{(i),g}} \tilde{f}_i = \ell'^T \mathbf{U}_g \\ \nabla_{\mathbf{U}_{(i),l}} \tilde{f}_i = \ell' \mathbf{V}_{(i),l} + 2\beta \mathbf{U}_{(i),l} \left(\mathbf{U}_{(i),l}^T \mathbf{U}_{(i),l} - \mathbf{I}\right) \\ \nabla_{\mathbf{V}_{(i),l}} \tilde{f}_i = \ell'^T \mathbf{U}_{(i),l} \end{cases} \tag{7}$$

where $\ell'$ denotes the gradient of the empirical loss $\ell$, $\ell' = \nabla_{\hat{\mathbf{M}}_{(i)}} \ell(\mathbf{M}_{(i)}, \hat{\mathbf{M}}_{(i)})$.

The constraint $\mathbf{U}_g^T \mathbf{U}_{(i),l} = 0$ poses challenges to the optimization. One naive idea is to use projected gradient descent to handle this constraint. However, the projection cannot be easily implemented as the feasible region is nonconvex. A few works also introduce infeasible updates when the constraint is complicated, including ADMM (Hong & Luo, 2017) or SQP (Curtis & Overton, 2012). Such approaches often introduce additional tuning parameters to balance the objective and the constraint.

*We take a different route by exploiting a special invariance property* in equation 3. More specifically, for any $\mathbf{R} \in \mathbb{R}^{r_1 \times r_{2,(i)}}$, we can apply the transform $\varphi_{\mathbf{R}}$ on $\mathbf{U}_g$ and $\mathbf{U}_{(i),l}$,

$$\varphi_{\mathbf{R}} : \left(\mathbf{U}_g, \mathbf{V}_{(i),g}, \mathbf{U}_{(i),l}, \mathbf{V}_{(i),l}\right) \mapsto \left(\mathbf{U}_g, \mathbf{V}_{(i),g} + \mathbf{V}_{(i),l}\mathbf{R}^T, \mathbf{U}_{(i),l} - \mathbf{U}_g\mathbf{R}, \mathbf{V}_{(i),l}\right)$$

without changing $f_i$: $f_i(\mathbf{U}_g, \mathbf{V}_{(i),g}, \mathbf{U}_{(i),l}, \mathbf{V}_{(i),l}) = f_i\left(\varphi_{\mathbf{R}}\left(\mathbf{U}_g, \mathbf{V}_{(i),g}, \mathbf{U}_{(i),l}, \mathbf{V}_{(i),l}\right)\right)$.

The invariance property is a result of the special bi-linear structure in equation 4. One can use such invariance to ensure the feasibility of the iterations. By choosing $\mathbf{R} = \left(\mathbf{U}_g^T \mathbf{U}_g\right)^{-1} \mathbf{U}_g^T \mathbf{U}_{(i),l}$, the transform $\varphi_{\mathbf{R}}$ automatically corrects $\mathbf{V}_{(i),g}$ and $\mathbf{U}_{(i),l}$, such that $\mathbf{U}_{(i),l}$ is orthogonal to $\mathbf{U}_g$, without changing $f_i$. Based on this fact, we can propose an iterative algorithm. In each epoch, we use $\varphi_{\mathbf{R}}$ to correct the variables to ensure feasibility. Then we use gradient descent on $(\mathbf{U}_g, \mathbf{V}_{(i),g}, \mathbf{U}_{(i),l}, \mathbf{V}_{(i),l})$ to decrease the regularized objective. A pseudo-code is shown in Algorithm 1.

In Algorithm 1, we use $\tau$ to denote the iteration index, where a half-integer denotes that the update of the variable is half complete: it is updated by gradient descent but is not feasible yet.

One salient feature of algorithm 1 is that it is distributed in nature. Suppose there are $N$ computation nodes, each holding one observation matrix $\mathbf{M}_{(i)}$. Then, Algorithm 1 can be run on these computation nodes with the help of a central server. In such a scenario, node $i$ carries out all computation from line 5 to line 10 in Algorithm 1 (colored in teal) and sends the updated copies of $\mathbf{U}_{(i),g}$ to the central server. The server then takes the average in line 12 (colored in brown) and broadcasts the averaged $\mathbf{U}_g$ to all nodes. Per round, each client communicates an update of size $O(n_1 \times r_1)$ to the server, and receives an updated $\mathbf{U}_g$ of the same size back. The local data matrix $\mathbf{M}_{(i)}$ and local factors are never shared. The total communication per epoch is therefore $2N \times n_1 \times r_1$ (bidirectional), which grows only in the shared rank $r_1$. The matrix inversion in algorithm 1 takes $O(r_1^3 + r_{2,(i)}^3)$ operations, all other matrix productions take $O(n_1 n_2 (r_1 + r_{2,(i)}))$ operations. Therefore if $r_1$ and $r_{2,(i)}$ are dominated by $n_1$ and $n_2$, the computational complexity of one single iteration is $O(n_1 n_2 (r_1 + r_{2,(i)}))$.

In its distributed implementation, `HMF` avoids sending raw local data and local factors to the server, but it does not by itself provide a formal privacy guarantee such as differential privacy or cryptographic security. However, since HMF resembles standard distributed learning, popular privacy-preserving techniques, including differential privacy or secure aggregation are directly applicable.

Algorithm 1 can be readily extended to train the auto-encoder in equation 5.

---

**Algorithm 1** `HMF`

---

1: Input matrices $\{\mathbf{M}_{(i)}\}_{i=1}^N$, stepsize $\eta$
2: Initialize $\mathbf{U}_{g,1}, \mathbf{V}_{(i),g,\frac{1}{2}}, \mathbf{U}_{(i),l,\frac{1}{2}}, \mathbf{V}_{(i),l,1}$ to be small random matrices.
3: **for** Iteration $\tau = 1, ..., R$ **do**
4:     **for** $i = 1, \cdots, N$ **do**
5:         Correct $\mathbf{U}_{(i),l,\tau} = \mathbf{U}_{(i),l,\tau-\frac{1}{2}} - \mathbf{U}_{g,\tau} \left( \mathbf{U}_{g,\tau}^T \mathbf{U}_{g,\tau} \right)^{-1} \mathbf{U}_{g,\tau}^T \mathbf{U}_{(i),l,\tau-\frac{1}{2}}$
6:         Correct $\mathbf{V}_{(i),g,\tau} = \mathbf{V}_{(i),g,\tau-\frac{1}{2}} + \mathbf{V}_{(i),l,\tau} \mathbf{U}_{(i),l,\tau-\frac{1}{2}}^T \mathbf{U}_{g,\tau} \left( \mathbf{U}_{g,\tau}^T \mathbf{U}_{g,\tau} \right)^{-1}$
7:         Update $\mathbf{U}_{(i),g,\tau+1} = \mathbf{U}_{g,\tau} - \eta \nabla_{\mathbf{U}_g} \tilde{f}_i$.
8:         Update $\mathbf{V}_{(i),g,\tau+\frac{1}{2}} = \mathbf{V}_{(i),g,\tau} - \eta \nabla_{\mathbf{V}_{(i),g}} \tilde{f}_i$.
9:         Update $\mathbf{U}_{(i),l,\tau+\frac{1}{2}} = \mathbf{U}_{(i),l,\tau} - \eta \nabla_{\mathbf{U}_{(i),l}} \tilde{f}_i$.
10:        Update $\mathbf{V}_{(i),l,\tau+1} = \mathbf{V}_{(i),l,\tau} - \eta \nabla_{\mathbf{V}_{(i),l}} \tilde{f}_i$.
11:     **end for**
12:     Calculate $\mathbf{U}_{g,\tau+1} = \frac{1}{N} \sum_{i=1}^N \mathbf{U}_{(i),g,\tau+1}$
13: **end for**
14: Return $\mathbf{U}_{g,R}, \{\mathbf{V}_{(i),g,R}\}, \{\mathbf{U}_{(i),l,R}\}, \{\mathbf{V}_{(i),l,R}\}$.

---

## 5 Convergence Analysis

In spite of its simplicity, Algorithm 1 has provable convergence guarantees. In this section, we first show that if $\ell$ is the square error (SE), `HMF` can converge into the optimal solutions, which also has a statistical error upper bound. Furthermore, when $\ell$ takes generic forms, `HMF` will converge into KKT points.

In this section we use $\mathcal{B}(B_1, B_2)$ to denote the set of variables with bounded norms, $\mathcal{B}(B_1, B_2) = (\mathbf{x}; \|\mathbf{U}_g\|, \|\mathbf{U}_{(i),l}\| \leq B_1, \|\mathbf{V}_{(i),g}\|, \|\mathbf{V}_{(i),l}\| \leq B_2)$. We relegate the proof of all theorems in this section to the supplementary materials.

### 5.1 Square Error

The square error, denoted as $\ell^{se}(\mathbf{M}_{(i)}, \hat{\mathbf{M}}_{(i)}) = \left\| \mathbf{M}_{(i)} - \hat{\mathbf{M}}_{(i)} \right\|_F^2$, measures the sum of the square of the element-wise difference between $\mathbf{M}_{(i)}$ and $\hat{\mathbf{M}}_{(i)}$. Setting $\ell$ to $\ell^{se}$ naturally generates the formulation of matrix factorization. Consequently, we thoroughly investigate the convergence behavior of `HMF` with respect to the SE.

Before presenting the convergence results, we briefly review the concept of misalignment proposed in Shi & Kontar (2024). Intuitively speaking, misalignment characterizes the average "minimal difference" among the subspace spanned by a series of vectors. More formally,

**Definition 5.1** *($\theta$-misalignment) We say $\{\mathbf{U}^\star_{(i),l}\}_{i=1}^N$ are $\theta$-misaligned if there exists a positive constant $\theta \in (0,1)$ such that, $\lambda_{\max} \left( \frac{1}{N} \sum_{i=1}^N \mathbf{P}_{\mathbf{U}^\star_{(i),l}} \right) \leq 1 - \theta$.*

where we define the projection matrix $\mathbf{P}_\mathbf{U} \in \mathbb{R}^{d \times d}$ for a matrix $\mathbf{U} \in \mathbb{R}^{d \times n}$ as $\mathbf{P}_\mathbf{U} = \mathbf{U} \left( \mathbf{U}^T \mathbf{U} \right)^{-1} \mathbf{U}^T$.

With the definition of misalignment, we are able to show the following upper bound on the statistical error.

**Theorem 1 (Statistical error)** *Consider the data generation model equation 1. Suppose that (i) the signal part $\mathbf{U}^\star_g \mathbf{V}^{\star T}_{(i),g} + \mathbf{U}^\star_{(i),l} \mathbf{V}^{\star T}_{(i),l}$ has $r_1 + r_2$ nonzero singular values upper bounded by $\sigma_{\max}$ and lower bounded by $\sigma_{\min}$, (ii) unique factors $\mathbf{U}^\star_{(i),l}$ are $\theta$-misaligned, (iii) $\hat{\mathbf{U}}_g, \{\hat{\mathbf{V}}_{(i),g}, \hat{\mathbf{U}}_{(i),l}, \hat{\mathbf{V}}_{(i),l}\}_{i=1}^N$ is one set of optimal*

*solutions to equation 3 with square error loss $\ell^{se}$, then, the following holds*

$$\sum_{i=1}^{N} \left\| \hat{\mathbf{U}}_g \hat{\mathbf{V}}_{(i),g}^T - \mathbf{U}^{\star}_g \mathbf{V}^{\star T}_{(i),g} \right\|_F^2 + \left\| \hat{\mathbf{U}}_{(i),l} \hat{\mathbf{V}}_{(i),l}^T - \mathbf{U}^{\star}_{(i),l} \mathbf{V}^{\star T}_{(i),l} \right\|_F^2$$

$$= O\Big( \frac{1}{\theta} \sum_{i=1}^{N} \Big( \left\| \mathbf{E}^{\star}_{(i)} \right\|_F + \left\| \mathbf{E}^{\star}_{(i)} \right\|_F^2 \Big)^2 + \Big( \left\| \mathbf{E}^{\star}_{(i)} \right\|_F + \left\| \mathbf{E}^{\star}_{(i)} \right\|_F^2 \Big) \Big) \tag{8}$$

Theorem 1 provides an upper bound on the distance between the optimal solution to equation 3 and the ground truth. To the best of our knowledge, we are the first to prove such statistical error bound in the context of heterogeneous matrix factorization. Interestingly, when the misalignment parameter $\theta$ is larger, the statistical error upper bound in equation 8 is smaller, which means the features are easier to identify. In the following, we will continue to discuss the convergence into the optimal solutions.

We use $\phi_{(i),\tau}$ to denote the optimality gap at iteration $\tau$ of `HMF` for client $i$, $\phi_{(i),\tau} = \tilde{f}_i(\mathbf{U}_{g,\tau}, \mathbf{V}_{(i),g,\tau}, \mathbf{U}_{(i),l,\tau}, \mathbf{V}_{(i),l,\tau}) - \tilde{f}_i(\hat{\mathbf{U}}_{g,\tau}, \hat{\mathbf{V}}_{(i),g,\tau}, \hat{\mathbf{U}}_{(i),l,\tau}, \hat{\mathbf{V}}_{(i),l,\tau})$. Accordingly, the total optimality gap is defined as $\phi_\tau = \sum_{i=1}^{N} \phi_{(i),\tau}$. The following theorem establishes `HMF`'s linear convergence.

**Theorem 2** *(Convergence of `HMF`) If the assumptions in Theorem 1 are satisfied, and additionally, the following conditions hold: (i) $\mathbf{M}_{(i)} = \mathbf{U}^{\star}_g \mathbf{V}^{\star T}_{(i),g} + \mathbf{U}^{\star}_{(i),l} \mathbf{V}^{\star T}_{(i),l} + \mathbf{E}^{\star}_{(i)}$, where $\left\| \mathbf{E}^{\star}_{(i)} \right\|_F = O(\theta\sigma_{\min})$, (ii) the initial optimality gap satisfies $\phi_1 = O\left( \theta^{1.5}\sigma_{\min}^2 \right)$, (iii) the stepsize satisfies $\eta = O\left( \frac{1}{\sigma_{\max}^2} \right)$.*

*Then, there exists a constant $C > 0$ such that the iterations of `HMF` satisfy*

$$\phi_\tau \leq (1 - C\eta)^{\tau-1} \phi_1 \tag{9}$$

The first condition in Theorem 2 imposes an upper bound on the noise matrix so that the benign optimization landscape is not destroyed by the gross noise. Intuitively, if the noise $\mathbf{E}^{\star}_{(i)}$ is too large relative to the signal gap $\sigma_{\min}$, the objective landscape becomes too challenging near the optimum, and gradient steps can no longer make reliable progress. The bound $\|\mathbf{E}^{\star}_{(i)}\|_F = O(\theta\sigma_{\min})$ ensures the noise is small enough that the global and local factors of the data remain identifiable.

The second condition in Theorem 2 requires an upper bound on the initial optimality gap. Roughly speaking, this condition ensures that the iterates do not become trapped at a sub-optimal local solution and lie within a basin of attraction of the global solution. In practice, this condition is mild: our experiments show that trajectories starting from a small Gaussian random initialization or a spectral initialization often enter the basin within a short transient phase. We note that recent results have relaxed this condition for other variants of nonconvex matrix factorization by resorting to small, random, or spectral initialization techniques (Li et al., 2018b; Stöger & Soltanolkotabi, 2021; Ma & Fattahi, 2022; Ma et al., 2022; Tu et al., 2016; Ma et al., 2018). We believe such techniques can be used in `HMF` to relax the aforementioned initial condition. In fact, in our simulations, we observed that `HMF` with a small Gaussian random initial point converges to a global solution in almost all instances.

We leave the rigorous analysis of this observation as an enticing challenge for future research. Finally, the last condition in Theorem 2 imposes an upper bound on the stepsize of our algorithm to guarantee its convergence. This is a standard condition in non-convex optimization: a step that is too large can overshoot and increase the objective. The bound $\eta = O(1/\sigma_{\max}^2)$ ensures each step is within the locally linear regime of the loss.

## 5.2 General Loss

Under a generic Lipschitz continuous loss function $\ell$, less geometric information is known for the loss landscape. Still, we are able to prove that Algorithm 1 will converge into a KKT point to problem equation 3. The Karush–Kuhn-Tucker (KKT) conditions are first-order necessary optimality conditions for constrained optimization problems (Bertsekas, 1997). For our problem, they consist of stationarity of the Lagrangian

and primal feasibility of the orthogonality constraints $\mathbf{U}_g^\top \mathbf{U}_{(i),l} = 0$. Convergence to a KKT point therefore means convergence to a first-order stationary point of the constrained HMF objective. In this section, we assume $r_{2,(i)} = r_2$, without loss of generality.

**Theorem 3 (Convergence of HMF under general loss)** *Suppose that the following conditions are satisfied for HMF: (i) there exist constants $B_1, B_2 > 0$ that can upper bound the norm of all iterates $\left(\mathbf{U}_g, \{\mathbf{V}_{(i),g}, \mathbf{U}_{(i),l}, \mathbf{V}_{(i),l}\}\right) \in \mathcal{B}(B_1, B_2)$, (ii) $\ell$ is $L$-Lipschitz continuous, (iii) the stepsize satisfies $\eta = O\left(\frac{1}{L^2}\right)$. Then*

$$\min_{\tau \in \{1, \cdots, T-1\}} \left\| \nabla \tilde{f}(\mathbf{x}_\tau) \right\|_F^2 = O\left(T^{-1}\right).$$

It is worth noting that at KKT points of problem equation 3, the gradient norm is zero $\left\| \nabla \tilde{f} \right\|_F = 0$. Hence, Theorem 3 essentially characterizes the rate for Algorithm 1 to converge into KKT solutions. Such convergence guarantees are not attainable for the previous heuristic algorithms, including JIVE (Lock et al., 2013), COBE (Zhou et al., 2015), BIDIFAC (Park & Lock, 2020), and many more, while HMF is guaranteed to converge. The improvement results from the introduced correction steps in Algorithm 1.

**Proof sketch.** The proofs of Theorems 2 and 3 share a common backbone. We first prove the sufficient decrease property that shows each HMF iteration reduces the objective by at least $\eta \| \nabla \tilde{f} \|_F^2 - O(\eta^2)$. This property shows that HMF is well-designed as the iterates are guaranteed to make consistent progress. We then establish the PL inequality for the SE loss. The PL inequality certifies the geometry of the optimization problem by connecting how far the current iterates can be from the optimum with the optimality gap. Combining the analysis of the algorithm and geometry, we can establish the convergence of HMF. The full proof is relegated to the appendix.

## 6 Experimental Results

In this section, we present the results of the numerical simulations. We use a synthetic example and three real-life case studies. Code for all numerical studies is available in the following repository: `https://github.com/UMDataScienceLab/hmf`.

### 6.1 Synthetic Data

We use synthetic data to examine the numerical convergence of HMF. We generate data according to the model equation 1, where $\mathbf{U}^\star_g$, $\mathbf{U}^\star_{(i),l}$, $\mathbf{V}^\star_{(i),g}$, $\mathbf{V}^\star_{(i),l}$ are randomly sampled from Gaussian distributions. Then we deflate $\mathbf{U}^\star_{(i),l}$ to satisfy the requirement $\mathbf{U}^{\star T}_g \mathbf{U}^\star_{(i),l} = 0$. The noise $\mathbf{E}^\star_{(i)}$ is set to 0 to examine the convergence behavior of our algorithm better. We fix $n_2 = 100$ and select $n_1 \in \{30, 60, 120\}$ to see the effect of different observation matrix sizes. The ranks of the global and local components are set to $r_1 = r_2 = 3$, and the number of clients is $N = 100$. We run HMF on the generated data with the decision variables $\mathbf{U}_g$, $\mathbf{U}_{(i),l}$, $\mathbf{V}_{(i),g}$, $\mathbf{V}_{(i),l}$ initialized as small Gaussian matrices. The errors $\sum_{i=1}^N \left\| \mathbf{U}_{g,\tau} \mathbf{V}^T_{(i),g,\tau} - \mathbf{U}^\star_g \mathbf{V}^{\star T}_{(i),g} \right\|_F^2$ and $\sum_{i=1}^N \left\| \mathbf{U}_{(i),l,\tau} \mathbf{V}^T_{(i),l,\tau} - \mathbf{U}^\star_{(i),l} \mathbf{V}^{\star T}_{(i),l} \right\|_F^2$ are calculated in each iteration and plotted as shared fea-

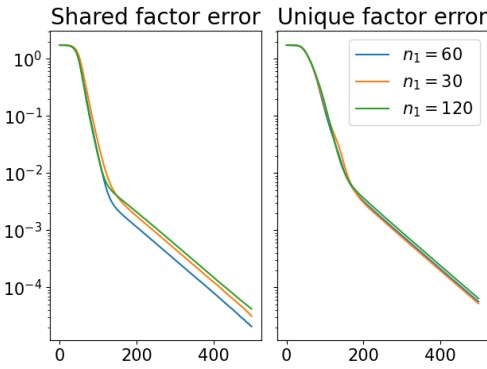

Figure 1: Shared factor error and unique factor error (log-scale) in each iteration.

Table 2: Subspace error under different ratios of missing entries.

| Algorithm | 50% | 10% | 5% | 1% |
|---|---|---|---|---|
| RJIVE | $7.9 \pm 0.1$ | $5.9 \pm 0.1$ | $5.9 \pm 0.1$ | $5.9 \pm 0.1$ |
| JIVE | $0.52 \pm 0.01$ | $(5.4 \pm 0.1) \times 10^{-2}$ | $(2.6 \pm 0.1) \times 10^{-2}$ | $(5.2 \pm 0.2) \times 10^{-3}$ |
| perPCA | $0.51 \pm 0.01$ | $(5.4 \pm 0.1) \times 10^{-2}$ | $(2.6 \pm 0.1) \times 10^{-2}$ | $(5.2 \pm 0.2) \times 10^{-3}$ |
| HMF | $\mathbf{(4.5 \pm 0.7) \times 10^{-2}}$ | $\mathbf{(2.0 \pm 0.2) \times 10^{-6}}$ | $\mathbf{(7.3 \pm 0.4) \times 10^{-7}}$ | $\mathbf{(3.4 \pm 0.3) \times 10^{-8}}$ |

ture error and unique feature error in Figure 1. It can be observed that after a few iterations[1], the errors decrease linearly, which is consistent with our theoretical guarantee in Theorem 2.

To directly verify the linear convergence guarantee of Theorem 2, we record the training loss $\tilde{f}$ at every epoch and plot the optimality gap on a log scale in Figure 2. It is clear that at the terminal stage of the convergence, the gap decreases linearly in log scale, confirming the exponential decay predicted by the theorem.

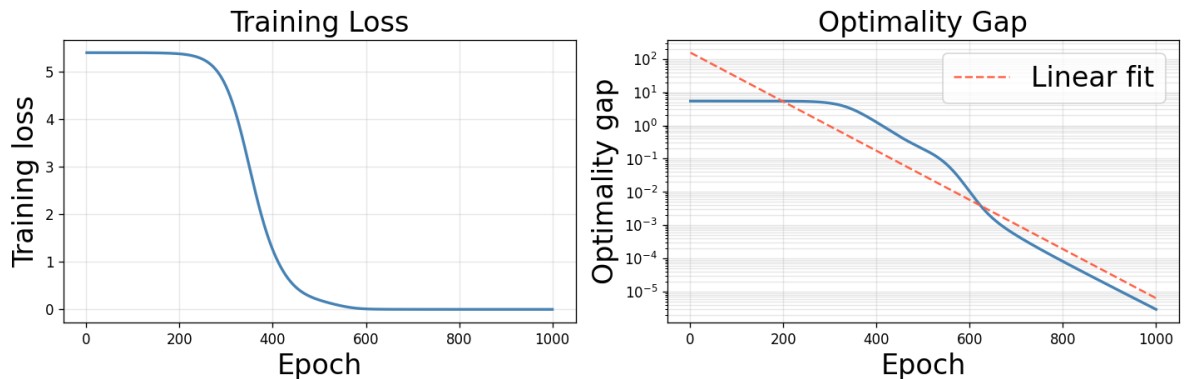

Figure 2: Convergence of HMF on the synthetic dataset. *Left*: training loss vs. epoch. *Right*: optimality gap on a log scale, confirming the linear convergence of Theorem 2. The linear fit has slope $-0.0170$.

**Changing ratios of missing entries**   We compare HMF with three representative baseline algorithms on the synthetic dataset under different ratios of missingness. Specifically, we uniformly randomly remove a subset of entries in each $\mathbf{M}_{(i)}$. Then we run HMF with $\ell^{pse}$, JIVE, RJIVE, and perPCA on the partially observed $\{\mathbf{M}_{(i)}\}$ and calculate the subspace error defined as $\left\|\mathbf{P}_{\mathbf{U}_g} - \mathbf{P}_{\mathbf{U}^\star_g}\right\|_F^2 + \frac{1}{N}\sum_{i=1}^N \left\|\mathbf{P}_{\mathbf{U}_{(i),l}} - \mathbf{P}_{\mathbf{U}^\star_{(i),l}}\right\|_F^2$. The missing entries are treated as zero in JIVE, RJIVE, and perPCA. We run the experiments from 3 different random seeds and calculate the mean and standard deviation of the subspace error. Results are reported in Table. 2. In Table. 2, HMF has consistently low subspace error, even when the ratio of missing elements is as high as 50%. The result highlights HMF's superior ability to recover shared and unique features from incomplete data.

**Full baseline comparison on the synthetic dataset.**   We further compare HMF against AJIVE, Pro-JIVE, MOFA+, and BIDIFAC on the same synthetic dataset with 10% entries randomly missing. We report the global subspace error $\|\mathbf{P}_{\mathbf{U}^\star_g} - \mathbf{P}_{\hat{\mathbf{U}}_g}\|_F^2$ and mean local subspace error $\frac{1}{N}\sum_i \|\mathbf{P}_{\mathbf{U}^\star_{(i),l}} - \mathbf{P}_{\hat{\mathbf{U}}_{(i),l}}\|_F^2$. Results are reported in Table 3.

HMF and BIDIFAC achieve near-perfect global subspace recovery, while all other methods incur substantially larger errors. This is expected as HMF and BIDIFAC explicitly use the projected square error loss while other approaches lack direct mechanisms to handle missingness.

---

[1]The initial sublinear convergence of the algorithm is due to the fact that our initial point does not satisfy the condition of Theorem 2. However, once the iterations reach the basin of attraction of the global solution, the iterations converge linearly.

Table 3: Subspace recovery error on the synthetic dataset with full baseline comparison.

| Method | Global error | Local error |
|---|---|---|
| HMF | $\mathbf{< 10^{-4}}$ | $\mathbf{< 10^{-4}}$ |
| BIDIFAC | $< 10^{-4}$ | $5 \times 10^{-4}$ |
| perPCA | $5.3 \times 10^{-3}$ | $4.8 \times 10^{-2}$ |
| JIVE | $5.3 \times 10^{-3}$ | $4.9 \times 10^{-2}$ |
| ProJIVE | $6.6 \times 10^{-3}$ | $4.9 \times 10^{-2}$ |
| AJIVE | $2.7 \times 10^{-2}$ | $5.4 \times 10^{-2}$ |
| MOFA+ | $0.244$ | $2.84$ |

**Ablation study on the role of orthogonality.** The orthogonality constraint $\mathbf{U}_g^\top \mathbf{U}_{(i),l} = 0$ is central to HMF's identifiability and convergence. To quantify its importance, we run HMF with and without the orthogonality correction step on the synthetic dataset and plot the subspace recovery loss in Figure 3. Without the correction, the reconstruction loss still decreases, but the recovered subspaces remain inaccurate. The comparison in Figure 3 highlights the necessity of the correction step, thus of maintaining the constraint $\mathbf{U}_g^\top \mathbf{U}_{(i),l} = 0$, for accurate factor separation.

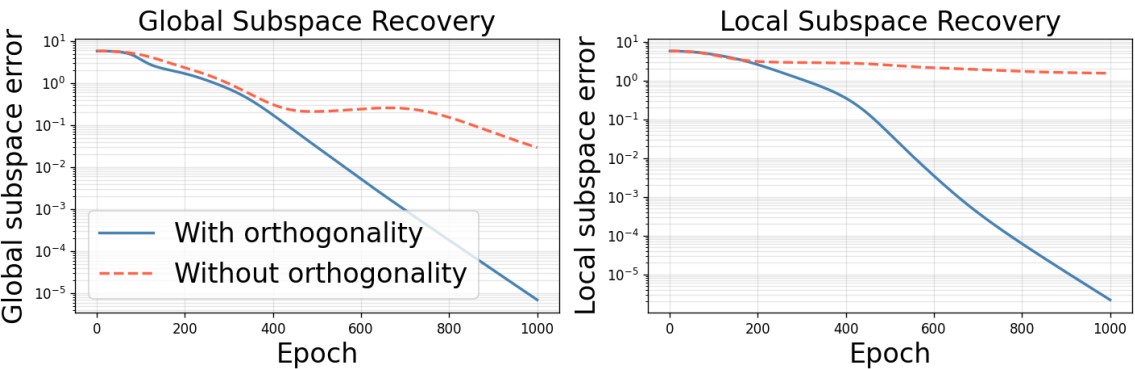

Figure 3: HMF with vs. without the orthogonality correction step. The correction enables near-exact recovery of both shared and local subspaces. Without orthogonalization, the local subspace error remains large even after 1000 epochs.

## 6.2 Case study: video segmentation from incomplete entries

We use an illustrative example in video segmentation to demonstrate the application of HMF. The video comes from a simulated surveillance dataset (Vacavant et al., 2013). In the video, multiple vehicles drive through a roundabout. We uniformly randomly remove 40% pixels to simulate the effects of missing observations. We divide each video frame $i$ into multiple $7 \times 7$ patches and flatten these patches into row vectors to construct observation matrix $\mathbf{M}_{(i)}$. Then, we apply HMF to identify the shared and unique signals from the observation matrices with $r_1 = 20$ and $r_{2,(i)} = r_2 = 30$. Naturally, the shared signals will correspond to the stationary components in the video, and unique signals correspond to the changing parts. We reconstruct the frames from the shared and unique signals and plot the results in Figure 4.

The first row of Figure 4 shows 5 sample frames from the video. They are corrupted as many pixels are missing. The second row plots the reconstructed $\mathbf{U}_g \mathbf{V}_{(i),g}^T$. One can see that HMF clearly identifies the fine details on the background. The third row of Figure 4 shows the reconstructed $\mathbf{U}_{(i),l} \mathbf{V}_{(i),l}^T$, which also conspicuously show the moving cars.

For numerical comparisons, we evaluate all methods under three experimental regimes that address the role of missing-data handling:

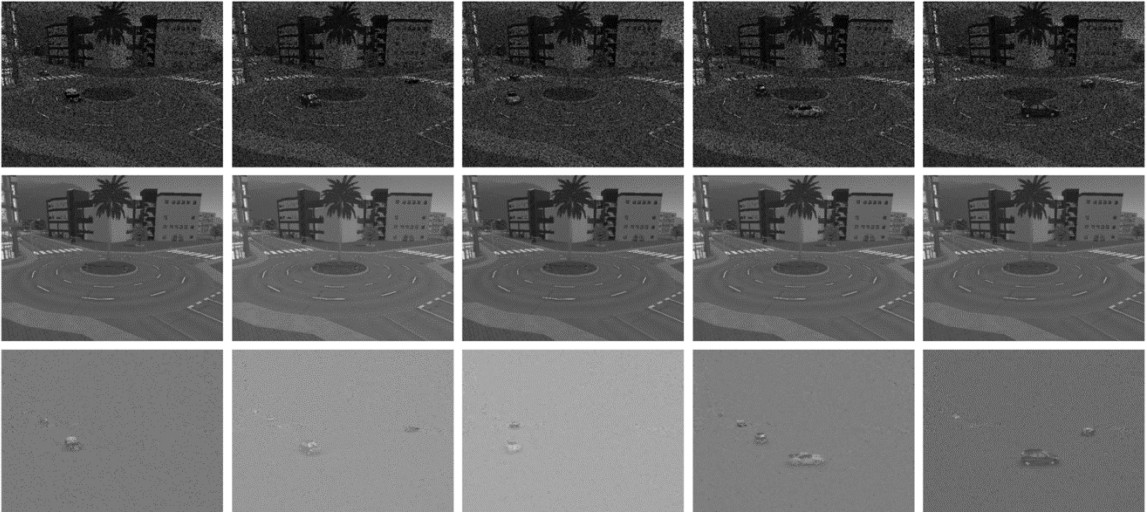

Figure 4: Video Segmentation. 5 frames from the video are plotted.

- **Scenario A (fully observed):** Every algorithm receives the original frames.

- **Scenario B (40% missing, zero-fill):** Algorithms use zero-filled frames as the observation. `HMF` uses the observation mask for the masked reconstruction objective.

- **Scenario C (40% missing, NN imputation):** Missing pixels are filled by spatial nearest-neighbor imputation.

Scenario A is the fully observed performance ceiling. Scenario B and C explore different data imputation strategies. In addition to JIVE, RJIVE, and perPCA from the visualization experiment, we include AJIVE (Feng et al., 2018), ProJIVE (Murden et al., 2026), MOFA+ (Argelaguet et al., 2020), BIDIFAC (Park & Lock, 2020), Stacked RPCA, and SoftImpute+JIVE (Mazumder et al., 2010) as additional baselines. The *nearest-neighbor imputation* for Scenario C fills each missing pixel from the nearest observed neighbor using expanding windows until all pixels are filled.

Results are shown in Tables 4–6. A higher level of quality in the reconstructed background and foreground is typically associated with a lower MSE and a higher PSNR.

Table 4: Scenario A: fully observed video. B = background, F = foreground. Best results in bold.

| Method | BMSE | BPSNR | BSSIM | FMSE | FPSNR | FSSIM |
|---|---|---|---|---|---|---|
| perPCA | 0.1211 | 9.17 | 0.0578 | **0.0077** | **21.14** | **0.7557** |
| JIVE | 0.0023 | 26.37 | 0.7091 | 0.0825 | 10.84 | 0.0708 |
| AJIVE | 0.0309 | 15.11 | 0.4209 | 0.0595 | 12.26 | 0.3105 |
| ProJIVE | 0.0011 | 29.68 | 0.7982 | 0.0890 | 10.51 | 0.0356 |
| MOFA+ | 0.1240 | 9.06 | 0.0341 | 0.1846 | 7.34 | 0.0007 |
| BIDIFAC | 0.0035 | 24.61 | 0.6586 | 0.0784 | 11.05 | 0.1067 |
| Stacked RPCA | 0.0030 | 25.16 | 0.6827 | 0.0820 | 10.86 | 0.0397 |
| `HMF` | **0.0011** | **29.71** | **0.7988** | 0.0905 | 10.44 | 0.0323 |

From the comparisons, it is evident that `HMF` consistently achieves lower MSE and higher PSNR values for background reconstructions.

Table 5: Scenario B: 40% missing. Missing entries are filled with zero.

| Method | BMSE | BPSNR | BSSIM | FMSE | FPSNR | FSSIM |
|---|---|---|---|---|---|---|
| perPCA | 0.0236 | 16.26 | 0.5053 | 0.1268 | 8.97 | 0.0776 |
| JIVE | 0.0248 | 16.05 | 0.4808 | 0.1096 | 9.60 | 0.0759 |
| RJIVE | 0.0231 | 16.36 | — | 0.1126 | 9.48 | — |
| AJIVE | 0.0235 | 16.28 | 0.5165 | 0.1114 | 9.53 | 0.0760 |
| ProJIVE | 0.0232 | 16.34 | 0.5528 | 0.1151 | 9.39 | 0.0741 |
| MOFA+ | 0.1304 | 8.85 | 0.0082 | 0.1846 | 7.34 | 0.0007 |
| BIDIFAC | 0.0263 | 15.81 | 0.4710 | 0.1084 | 9.65 | 0.0772 |
| Stacked RPCA | 0.0054 | 22.68 | 0.6307 | 0.1149 | 9.40 | 0.0678 |
| SoftImpute+JIVE | 0.0241 | 16.17 | 0.5006 | 0.1101 | 9.58 | 0.0811 |
| **HMF** | **0.0028** | **25.59** | **0.6847** | **0.1035** | **9.85** | **0.0798** |

Table 6: Scenario C: 40% missing. Missing entries are filled with by nearest neighbor imputation.

| Method | BMSE | BPSNR | BSSIM | FMSE | FPSNR | FSSIM |
|---|---|---|---|---|---|---|
| perPCA | 0.1219 | 9.14 | 0.0465 | **0.0110** | **19.58** | **0.6276** |
| JIVE | 0.0025 | 26.05 | 0.6982 | 0.1008 | 9.97 | 0.0658 |
| AJIVE | 0.0021 | 26.78 | 0.7411 | 0.0925 | 10.34 | 0.0374 |
| ProJIVE | 0.0015 | 28.27 | 0.7646 | 0.1091 | 9.62 | 0.0286 |
| MOFA+ | 0.1260 | 9.00 | 0.0238 | 0.1846 | 7.34 | 0.0007 |
| BIDIFAC | 0.0034 | 24.67 | 0.6613 | 0.0971 | 10.13 | 0.1003 |
| Stacked RPCA | 0.0033 | 24.87 | 0.6564 | 0.0915 | 10.39 | 0.0297 |
| SoftImpute+JIVE | 0.0241 | 16.18 | 0.4998 | 0.1102 | 9.58 | 0.0810 |
| **HMF +NN init** | **0.0014** | **28.64** | **0.7755** | 0.1089 | 9.63 | 0.0266 |

### 6.3 Case Study: Stock Market Data

HMF can also be applied to data from the financial market. As a proof-of-concept, we analyze the daily stock prices of 214 stocks from January 2, 1990 to November 9, 2017. The goal is to understand the time-specific patterns in stock prices. These patterns are often related to abnormal market behaviors and provide insightful information for subsequent trading decisions.

Similar to Fattahi & Gomez (2021), we use a time window of 30 days to group the stock prices into different batches, and analyze the common and unique features among these batches. Each batched data matrix $\mathbf{M}_{(i)}$ has dimension $214 \times 30$. The unique features represent structural differences in the stock prices in each batch, thus signaling sudden changes in the market. To find the unique features, we apply HMF on the batched data to extract $\mathbf{U}_{(i),l}$'s and $\mathbf{V}_{(i),l}$'s. To measure the "heterogeneity index" in each batch, we

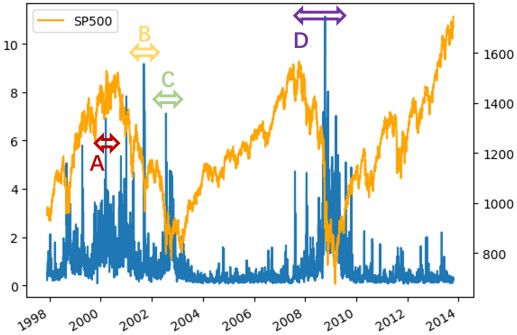

Figure 5: The blue curve denotes the heterogeneity index of 214 stock returns from 1998 to 2014. The orange curve is the *SP500* closing prices at the corresponding dates. We label 4 periods where the heterogeneity index has large peaks

calculate the column-wise $\ell_1$ norm of the signal matrix $\mathbf{U}_{(i),l}\mathbf{V}_{(i),l}^T$ for each $i$. The heterogeneity indices are plotted as the blue curve in Figure 5. To provide better insight, we also plot the *SP500* closing prices in the same figure.

From Figure 5, one can observe that almost every significant historical market crash (shown as sudden drops in the orange curve) corresponds to a peak in the heterogeneity index. We also identify 4 major periods when the heterogeneity index has several large peaks. By comparing these periods with the history of the global financial market (Wikipedia, 2023), one can see that A corresponds to the "dot-com bubble", B corresponds to the "September 11 attack", C corresponds to "stock market downturn of 2002", and D corresponds to the "2007-2008 financial crisis and the aftermath".

**Quantitative baseline comparison.** To complement the interpretability analysis, we add a matrix-completion on the time-series stocks return data. We use the daily adjusted-close log-returns for 50 S&P 500 stocks across five GICS sectors (Technology, Healthcare, Financials, Energy, Utilities) from January 2020 to December 2023. Each sector's data is an observation matrix $\mathbf{M}_{(i)} \in \mathbb{R}^{1005 \times 10}$. We randomly hold out 20% of entries and measure held-out reconstruction MSE. Baselines that require complete data, including JIVE, AJIVE, ProJIVE, perPCA receive zero-filled training matrices.

Table 7: Stock-market matrix completion.

| Method | Held-out MSE |
|---|---|
| **HMF** | **1.165** |
| AJIVE | 1.879 |
| ProJIVE | 2.020 |
| JIVE | 2.141 |
| BIDIFAC | 4.456 |
| perPCA | 3.335 |

`HMF` achieves the best held-out MSE, confirming that the projected square error loss used in `HMF` improves predictive accuracy on held-out returns.

## 6.4 Case study: rating prediction on MovieLens

The HMF framework is also applicable to recommender systems. We include the MovieLens-100k dataset (Harper & Konstan, 2015) as an illustrative case study showing that `HMF` extends naturally to sparse, non-uniformly observed data via the masked matrix completion objective. The dataset contains $10^5$ ratings (1-5) on 1682 movies from 943 users. We note that state-of-the-art recommender systems rely on graph-based collaborative filtering (Rashed et al., 2019; Darban & Valipour, 2022), contrastive learning (Zhang et al., 2024), and generative models (Rajput et al., 2023). The use of `HAE` in recommender systems only illustrates that the `HMF` framework extends naturally to sparse and non-uniformly sampled data.

In MovieLens-100k, each movie is labeled with genre labels, such as action, adventure, or animation. There are 19 genres in total. Notice that each movie can have more than one genre label. For example, the movie *Titanic* belongs to both the romance and drama genres. To characterize the genre information, we first cluster movies into different groups according to the genre labels. Different groups represent different sources. More specifically, for movie $m$, we use multi-hot encoding to create its genre information vector $g_m \in \mathbb{R}^{19}$, whose $j$-th element is 1 if movie $m$ belongs to genre $j$. Next, we normalize $g_m$ and use K-means clustering to cluster the normalized $g_m$s into 10 groups. For each group, we construct user-movie rating matrices $\mathbf{M}_{(i)}$. Each $\mathbf{M}_{(i)}$ has 943 rows, and its number of columns is equal to the number of movies in cluster $i$. Thus $\mathbf{M}_{(i)}$ is highly sparse. Then we randomly split each $\Omega_{(i)}$ into 90% training set $\Omega_{(i)}^{train}$ and 10% test set $\Omega_{(i)}^{test}$ and apply `HMF` with $\ell^{pse}$ on the $\Omega_{(i)}^{train}$.

We use two methods to predict the user ratings on unseen movies: (i) collaborative filtering, which is based on linear model equation 3, and (ii) auto-encoders, which is based on the nonlinear model equation 5.

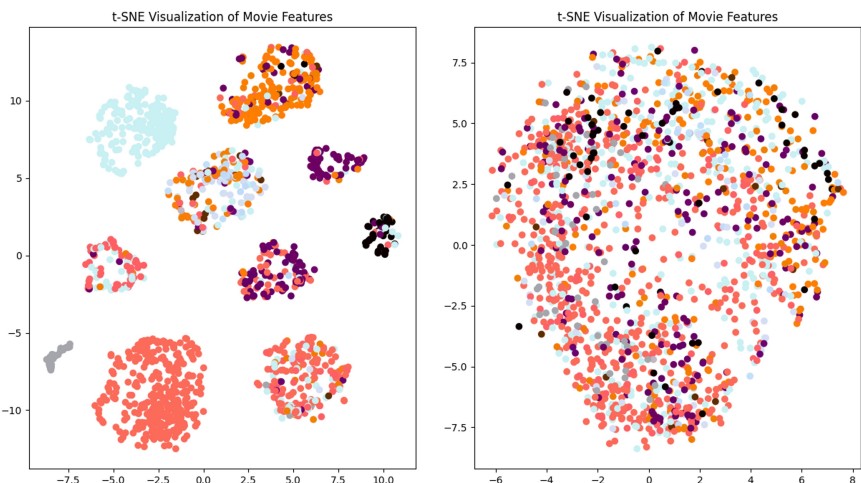

Figure 6: t-SNE plots of movie embeddings. Each dot represents a movie. Its color denotes one genre of the movie. *Left*: t-SNE of embeddings from `HAE`. *Right*: t-SNE of embeddings from standard auto-encoders

*Collaborative filtering* The standard collaborative filtering method pools ratings for movies from all genres and exploits matrix completion to extract the low-rank movie features and user features on $\cup_i \Omega_{(i)}^{train}$ (Koren et al., 2009). These features can then be leveraged for new rating predictions on unseen movies. Though achieving decent performance, the standard collaborative filtering only finds generic movie features and neglects the genre information. To incorporate the genre information, we apply `HMF` to the genre-clustered ratings $\mathbf{M}_{(i)}$ to identify both generic features and genre-specific features. The new predictions are given as the entries of $\hat{\mathbf{U}}_g \hat{\mathbf{V}}_{(i),g}^T + \hat{\mathbf{U}}_{(i),l} \hat{\mathbf{V}}_{(i),l}^T$.

*Auto-encoder* As discussed in Section 3.2, auto-encoders are nonlinear extensions to matrix factorization models and can potentially encode richer information. The auto-encoders have been employed in the rating prediction tasks and have been shown to outperform collaborative filtering based on linear features (Muller et al., 2018). The standard auto-encoder approach for rating prediction minimizes the following reconstruction loss,

$$\min_{\mathbf{U},\mathbf{V}} \left\| \mathcal{P}_{\Omega_{train}} \left( \mathbf{M} - \mathtt{Net_V} \left( \mathtt{Net_U} \left( \mathbf{M}^{obs} \right) \right) \right) \right\|_F^2 + \mathcal{R}\left( \mathbf{U}, \mathbf{V} \right) \quad , \tag{10}$$

where $\mathbf{M}^{obs}$ is the observed rating matrix $\mathbf{M}$ padded with zero in all entries outside the training set $\Omega_{train}$. $\mathcal{R}\left( \mathbf{U}, \mathbf{V} \right)$ is the regularization term dependent on the architecture of the neural network.

There are numerous works focusing on the architecture of the encoder and decoder network (Zhang et al., 2017; Darban & Valipour, 2022; Rashed et al., 2019). Among them, sparsely supported fully connected neural network (SparseFC) (Muller et al., 2018) achieves decent performance. We briefly introduce the main idea of sparse FC for consistency. A standard full-connected neural network consists of multiple layers. Layer $l$ will transform the input $x^{(l-1)}$ to the output $x^{(l)}$ via a linear mapping $W^{(l)}$ followed by a nonlinear activation $\sigma^{(l)}$, i.e., $x^{(l)} = \sigma^{(l)}(W^{(l)}x^{(l-1)})$. A sparse kernel layer chooses a different parametrization of the linear mapping $W^{(l)}$. More specifically, the $j$-th element of the output $x^{(l)}$ is given by, $x_j^{(l)} = \sigma^{(l)} \left( \sum_i \alpha_i^{(l)} K(u_i^{(l)}, u_j^{(l)}) x_i^{(l-1)} \right)$, where $\alpha_i^{(l)}$, $u_i^{(l)}$, and $u_j^{(l)}$ are trainable parameters. $K$ is a kernel function that can be defined as the Gaussian kernel $K(x,y) = \exp\left( -\gamma \left\| x - y \right\|^2 \right)$. Intuitively, such parametrization would make the weights $K(u_i^{(l)}, u_j^{(l)})$ sparse. Sparse FC networks are constructed by sequentially applying sparse kernel layers. Because of their superior generalization performance, we use sparse neural networks as the backbones for the encoder and decoder networks $\mathtt{Net_U}$ and $\mathtt{Net_V}$ in equation 10. We implement the architecture and pool $\mathbf{M}_{(i)}$ to build $\mathbf{M}$, then solve equation 10 with AdamW (Loshchilov & Hutter, 2019).

As introduced in Section 3.2, we can also employ a heterogeneous version of auto-encoders `HAE`. More specifically, we use a shared encoder network $\mathtt{Net_{U_g}}$ and 10 genre-specific encoders $\mathtt{Net_{U_{(i),l}}}$ to learn both shared

and unique nonlinear embeddings of movies. Then, we use a decoder $\texttt{Net}_{\mathbf{V}}$ to predict user ratings. We still use the sparse FC network as the backbones of the encoders and decoders.

To visualize the embeddings from standard auto-encoders and heterogeneous auto-encoders, we show t-SNE plots of different movie embeddings in Figure 6.

In Figure 6, it is clear that the embeddings from $\texttt{HAE}$ form clusters dependent on the genre, while the embeddings from standard auto-encoders do not have a clear structure. These observations are consistent with our intuition that movies with similar genres should have similar embeddings.

With the fitted models from $\texttt{HMF}$ or $\texttt{HAE}$, we can predict user ratings on unseen movies. To evaluate the predictive performance, we calculate the predicted ratings on the test set $\Omega_{(i)}^{test}$ and compare it with the ground truth. As the Movielens-100k dataset is extensively analyzed in the literature, we also report the RMSE of a few representative benchmark methods detailed as follows,

- Matrix completion (Koren et al., 2009): We implement the standard matrix completion on the pooled rating matrix $\mathbf{M}$ to extract linear features.

- AutoSVD++ (Zhang et al., 2017): AutoSVD++ is a hybrid model combining a variant of biased SVD and a 1-layer fully connected neural network.

- GHRS (Darban & Valipour, 2022): Graph-based Hybrid Recommendation System builds user similarity graph based on user ratings and side information, including age and occupation, then clusters users into different groups and uses group features to make predictions on the unseen ratings.

- GraphRec (Rashed et al., 2019): GraphRec is a nonlinear model that uses a neural network to create user and movie embeddings. It also considers the graph-based features from the user similarity graph.

The root mean square error (RMSE) of the prediction on the test set is shown in Table 8. For matrix completion, $\texttt{HMF}$, Sparse FC, and $\texttt{HAE}$, we run each experiment from 80 different random seeds and report the mean and standard deviation. For other methods, we report the performance from the literature.

Table 8: Prediction root mean square error on Movielens-100K (10% randomly selected test set).

| Method | Source | RMSE |
|---|---|---|
| Matrix completion | Koren et al. (2009) | $0.981 \pm 0.001$ |
| HMF | Ours | $0.977 \pm 0.001$ |
| AutoSVD++ | Zhang et al. (2017) | $0.901$ |
| GHRS | Darban & Valipour (2022) | $0.887$ |
| GraphRec | Rashed et al. (2019) | $0.883$ |
| Sparse FC | Muller et al. (2018) | $0.8838 \pm 0.0006$ |
| HAE | Ours | $\mathbf{0.8800} \pm 0.0006$ |

In Table 8, it is clear that $\texttt{HAE}$ has the lowest test error, suggesting that the heterogeneous features learned by $\texttt{HAE}$ are more accurate in terms of predicting the users' preferences. Therefore, the genre features bring useful information to the movie ratings prediction.

## 6.5 Rank Determination

The ranks $r_1$ and $r_2$ are hyperparameters of $\texttt{HMF}$. In practice, we recommend selecting them via cross-validation. Specifically, we randomly select 20% of the observed entries as the validation set, and fit the model on the remaining 80% of the observation matrix. The pair $(r_1, r_2)$ that minimizes the validation MSE is selected as the rank. We evaluate the rank-selection mechanism on the synthetic and video segmentation dataset.

**Synthetic dataset.** We sweep $(r_1, r_2) \in \{1, \ldots, 6\}^2$ and follow the cross-validation procedure. The MSE, global subspace error, and local subspace errors are plotted in Figure 7. Both local and global subspace errors are robust to small over-specification in the rank $r_2$ as the global subspace error at $(3, 4)$ is $2.6 \times 10^{-5}$. From Figure 7, the validation MSE is even more robust to rank overspecification: once the global rank $r_1 \geq 3$, both validation MSE and global error are near zero, illustrating robustness to local rank over-specification.

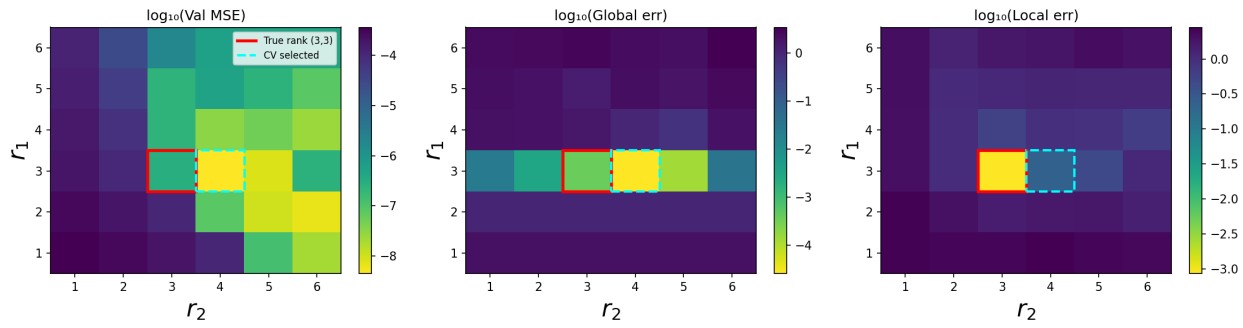

Figure 7: Rank selection on the synthetic dataset. *Left*: Heatmap of $\log_{10}$ validation MSE. *Middle*: $\log_{10}$ Heatmap of global subspace error. *Right*: Heatmap of $\log_{10}$ local subspace error. Red solid box marks the true rank $(3, 3)$. Cyan dashed box marks the CV-selected rank $(3, 4)$.

**Video segmentation dataset.** We evaluate 8 rank pairs spanning total ranks 15–70. Results are shown in Table 9. CV selects $(r_1, r_2) = (30, 40)$. Across all pairs, BPSNR varies by only 0.67 dB, demonstrating that `HMF` is robust to different choices of ranks.

Table 9: Rank selection on the video dataset with 40% entries missing.

| $(r_1, r_2)$ | Total rank | MSE | BPSNR | BSSIM |
|---|---|---|---|---|
| $(5, 10)$ | 15 | 0.00652 | 22.19 | 0.548 |
| $(10, 15)$ | 25 | 0.00601 | 22.57 | 0.549 |
| $(15, 20)$ | 35 | 0.00587 | 22.68 | 0.550 |
| $(15, 30)$ | 45 | 0.00572 | 22.80 | 0.550 |
| $(20, 25)$ | 45 | 0.00578 | 22.76 | 0.550 |
| $(20, 30)$ | 50 | 0.00570 | 22.82 | 0.550 |
| $(25, 35)$ | 60 | 0.00567 | 22.85 | 0.550 |
| $(30, 40)$ | 70 | 0.00565 | 22.86 | 0.550 |

# 7 Conclusion and Discussion of Limitations

This work proposes `HMF` that solves a constrained matrix factorization problem to extract shared and unique features from heterogeneous data, and extends the model to auto-encoders. One avenue for future research is to consider scenarios where the rank of feature matrices $r_1$ and $r_2$ are unknown and must be over-estimated instead (Ma & Fattahi, 2022; Stöger & Soltanolkotabi, 2021). Such a setting (also known as overparameterization) has been extensively studied for the classical matrix factorization literature. Another promising direction is to improve the initialization condition for the guaranteed convergence of our proposed algorithm. Although the theoretical guarantee of our algorithm (Theorem 2) relies on the availability of a good initial point, we hypothesize that this requirement can be relaxed, as the algorithm works well in practice with a small and random initialization.

## 8 Acknowledgements

We are grateful to the AE and anonymous reviewers for their helpful suggestions. Salar Fattahi is supported, in part, by NSF CAREER grant 2337776 and ONR grant N00014-26-1-2074. Raed Al Kontar is supported, in part, by NSF CMMI grants 2328010 and 2144147. This research is also supported, in part, by an Amazon Research Award in Fall 2025.

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

## A  Appendix

This supplementary material presents the proofs of Theorems 1, 2, and 3 from the main paper. We first present Table 10 to illustrate the dependence structure of the lemmas and connect each theorem to the lemmas that directly support it.

Table 10: Dependency guide.

| Lemma | Content summary | Used in |
|-------|-----------------|---------|
| C.4 | KKT conditions (SE loss) | Thm. 1, Thm. 2 |
| C.1 | Noise assumption $\Rightarrow$ properties of optimal solution | Thm. 2 |
| C.2 | Lipschitz constant of $\tilde{f}$ (SE) | Thm. 2 |
| C.3 | Sufficient descent per gradient step | Thm. 2 |
| C.5 | Subspace perturbation bound | Thm. 2 |
| C.6 | Optimality gap upper-bounds deviation | Thm. 2 |
| C.7 | $\mathbf{U}$ iterates stay near optimum | Thm. 2 |
| C.8 | $\mathbf{V}$ iterates stay near optimum | Thm. 2 |
| C.9 | Gradient aligns with descent direction | Thm. 2 |
| C.10 | Polyak–Łojasiewicz inequality | Thm. 2 (final step) |
| D.1 | KKT conditions (general $\ell$) | Thm. 3 |
| D.2 | Lipschitz bound (general $\ell$) | Thm. 3 |

## B  Proof of Theorem 1

For a matrix $\mathbf{A} \in \mathbb{R}^{m \times n}$, we use $\|\mathbf{A}\|$ or $\|\mathbf{A}\|_2$ to denote $\mathbf{A}$'s operator norm and $\|\mathbf{A}\|_F$ to denote its Frobenius norm.

The proof mainly consists of a perturbation analysis of the objective (4) in the main paper. We assume that the loss metric is the square loss $\ell^{se}$ throughout this section. We firstly estimate the subspace error of $\left\|\mathbf{P}_{\hat{\mathbf{U}}_g} - \mathbf{P}_{\mathbf{U}^\star_g}\right\|_F$ and $\left\|\mathbf{P}_{\hat{\mathbf{U}}_{(i),l}} - \mathbf{P}_{\mathbf{U}^\star_{(i),l}}\right\|_F$. Then we estimate the reconstruction errors.

The KKT conditions for the optimal $\hat{\mathbf{V}}_{(i),g}$ and $\hat{\mathbf{V}}_{(i),l}$ are,

$$\left(\mathbf{M}_{(i)} - \hat{\mathbf{U}}_g\hat{\mathbf{V}}_{(i),g}^T - \hat{\mathbf{U}}_{(i),l}\hat{\mathbf{V}}_{(i),l}^T\right)^T \hat{\mathbf{U}}_g = 0$$

$$\left(\mathbf{M}_{(i)} - \hat{\mathbf{U}}_g\hat{\mathbf{V}}_{(i),g}^T - \hat{\mathbf{U}}_{(i),l}\hat{\mathbf{V}}_{(i),l}^T\right)^T \hat{\mathbf{U}}_{(i),l} = 0$$

Considering the constraint $\hat{\mathbf{U}}_g^T\hat{\mathbf{U}}_{(i),l} = 0$, we can solve $\hat{\mathbf{V}}_{(i),g}$ and $\hat{\mathbf{V}}_{(i),l}$ as,

$$\hat{\mathbf{V}}_{(i),g} = \mathbf{M}_{(i)}^T\hat{\mathbf{U}}_g\left(\hat{\mathbf{U}}_g^T\hat{\mathbf{U}}_g\right)^{-1}$$

$$\hat{\mathbf{V}}_{(i),l} = \mathbf{M}_{(i)}^T\hat{\mathbf{U}}_{(i),l}\left(\hat{\mathbf{U}}_{(i),l}^T\hat{\mathbf{U}}_{(i),l}\right)^{-1}$$

Then the objective becomes,

$$
\tilde{f}_i(\hat{\mathbf{U}}_g, \{\hat{\mathbf{U}}_{(i),l}\}) = \frac{1}{2}\left\|\mathbf{M}_{(i)} - \hat{\mathbf{U}}_g\left(\hat{\mathbf{U}}_g^T\hat{\mathbf{U}}_g\right)^{-1}\hat{\mathbf{U}}_g^T\mathbf{M}_{(i)} - \hat{\mathbf{U}}_{(i),l}\left(\hat{\mathbf{U}}_{(i),l}^T\hat{\mathbf{U}}_{(i),l}\right)^{-1}\hat{\mathbf{U}}_{(i),l}\mathbf{M}_{(i)}\right\|_F^2
$$
$$
+ \frac{\beta}{2}\left\|\hat{\mathbf{U}}_g^T\hat{\mathbf{U}}_g - \mathbf{I}\right\|_F^2 + \frac{\beta}{2}\left\|\hat{\mathbf{U}}_{(i),l}^T\hat{\mathbf{U}}_{(i),l} - \mathbf{I}\right\|_F^2
$$
$$
= \frac{1}{2}\left\|\mathbf{M}_{(i)} - \mathbf{P}_{\hat{\mathbf{U}}_g}\mathbf{M}_{(i)} - \mathbf{P}_{\hat{\mathbf{U}}_{(i),l}}\mathbf{M}_{(i)}\right\|_F^2 + \frac{\beta}{2}\left\|\hat{\mathbf{U}}_g^T\hat{\mathbf{U}}_g - \mathbf{I}\right\|_F^2 + \frac{\beta}{2}\left\|\hat{\mathbf{U}}_{(i),l}^T\hat{\mathbf{U}}_{(i),l} - \mathbf{I}\right\|_F^2
$$

Notice that if $\hat{\mathbf{U}}_g$ and $\{\hat{\mathbf{U}}_{(i),l}\}$ are optimal, we must have $\hat{\mathbf{U}}_g^T\hat{\mathbf{U}}_g = \mathbf{I}$ and $\hat{\mathbf{U}}_{(i),l}^T\hat{\mathbf{U}}_{(i),l} = \mathbf{I}$, since we can always use Schmidt-Gram procedure to orthonormalize the columns of $\hat{\mathbf{U}}_g$ and $\{\hat{\mathbf{U}}_{(i),l}\}$ without changing $\mathbf{P}_{\hat{\mathbf{U}}_g}$ and $\{\mathbf{P}_{\hat{\mathbf{U}}_{(i),l}}\}$. Therefore without loss of generality, we consider the problem

$$
\min_{\hat{\mathbf{U}}_g, \{\hat{\mathbf{U}}_{(i),l}\}} \sum_{i=1}^N \left\|\mathbf{M}_{(i)} - \mathbf{P}_{\hat{\mathbf{U}}_g}\mathbf{M}_{(i)} - \mathbf{P}_{\hat{\mathbf{U}}_{(i),l}}\mathbf{M}_{(i)}\right\|_F^2
$$
$$
\text{such that } \hat{\mathbf{U}}_g^T\hat{\mathbf{U}}_{(i),l} = 0, \quad \forall i \in [N]
$$

By our assumption on the data generation process, $\mathbf{U}^\star{}_g$ and $\{\mathbf{U}^\star{}_{(i),l}\}$'s are also feasible. Thus the objective evaluated at $\mathbf{U}^\star{}_g$ and $\{\mathbf{U}^\star{}_{(i),l}\}$'s cannot be smaller than the optimal objective,

$$
\sum_{i=1}^N \left\|\mathbf{M}_{(i)} - \mathbf{P}_{\hat{\mathbf{U}}_g}\mathbf{M}_{(i)} - \mathbf{P}_{\hat{\mathbf{U}}_{(i),l}}\mathbf{M}_{(i)}\right\|_F^2 \leq \sum_{i=1}^N \left\|\mathbf{M}_{(i)} - \mathbf{P}_{\mathbf{U}^\star{}_g}\mathbf{M}_{(i)} - \mathbf{P}_{\mathbf{U}^\star{}_{(i),l}}\mathbf{M}_{(i)}\right\|_F^2
$$

This is equivalent to

$$
\sum_{i=1}^N \text{Tr}\left(\left(\mathbf{P}_{\hat{\mathbf{U}}_g} + \mathbf{P}_{\hat{\mathbf{U}}_{(i),l}}\right)\mathbf{M}_{(i)}\mathbf{M}_{(i)}^T\right) \geq \sum_{i=1}^N \text{Tr}\left(\left(\mathbf{P}_{\mathbf{U}^\star{}_g} + \mathbf{P}_{\mathbf{U}^\star{}_{(i),l}}\right)\mathbf{M}_{(i)}\mathbf{M}_{(i)}^T\right) \tag{11}
$$

On the other hand, we can expand $\mathbf{M}_{(i)}\mathbf{M}_{(i)}^T$ as

$$
\mathbf{M}_{(i)}\mathbf{M}_{(i)}^T = \left(\mathbf{U}^\star{}_g\mathbf{V}^\star{}_{(i),g}^T + \mathbf{U}^\star{}_{(i),l}\mathbf{V}^\star{}_{(i),l}^T\right)\left(\mathbf{U}^\star{}_g\mathbf{V}^\star{}_{(i),g}^T + \mathbf{U}^\star{}_{(i),l}\mathbf{V}^\star{}_{(i),l}^T\right)^T
$$
$$
+ \left(\mathbf{U}^\star{}_g\mathbf{V}^\star{}_{(i),g}^T + \mathbf{U}^\star{}_{(i),l}\mathbf{V}^\star{}_{(i),l}^T\right)\mathbf{E}^\star{}_{(i)}^T + \mathbf{E}^\star{}_{(i)}\left(\mathbf{U}^\star{}_g\mathbf{V}^\star{}_{(i),g}^T + \mathbf{U}^\star{}_{(i),l}\mathbf{V}^\star{}_{(i),l}^T\right)^T + \mathbf{E}^\star{}_{(i)}\mathbf{E}^\star{}_{(i)}^T
$$

For simplicity, we use $\mathbf{F}^\star{}_{(i)}$ to denote

$$
\mathbf{F}^\star{}_{(i)} = \left(\mathbf{U}^\star{}_g\mathbf{V}^\star{}_{(i),g}^T + \mathbf{U}^\star{}_{(i),l}\mathbf{V}^\star{}_{(i),l}^T\right)\mathbf{E}^\star{}_{(i)}^T + \mathbf{E}^\star{}_{(i)}\left(\mathbf{U}^\star{}_g\mathbf{V}^\star{}_{(i),g}^T + \mathbf{U}^\star{}_{(i),l}\mathbf{V}^\star{}_{(i),l}^T\right)^T + \mathbf{E}^\star{}_{(i)}\mathbf{E}^\star{}_{(i)}^T
$$

Then the following holds,

$$
\text{Tr}\left(\mathbf{M}_{(i)}\mathbf{M}_{(i)}^T\left(\left(\mathbf{P}_{\mathbf{U}^\star{}_g} + \mathbf{P}_{\mathbf{U}^\star{}_{(i),l}}\right) - \left(\mathbf{P}_{\hat{\mathbf{U}}_g} + \mathbf{P}_{\hat{\mathbf{U}}_{(i),l}}\right)\right)\right)
$$
$$
= \text{Tr}\left(\left(\mathbf{U}^\star{}_g\mathbf{V}^\star{}_{(i),g}^T + \mathbf{U}^\star{}_{(i),l}\mathbf{V}^\star{}_{(i),l}^T\right)\left(\mathbf{U}^\star{}_g\mathbf{V}^\star{}_{(i),g}^T + \mathbf{U}^\star{}_{(i),l}\mathbf{V}^\star{}_{(i),l}^T\right)^T\left(\left(\mathbf{P}_{\mathbf{U}^\star{}_g} + \mathbf{P}_{\mathbf{U}^\star{}_{(i),l}}\right) - \left(\mathbf{P}_{\hat{\mathbf{U}}_g} + \mathbf{P}_{\hat{\mathbf{U}}_{(i),l}}\right)\right)\right)
$$
$$
+ \text{Tr}\left(\mathbf{F}^\star{}_{(i)}\left(\left(\mathbf{P}_{\mathbf{U}^\star{}_g} + \mathbf{P}_{\mathbf{U}^\star{}_{(i),l}}\right) - \left(\mathbf{P}_{\hat{\mathbf{U}}_g} + \mathbf{P}_{\hat{\mathbf{U}}_{(i),l}}\right)\right)\right) \tag{12}
$$

The first term in equation 12 is equal to

$$
\text{Tr}\left(\left(\mathbf{U}^\star{}_g\mathbf{V}^\star{}_{(i),g}^T + \mathbf{U}^\star{}_{(i),l}\mathbf{V}^\star{}_{(i),l}^T\right)\left(\mathbf{U}^\star{}_g\mathbf{V}^\star{}_{(i),g}^T + \mathbf{U}^\star{}_{(i),l}\mathbf{V}^\star{}_{(i),l}^T\right)^T\left(\mathbf{I} - \left(\mathbf{P}_{\hat{\mathbf{U}}_g} + \mathbf{P}_{\hat{\mathbf{U}}_{(i),l}}\right)\right)\right)
$$

From assumption (i) in Theorem 1, we know

$$\left(\mathbf{U}^{\star}_{g}\mathbf{V}^{\star T}_{(i),g}+\mathbf{U}^{\star}_{(i),l}\mathbf{V}^{\star T}_{(i),l}\right)\left(\mathbf{U}^{\star}_{g}\mathbf{V}^{\star T}_{(i),g}+\mathbf{U}^{\star}_{(i),l}\mathbf{V}^{\star T}_{(i),l}\right)^{T}\succeq\left(\mathbf{P}_{\mathbf{U}^{\star}_{g}}+\mathbf{P}_{\mathbf{U}^{\star}_{(i),l}}\right)\sigma^{2}_{\min}$$

Since $\left(\mathbf{I}-\left(\mathbf{P}_{\hat{\mathbf{U}}_{g}}+\mathbf{P}_{\hat{\mathbf{U}}_{(i),l}}\right)\right)$ is symmetric positive semidefinite, we have,

$$\mathrm{Tr}\left(\left(\mathbf{U}^{\star}_{g}\mathbf{V}^{\star T}_{(i),g}+\mathbf{U}^{\star}_{(i),l}\mathbf{V}^{\star T}_{(i),l}\right)\left(\mathbf{U}^{\star}_{g}\mathbf{V}^{\star T}_{(i),g}+\mathbf{U}^{\star}_{(i),l}\mathbf{V}^{\star T}_{(i),l}\right)^{T}\left(\mathbf{I}-\left(\mathbf{P}_{\hat{\mathbf{U}}_{g}}+\mathbf{P}_{\hat{\mathbf{U}}_{(i),l}}\right)\right)\right)$$
$$\geq\sigma^{2}_{\min}\mathrm{Tr}\left(\left(\mathbf{P}_{\mathbf{U}^{\star}_{g}}+\mathbf{P}_{\mathbf{U}^{\star}_{(i),l}}\right)\left(\mathbf{I}-\left(\mathbf{P}_{\hat{\mathbf{U}}_{g}}+\mathbf{P}_{\hat{\mathbf{U}}_{(i),l}}\right)\right)\right)$$

By Cauchy-Schwartz inequality, the second term in equation 12 is lower bounded by,

$$\mathrm{Tr}\left(\mathbf{F}^{\star}_{(i)}\left(\left(\mathbf{P}_{\mathbf{U}^{\star}_{g}}+\mathbf{P}_{\mathbf{U}^{\star}_{(i),l}}\right)-\left(\mathbf{P}_{\hat{\mathbf{U}}_{g}}+\mathbf{P}_{\hat{\mathbf{U}}_{(i),l}}\right)\right)\right)$$
$$\geq-\left\|\mathbf{F}^{\star}_{(i)}\right\|_{F}\left\|\left(\left(\mathbf{P}_{\mathbf{U}^{\star}_{g}}+\mathbf{P}_{\mathbf{U}^{\star}_{(i),l}}\right)-\left(\mathbf{P}_{\hat{\mathbf{U}}_{g}}+\mathbf{P}_{\hat{\mathbf{U}}_{(i),l}}\right)\right)\right\|_{F}$$
$$=-\sqrt{2}\left\|\mathbf{F}^{\star}_{(i)}\right\|_{F}\sqrt{\mathrm{Tr}\left(\left(\mathbf{P}_{\mathbf{U}^{\star}_{g}}+\mathbf{P}_{\mathbf{U}^{\star}_{(i),l}}\right)\left(\mathbf{I}-\left(\mathbf{P}_{\hat{\mathbf{U}}_{g}}+\mathbf{P}_{\hat{\mathbf{U}}_{(i),l}}\right)\right)\right)}$$

Combining them, we can derive a lower bound on equation 12 as

$$\mathrm{Tr}\left(\mathbf{M}_{(i)}\mathbf{M}^{T}_{(i)}\left(\left(\mathbf{P}_{\mathbf{U}^{\star}_{g}}+\mathbf{P}_{\mathbf{U}^{\star}_{(i),l}}\right)-\left(\mathbf{P}_{\hat{\mathbf{U}}_{g}}+\mathbf{P}_{\hat{\mathbf{U}}_{(i),l}}\right)\right)\right)$$
$$\geq\sigma^{2}_{\min}\mathrm{Tr}\left(\left(\mathbf{P}_{\mathbf{U}^{\star}_{g}}+\mathbf{P}_{\mathbf{U}^{\star}_{(i),l}}\right)\left(\mathbf{I}-\left(\mathbf{P}_{\hat{\mathbf{U}}_{g}}+\mathbf{P}_{\hat{\mathbf{U}}_{(i),l}}\right)\right)\right)$$
$$-\sqrt{2}\left\|\mathbf{F}^{\star}_{(i)}\right\|_{F}\sqrt{\mathrm{Tr}\left(\left(\mathbf{P}_{\mathbf{U}^{\star}_{g}}+\mathbf{P}_{\mathbf{U}^{\star}_{(i),l}}\right)\left(\mathbf{I}-\left(\mathbf{P}_{\hat{\mathbf{U}}_{g}}+\mathbf{P}_{\hat{\mathbf{U}}_{(i),l}}\right)\right)\right)}$$

Summing up both sides for $i$ from 1 to $N$, and considering equation 11, we have,

$$0\geq\sigma^{2}_{\min}\sum_{i=1}^{N}\mathrm{Tr}\left(\left(\mathbf{P}_{\mathbf{U}^{\star}_{g}}+\mathbf{P}_{\mathbf{U}^{\star}_{(i),l}}\right)\left(\mathbf{I}-\left(\mathbf{P}_{\hat{\mathbf{U}}_{g}}+\mathbf{P}_{\hat{\mathbf{U}}_{(i),l}}\right)\right)\right)$$
$$-\sqrt{2}\sum_{i=1}^{N}\left\|\mathbf{F}^{\star}_{(i)}\right\|_{F}\sqrt{\mathrm{Tr}\left(\left(\mathbf{P}_{\mathbf{U}^{\star}_{g}}+\mathbf{P}_{\mathbf{U}^{\star}_{(i),l}}\right)\left(\mathbf{I}-\left(\mathbf{P}_{\hat{\mathbf{U}}_{g}}+\mathbf{P}_{\hat{\mathbf{U}}_{(i),l}}\right)\right)\right)}$$
$$\geq\sigma^{2}_{\min}\sum_{i=1}^{N}\mathrm{Tr}\left(\left(\mathbf{P}_{\mathbf{U}^{\star}_{g}}+\mathbf{P}_{\mathbf{U}^{\star}_{(i),l}}\right)\left(\mathbf{I}-\left(\mathbf{P}_{\hat{\mathbf{U}}_{g}}+\mathbf{P}_{\hat{\mathbf{U}}_{(i),l}}\right)\right)\right)$$
$$-\sqrt{2}\sqrt{\sum_{i=1}^{N}\left\|\mathbf{F}^{\star}_{(i)}\right\|^{2}_{F}}\sqrt{\sum_{i=1}^{N}\mathrm{Tr}\left(\left(\mathbf{P}_{\mathbf{U}^{\star}_{g}}+\mathbf{P}_{\mathbf{U}^{\star}_{(i),l}}\right)\left(\mathbf{I}-\left(\mathbf{P}_{\hat{\mathbf{U}}_{g}}+\mathbf{P}_{\hat{\mathbf{U}}_{(i),l}}\right)\right)\right)}$$

where we applied Cauchy-Schwartz inequality in the second inequality.

Therefore, one can deduce,

$$\sum_{i=1}^{N}\mathrm{Tr}\left(\left(\mathbf{P}_{\mathbf{U}^{\star}_{g}}+\mathbf{P}_{\mathbf{U}^{\star}_{(i),l}}\right)\left(\mathbf{I}-\left(\mathbf{P}_{\hat{\mathbf{U}}_{g}}+\mathbf{P}_{\hat{\mathbf{U}}_{(i),l}}\right)\right)\right)\leq\frac{2\sum_{i=1}^{N}\left\|\mathbf{F}^{\star}_{(i)}\right\|^{2}_{F}}{\sigma^{4}_{\min}}$$

As $\{\mathbf{U}^{\star}_{(i),l}\}$ are $\theta$-misaligned, from Lemma 2 in Shi & Kontar (2024), we have

$$\sum_{i=1}^{N}\mathrm{Tr}\left(\left(\mathbf{P}_{\mathbf{U}^{\star}_{g}}+\mathbf{P}_{\mathbf{U}^{\star}_{(i),l}}\right)\left(\mathbf{I}-\left(\mathbf{P}_{\hat{\mathbf{U}}_{g}}+\mathbf{P}_{\hat{\mathbf{U}}_{(i),l}}\right)\right)\right)$$

$$\geq \frac{\theta}{2} \left( N \mathrm{Tr} \left( \mathbf{P}_{\mathbf{U}^\star{}_g} - \mathbf{P}_{\mathbf{U}^\star{}_g} \mathbf{P}_{\hat{\mathbf{U}}_g} \right) + \sum_{i=1}^{N} \mathrm{Tr} \left( \mathbf{P}_{\mathbf{U}^\star{}_{(i),l}} - \mathbf{P}_{\mathbf{U}^\star{}_{(i),l}} \mathbf{P}_{\hat{\mathbf{U}}_{(i),l}} \right) \right)$$

We can thus conclude,

$$N \left\| \mathbf{P}_{\mathbf{U}^\star{}_g} - \mathbf{P}_{\hat{\mathbf{U}}_g} \right\|_F^2 + \sum_{i=1}^{N} \left\| \mathbf{P}_{\mathbf{U}^\star{}_{(i),l}} - \mathbf{P}_{\hat{\mathbf{U}}_{(i),l}} \right\|_F^2 \leq \frac{2 \sum_{i=1}^{N} \left\| \mathbf{F}^\star{}_{(i)} \right\|_F^2}{\theta \sigma_{\min}^4} \tag{13}$$

Finally, we prove an upper bound on the reconstruction error of the local and global signals. Notice that

$$\left\| \hat{\mathbf{U}}_g \hat{\mathbf{V}}_{(i),g}^T - \mathbf{U}^\star{}_g \mathbf{V}^\star{}_{(i),g}^T \right\|_F = \left\| \mathbf{P}_{\hat{\mathbf{U}}_g} \mathbf{M}_{(i)} - \mathbf{U}^\star{}_g \mathbf{V}^\star{}_{(i),g}^T \right\|_F$$

$$= \left\| \mathbf{P}_{\hat{\mathbf{U}}_g} \left( \mathbf{U}^\star{}_g \mathbf{V}^\star{}_{(i),g}^T + \mathbf{U}^\star{}_{(i),l} \mathbf{V}^\star{}_{(i),i}^T \right) + \mathbf{P}_{\hat{\mathbf{U}}_g} \mathbf{E}^\star{}_{(i)} - \mathbf{U}^\star{}_g \mathbf{V}^\star{}_{(i),g}^T \right\|_F$$

$$\leq \left\| \mathbf{P}_{\hat{\mathbf{U}}_g} \left( \mathbf{U}^\star{}_g \mathbf{V}^\star{}_{(i),g}^T + \mathbf{U}^\star{}_{(i),l} \mathbf{V}^\star{}_{(i),i}^T \right) - \mathbf{U}^\star{}_g \mathbf{V}^\star{}_{(i),g}^T \right\|_F + \left\| \mathbf{P}_{\hat{\mathbf{U}}_g} \mathbf{E}^\star{}_{(i)} \right\|_F$$

$$\leq \left\| \mathbf{P}_{\hat{\mathbf{U}}_g} - \mathbf{P}_{\mathbf{U}^\star{}_g} \right\|_F \sigma_{\max} + \left\| \mathbf{E}^\star{}_{(i)} \right\|_F \tag{14}$$

We have,

$$\left\| \hat{\mathbf{U}}_g \hat{\mathbf{V}}_{(i),g}^T - \mathbf{U}^\star{}_g \mathbf{V}^\star{}_{(i),g}^T \right\|_F^2 = \left\| \mathbf{P}_{\hat{\mathbf{U}}_g} \mathbf{M}_{(i)} - \mathbf{U}^\star{}_g \mathbf{V}^\star{}_{(i),g}^T \right\|_F^2$$

$$\leq 2 \left\| \mathbf{P}_{\hat{\mathbf{U}}_g} - \mathbf{P}_{\mathbf{U}^\star{}_g} \right\|_F^2 \sigma_{\max}^2 + 2 \left\| \mathbf{E}^\star{}_{(i)} \right\|_F^2$$

Similarly we can prove

$$\left\| \hat{\mathbf{U}}_{(i),l} \hat{\mathbf{V}}_{(i),l}^T - \mathbf{U}^\star{}_{(i),l} \mathbf{V}^\star{}_{(i),l}^T \right\|_F^2 = \left\| \mathbf{P}_{\hat{\mathbf{U}}_{(i),l}} \mathbf{M}_{(i)} - \mathbf{U}^\star{}_{(i),l} \mathbf{V}^\star{}_{(i),l}^T \right\|_F^2$$

$$\leq 2 \left\| \mathbf{P}_{\hat{\mathbf{U}}_{(i),l}} - \mathbf{P}_{\mathbf{U}^\star{}_{(i),l}} \right\|_F^2 \sigma_{\max}^2 + 2 \left\| \mathbf{E}^\star{}_{(i)} \right\|_F^2$$

Combining the two inequalities with equation 13 and considering the fact that $\left\| \mathbf{F}^\star{}_{(i)} \right\|_F \leq 2\sigma_{\max} \left\| \mathbf{E}^\star{}_{(i)} \right\|_F + \left\| \mathbf{E}^\star{}_{(i)} \right\|_F^2$, we can prove (7) in the main paper.

## C  Proof of Theorem 2

In this section we will present the proof of Theorem 2 in the main paper. The proof of Theorem 2 consists of 2 stages. In the first stage, we show Algorithm 1 converges into stationary points. In the second stage, we analyze the local geometry of objective $\tilde{f}$ and show that Algorithm 1 converges into the optimal solutions linearly. The major technical difficulties lie in the second stage, where the constraints $\mathbf{U}_g^T \mathbf{U}_{(i),l} = 0$ make the local geometry analysis complicated.

We will start with a few lemmas and prove Theorem 2 at the end. More specifically, Lemma C.1 connects the assumption on $\mathbf{E}_{(i)}$ to a few properties on the optimal solution $\left( \hat{\mathbf{U}}_g, \{\hat{\mathbf{V}}_{(i),g}, \hat{\mathbf{U}}_{(i),l}, \hat{\mathbf{V}}_{(i),l}\} \right)$. Lemma C.2 estimates the Lipschitz constant of objective $\tilde{f}$. Lemma C.3 constructs the so-called sufficient descent inequality. Lemmas C.5, C.6 C.7, and C.8 are related to the geometry analysis. Finally, Lemma C.10 shows the PL inequality of the objective.

**Lemma C.1** *If* $\mathbf{M}_{(i)} = \mathbf{U}^\star{}_g \mathbf{V}^\star{}_{(i),g}^T + \mathbf{U}^\star{}_{(i),l} \mathbf{V}^\star{}_{(i),l}^T + \mathbf{E}_{(i)}$, *where*

- $\mathbf{E}_{(i)}$'s norm are upper bounded

$$\left\| \mathbf{E}_{(i)} \right\|_F \leq \min \left\{ \sigma_{\min} \theta^{1.5} \frac{1}{6\sqrt{2}} \frac{\sigma_{\min}}{\sigma_{\max}}, \sigma_{\min} \frac{2 - \sqrt{3}}{2} \left( 2 + \frac{6}{\sqrt{\theta}} \left( \frac{\sigma_{\max}}{\sigma_{\min}} \right)^2 \right)^{-1}, \sigma_{\min} \theta \frac{1}{64\sqrt{N}} \frac{\sigma_{\min}}{\sigma_{\max}} \right\}$$

- $\mathbf{U}^\star_{(i),l}$'s are $\theta$-misaligned

- the singular values of $\mathbf{U}^\star_g \mathbf{V}^{\star T}_{(i),g} + \mathbf{U}^\star_{(i),l} \mathbf{V}^{\star T}_{(i),l}$ are upper bounded by $\sigma_{\max}$ and lower bounded by $\sigma_{\min}$

, we can introduce three new constants $\hat{\sigma}_{\max} = 2\sigma_{\max}$, $\hat{\theta} = \frac{\theta}{2}$, and $\hat{\sigma}^2_{gap} = \sigma^2_{\min}/2$ such that:

1. The largest singular values of $\mathbf{M}_{(i)}$'s are upper bounded by $\hat{\sigma}_{\max}$.

2. There exist constants $\hat{\theta}$, $\hat{\sigma}^2_{gap} > 0$ such that $\left\| \frac{1}{N} \sum_{i=1}^{N} \mathbf{P}_{\hat{\mathbf{U}}_{(i),l}} \right\| \leq 1 - \hat{\theta}$, $\sigma^2_{min} \left( \hat{\mathbf{U}}_g \hat{\mathbf{V}}^T_{(i),g} + \hat{\mathbf{U}}_{(i),l} \hat{\mathbf{V}}^T_{(i),l} \right) - \left\| \mathbf{R}_{(i)} \right\|^2 \geq \hat{\sigma}^2_{gap}$, and $\left\| \mathbf{R}_{(i)} \right\| \leq \frac{\hat{\theta} \hat{\sigma}^2_{gap}}{8 \hat{\sigma}_{\max}}$ for every $i$.

**Remark** If the condition number $\frac{\sigma_{\max}}{\sigma_{\min}} = O(1)$, the upper bound on the norm of $\mathbf{E}_{(i)}$ can be written as $\left\| \mathbf{E}_{(i)} \right\|_F \leq O\left( \sigma_{\min} \theta^{1.5} \right)$.

**Proof.** We first prove Claim 1. By triangle inequality,

$$\left\| \mathbf{M}_{(i)} \right\| \leq \left\| \mathbf{U}^\star_g \mathbf{V}^{\star T}_{(i),g} + \mathbf{U}^\star_{(i),l} \mathbf{V}^{\star T}_{(i),l} \right\| + \left\| \mathbf{E}_{(i)} \right\| \leq \sigma_{\max} + \sigma_{\max} = 2\sigma_{\max}$$

where we used the condition $\left\| \mathbf{E}_{(i)} \right\| \leq \left\| \mathbf{E}_{(i)} \right\|_F \leq \sigma_{\min} \theta^{1.5} \frac{1}{6\sqrt{2}} \frac{\sigma_{\min}}{\sigma_{\max}} \leq \sigma_{\max}$.

Then we prove the first inequality of Claim 2. Notice that,

$$\left\| \frac{1}{N} \sum_{i=1}^{N} \mathbf{P}_{\hat{\mathbf{U}}_{(i),l}} \right\| \leq \left\| \frac{1}{N} \sum_{i=1}^{N} \mathbf{P}_{\mathbf{U}^\star_{(i),l}} \right\| + \frac{1}{N} \sum_{i=1}^{N} \left\| \mathbf{P}_{\hat{\mathbf{U}}_{(i),l}} - \mathbf{P}_{\mathbf{U}^\star_{(i),l}} \right\|$$

$$\leq 1 - \theta + \frac{1}{N} \sum_{i=1}^{N} \left\| \mathbf{P}_{\hat{\mathbf{U}}_{(i),l}} - \mathbf{P}_{\mathbf{U}^\star_{(i),l}} \right\|_F$$

From equation 13, we know that $\sum_{i=1}^{N} \left\| \mathbf{P}_{\mathbf{U}^\star_{(i),l}} - \mathbf{P}_{\hat{\mathbf{U}}_{(i),l}} \right\|_F^2 \leq \frac{2 \sum_{i=1}^{N} \left\| \mathbf{F}^\star_{(i)} \right\|_F^2}{\theta \sigma^4_{\min}}$. Therefore,

$$\frac{1}{N} \sum_{i=1}^{N} \left\| \mathbf{P}_{\hat{\mathbf{U}}_{(i),l}} - \mathbf{P}_{\mathbf{U}^\star_{(i),l}} \right\|_F \leq \frac{1}{\sqrt{N}} \sqrt{\sum_{i=1}^{N} \left\| \mathbf{P}_{\hat{\mathbf{U}}_{(i),l}} - \mathbf{P}_{\mathbf{U}^\star_{(i),l}} \right\|_F^2}$$

$$\leq \sqrt{\frac{2 \sum_{i=1}^{N} \left\| \mathbf{F}^\star_{(i)} \right\|_F^2}{\theta \sigma^4_{\min} N}} \leq \theta/2$$

where the last inequality comes from the fact that $\left\| \mathbf{E}_{(i)} \right\|_F \leq \sigma_{\min} \theta^{1.5} \frac{1}{6\sqrt{2}} \frac{\sigma_{\min}}{\sigma_{\max}}$ thus $\left\| \mathbf{F}_{(i)} \right\|_F \leq 2\sigma_{\max} \left\| \mathbf{E}_{(i)} \right\|_F + \left\| \mathbf{E}_{(i)} \right\|_F^2 \leq 3\sigma_{\max} \left\| \mathbf{E}_{(i)} \right\|_F \leq \sigma^2_{\min} \theta^{1.5} \frac{1}{2\sqrt{2}}$.

Next we will show a lower bound on $\sigma_{min} \left( \hat{\mathbf{U}}_g \hat{\mathbf{V}}^T_{(i),g} + \hat{\mathbf{U}}_{(i),l} \hat{\mathbf{V}}^T_{(i),l} \right)$. A fact is that

$$\sigma_{min} \left( \hat{\mathbf{U}}_g \hat{\mathbf{V}}^T_{(i),g} + \hat{\mathbf{U}}_{(i),l} \hat{\mathbf{V}}^T_{(i),l} \right)$$
$$\geq \sigma_{min} \left( \mathbf{U}^\star_g \mathbf{V}^{\star T}_{(i),g} + \mathbf{U}^\star_{(i),l} \mathbf{V}^{\star T}_{(i),l} \right) - \left\| \hat{\mathbf{U}}_g \hat{\mathbf{V}}^T_{(i),g} - \mathbf{U}^\star_g \mathbf{V}^{\star T}_{(i),g} \right\| - \left\| \hat{\mathbf{U}}_{(i),l} \hat{\mathbf{V}}^T_{(i),l} - \mathbf{U}^\star_{(i),l} \mathbf{V}^{\star T}_{(i),l} \right\|$$

We know from equation 14 that

$$\left\| \hat{\mathbf{U}}_g \hat{\mathbf{V}}^T_{(i),g} - \mathbf{U}^\star_g \mathbf{V}^{\star T}_{(i),g} \right\|_F + \left\| \hat{\mathbf{U}}_{(i),l} \hat{\mathbf{V}}^T_{(i),l} - \mathbf{U}^\star_{(i),l} \mathbf{V}^{\star T}_{(i),l} \right\|_F$$
$$\leq \left\| \mathbf{P}_{\hat{\mathbf{U}}_g} - \mathbf{P}_{\mathbf{U}^\star_g} \right\|_F \sigma_{\max} + \left\| \mathbf{P}_{\hat{\mathbf{U}}_{(i),l}} - \mathbf{P}_{\mathbf{U}^\star_{(i),l}} \right\|_F \sigma_{\max} + 2 \left\| \mathbf{E}^\star_{(i)} \right\|_F$$

Therefore, by equation 13, the above is bounded by

$$\sigma_{\max}\sqrt{2}\sqrt{\left\|\mathbf{P}_{\hat{\mathbf{U}}_g} - \mathbf{P}_{\mathbf{U}^{\star}{}_g}\right\|_F^2 + \left\|\mathbf{P}_{\hat{\mathbf{U}}_{(i),l}} - \mathbf{P}_{\mathbf{U}^{\star}{}_{(i),l}}\right\|_F^2} + 2\left\|\mathbf{E}^{\star}{}_{(i)}\right\|_F$$

$$\leq \sigma_{\max}\sqrt{2}\sqrt{\frac{2\sum_{i=1}^{N}\left\|\mathbf{F}^{\star}{}_{(i)}\right\|_F^2}{\theta\sigma_{\min}^4}} + 2\left\|\mathbf{E}^{\star}{}_{(i)}\right\|_F$$

$$\leq \left\|\mathbf{E}^{\star}{}_{(i)}\right\|_F\left(2 + \frac{6\sigma_{\max}^2}{\sqrt{\theta}\sigma_{\min}^2}\right) \leq \sigma_{\min}\frac{2-\sqrt{3}}{2}$$

where we applied the condition $\left\|\mathbf{E}^{\star}{}_{(i)}\right\|_F \leq \sigma_{\min}\frac{2-\sqrt{3}}{2}\left(2 + \frac{6}{\sqrt{\theta}}\left(\frac{\sigma_{\max}}{\sigma_{\min}}\right)^2\right)^{-1}$ in the last inequality. Thus

$$\sigma_{min}\left(\hat{\mathbf{U}}_g\hat{\mathbf{V}}_{(i),g}^T + \hat{\mathbf{U}}_{(i),l}\hat{\mathbf{V}}_{(i),l}^T\right) \geq \sigma_{\min}\frac{\sqrt{3}}{2}$$

Also, since $\mathbf{R}_{(i)}$'s correspond to the residuals of the optimal solutions, we have,

$$\left\|\mathbf{R}_{(i)}\right\| \leq \left\|\mathbf{R}_{(i)}\right\|_F \leq \sqrt{\sum_{i=1}^{N}\left\|\mathbf{R}_{(i)}\right\|_F^2} \leq \sqrt{\sum_{i=1}^{N}\left\|\mathbf{E}^{\star}{}_{(i)}\right\|_F^2} \leq \sigma_{\min}/2 \tag{15}$$

where we applied the condition $\left\|\mathbf{E}^{\star}{}_{(i)}\right\|_F \leq \sigma_{\min}\theta\frac{1}{64\sqrt{N}}\frac{\sigma_{\min}}{\sigma_{\max}}$ in the last inequality.

As a result, we have $\sigma_{min}^2\left(\hat{\mathbf{U}}_g\hat{\mathbf{V}}_{(i),g}^T + \hat{\mathbf{U}}_{(i),l}\hat{\mathbf{V}}_{(i),l}^T\right) - \left\|\mathbf{R}_{(i)}\right\|^2 \geq \hat{\sigma}_{\text{gap}}^2$.

Finally, we will prove the third condition in Claim 2. We can do so by using the condition $\left\|\mathbf{E}^{\star}{}_{(i)}\right\|_F \leq \sigma_{\min}\theta\frac{1}{64\sqrt{N}}\frac{\sigma_{\min}}{\sigma_{\max}}$ in equation 15, and considering the fact that $\hat{\sigma}_{\text{gap}}^2 = \sigma_{\min}^2/2$, $\hat{\theta} = \theta/2$, and $\hat{\sigma}_{\max} = 2\sigma_{\max}$.

This completes our proof. □

In the following, we will always assume Claims 1 and 2 are true and prove our results under the assumptions of Lemma C.1.

The two claims are intuitively understandable. Claim 1 upper bounds the singular values of $\mathbf{M}_{(i)}$'s. This is the standard requirement in matrix factorization. Claim 2 implies $\hat{\mathbf{U}}_{(i),l}$'s should be $\hat{\theta}$-misaligned for some nonzero $\hat{\theta}$. Also, Claim 2 restricts the norm of $\mathbf{R}_{(i)}$'s to be smaller than a factor of the smallest nonzero singular value of $\hat{\mathbf{U}}_g\hat{\mathbf{V}}_{(i),g}^T + \hat{\mathbf{U}}_{(i),l}\hat{\mathbf{V}}_{(i),l}^T$. This requirement is satisfied when the norm of the noise in input $\mathbf{M}_{(i)}$ is not too large, i.e., $\mathbf{M}_{(i)}$ is not far from the linear combination of two low-rank matrices.

As introduced in the main paper, we use $\mathcal{B}\left(\zeta_1, \zeta_2\right)$ to denote the set of solutions with bounded norms:

$$\mathcal{B}\left(\zeta_1, \zeta_2\right) = \{\mathbf{U}_g, \{\mathbf{U}_{(i),l}\}, \{\mathbf{V}_{(i),g}\}, \{\mathbf{V}_{(i),l}\}\big|\left\|\mathbf{U}_g\right\|, \left\|\mathbf{U}_{(i),l}\right\| \leq \zeta_1, \left\|\mathbf{V}_{(i),g}\right\|, \left\|\mathbf{V}_{(i),l}\right\| \leq \zeta_2\} \tag{16}$$

The following lemma gives an upper bound on the Lipschitz constant when all iterates are bounded.

**Lemma C.2 (Lipschitz continuity)** *In region $\mathcal{B}\left(B_1, B_2\right)$ as defined in equation 16, $g_i$ and $\tilde{f}_i$ are Lipschitz continuous:*

$$\left\|\nabla g_i(\mathbf{U}_g', \mathbf{U}_{(i),l}') - \nabla g_i(\mathbf{U}_g, \mathbf{U}_{(i),l})\right\|_F$$
$$\leq L_g\sqrt{\left\|\mathbf{U}_g' - \mathbf{U}_g\right\|_F^2 + \left\|\mathbf{U}_{(i),l}' - \mathbf{U}_{(i),l}\right\|_F^2} \tag{17}$$

*and*

$$\left\|\nabla\tilde{f}_i(\mathbf{U}_g', \mathbf{V}_{(i),g}', \mathbf{U}_{(i),l}', \mathbf{V}_{(i),l}') - \nabla\tilde{f}_i(\mathbf{U}_g, \mathbf{V}_{(i),g}, \mathbf{U}_{(i),l}, \mathbf{V}_{(i),l})\right\|_F$$
$$\leq L\sqrt{\left\|\mathbf{U}_g' - \mathbf{U}_g\right\|_F^2 + \left\|\mathbf{V}_{(i),g}' - \mathbf{V}_{(i),g}\right\|_F^2 + \left\|\mathbf{U}_{(i),l}' - \mathbf{U}_{(i),l}\right\|_F^2 + \left\|\mathbf{V}_{(i),l}' - \mathbf{V}_{(i),l}\right\|_F^2} \tag{18}$$

for $\{\mathbf{U}'_g, \{\mathbf{V}'_{(i),g}\}, \{\mathbf{U}'_{(i),l}\}, \{\mathbf{V}'_{(i),l}\}\}, \{\mathbf{U}_g, \{\mathbf{V}_{(i),g}\}, \{\mathbf{U}_{(i),l}\}, \{\mathbf{V}_{(i),l}\}\} \in \mathcal{B}(B_1, B_2)$, *where* $L_g$ *and* $L$ *are constants dependent on* $B_1$, $B_2$, $\hat{\sigma}_{\max}$, *and* $\beta$,

$$L_g = 2\beta(3B_1^2 + 1)$$

*and,*

$$L = \sqrt{(\hat{\sigma}_{\max} + 3B_1 B_2)^2 + B_1^2 B_2^2 + 2B_1^4 + B_2^4 + (B_2^2 + 2\beta(3B_1^2 + 1))^2}$$

**Proof.** We will firstly show the Lipschitz continuity of $g_i$. We know that

$$\nabla_{\mathbf{U}_g} g_i(\mathbf{U}'_g, \mathbf{U}'_{(i),l}) - \nabla_{\mathbf{U}_g} g_i(\mathbf{U}_g, \mathbf{U}_{(i),l})$$
$$= 2\beta \mathbf{U}'_g \left((\mathbf{U}'_g)^T \mathbf{U}'_g - \mathbf{I}\right) - 2\beta \mathbf{U}_g \left(\mathbf{U}_g^T \mathbf{U}_g - \mathbf{I}\right)$$

Therefore by some applications of triangle inequalities, we have,

$$\left\|\nabla_{\mathbf{U}_g} g_i(\mathbf{U}'_g, \mathbf{U}'_{(i),l}) - \nabla_{\mathbf{U}_g} g_i(\mathbf{U}_g, \mathbf{U}_{(i),l})\right\|_F \le \left\|\mathbf{U}'_g - \mathbf{U}_g\right\| 2\beta(3B_1^2 + 1)$$

This proves inequality equation 17. The Lipschitz continuity of $\tilde{f}_i$ can be proved similarly. We will calculate the gradient of $\tilde{f}_i$ over each variable, and bound the norm of the difference of the gradients.

$$\nabla_{\mathbf{U}_g} \tilde{f}_i(\mathbf{U}'_g, \mathbf{V}'_{(i),g}, \mathbf{U}'_{(i),l}, \mathbf{V}'_{(i),l}) - \nabla_{\mathbf{U}_g} \tilde{f}_i(\mathbf{U}_g, \mathbf{V}_{(i),g}, \mathbf{U}_{(i),l}, \mathbf{V}_{(i),l})$$
$$= \left(\mathbf{U}'_g (\mathbf{V}'_{(i),g})^T + \mathbf{U}'_{(i),l}(\mathbf{V}'_{(i),l})^T - \mathbf{M}_{(i)}\right) \mathbf{V}'_{(i),g} + 2\beta \mathbf{U}'_g((\mathbf{U}'_g)^T \mathbf{U}'_g - \mathbf{I})$$
$$- \left(\mathbf{U}_g (\mathbf{V}_{(i),g})^T + \mathbf{U}_{(i),l}(\mathbf{V}_{(i),l})^T - \mathbf{M}_{(i)}\right) \mathbf{V}_{(i),g} - 2\beta \mathbf{U}_g(\mathbf{U}_g^T \mathbf{U}_g - \mathbf{I})$$

Also by triangle inequalities, when $(\mathbf{U}'_g, \mathbf{V}'_{(i),g}, \mathbf{U}'_{(i),l}, \mathbf{V}'_{(i),l})$ and $(\mathbf{U}_g, \mathbf{V}_{(i),g}, \mathbf{U}_{(i),l}, \mathbf{V}_{(i),l})$ are in $\mathcal{B}(B_1, B_2)$, we have:

$$\left\|\nabla_{\mathbf{U}_g} \tilde{f}_i(\mathbf{U}'_g, \mathbf{V}'_{(i),g}, \mathbf{U}'_{(i),l}, \mathbf{V}'_{(i),l}) - \nabla_{\mathbf{U}_g} \tilde{f}_i(\mathbf{U}_g, \mathbf{V}_{(i),g}, \mathbf{U}_{(i),l}, \mathbf{V}_{(i),l})\right\|_F$$
$$\le (B_2^2 + 2\beta(3B_1^2 + 1)) \left\|\mathbf{U}'_g - \mathbf{U}_g\right\|_F + (\hat{\sigma}_{\max} + 3B_1 B_2) \left\|\mathbf{V}'_{(i),g} - \mathbf{V}_{(i),g}\right\|_F$$
$$+ B_2^2 \left\|\mathbf{U}'_{(i),l} - \mathbf{U}_{(i),l}\right\|_F + B_1 B_2 \left\|\mathbf{V}'_{(i),l} - \mathbf{V}_{(i),l}\right\|_F$$

Then on the derivative over $\mathbf{V}_{(i),g}$,

$$\nabla_{\mathbf{V}_{(i),g}} \tilde{f}_i(\mathbf{U}'_g, \mathbf{V}'_{(i),g}, \mathbf{U}'_{(i),l}, \mathbf{V}'_{(i),l}) - \nabla_{\mathbf{V}_{(i),g}} \tilde{f}_i(\mathbf{U}_g, \mathbf{V}_{(i),g}, \mathbf{U}_{(i),l}, \mathbf{V}_{(i),l})$$
$$= \left(\mathbf{U}'_g (\mathbf{V}'_{(i),g})^T + \mathbf{U}'_{(i),l}(\mathbf{V}'_{(i),l})^T - \mathbf{M}_{(i)}\right)^T \mathbf{U}'_g - \left(\mathbf{U}_g (\mathbf{V}_{(i),g})^T + \mathbf{U}_{(i),l}(\mathbf{V}_{(i),l})^T - \mathbf{M}_{(i)}\right)^T \mathbf{U}_g$$

Thus by similar calculations, we have,

$$\left\|\nabla_{\mathbf{V}_{(i),g}} \tilde{f}_i(\mathbf{U}'_g, \mathbf{V}'_{(i),g}, \mathbf{U}'_{(i),l}, \mathbf{V}'_{(i),l}) - \nabla_{\mathbf{V}_{(i),g}} \tilde{f}_i(\mathbf{U}_g, \mathbf{V}_{(i),g}, \mathbf{U}_{(i),l}, \mathbf{V}_{(i),l})\right\|_F$$
$$\le (\hat{\sigma}_{\max} + 3B_1 B_2) \left\|\mathbf{U}'_g - \mathbf{U}_g\right\|_F + B_1^2 \left\|\mathbf{V}'_{(i),g} - \mathbf{V}_{(i),g}\right\|_F$$
$$+ B_1 B_2 \left\|\mathbf{U}'_{(i),l} - \mathbf{U}_{(i),l}\right\|_F + B_1^2 \left\|\mathbf{V}'_{(i),l} - \mathbf{V}_{(i),l}\right\|_F$$

And,

$$\left\|\nabla_{\mathbf{U}_{(i),l}} \tilde{f}_i(\mathbf{U}'_g, \mathbf{V}'_{(i),g}, \mathbf{U}'_{(i),l}, \mathbf{V}'_{(i),l}) - \nabla_{\mathbf{U}_{(i),l}} \tilde{f}_i(\mathbf{U}_g, \mathbf{V}_{(i),g}, \mathbf{U}_{(i),l}, \mathbf{V}_{(i),l})\right\|_F$$
$$\le B_2^2 \left\|\mathbf{U}'_g - \mathbf{U}_g\right\|_F + B_1 B_2 \left\|\mathbf{V}'_{(i),g} - \mathbf{V}_{(i),g}\right\|_F$$
$$+ (B_2^2 + 2\beta(3B_1^2 + 1)) \left\|\mathbf{U}'_{(i),l} - \mathbf{U}_{(i),l}\right\|_F + (\hat{\sigma}_{\max} + 3B_1 B_2) \left\|\mathbf{V}'_{(i),l} - \mathbf{V}_{(i),l}\right\|_F$$

Also,

$$\left\|\nabla_{\mathbf{V}_{(i),l}}\tilde{f}_i(\mathbf{U}'_g,\mathbf{V}'_{(i),g},\mathbf{U}'_{(i),l},\mathbf{V}'_{(i),l}) - \nabla_{\mathbf{V}_{(i),l}}\tilde{f}_i(\mathbf{U}_g,\mathbf{V}_{(i),g},\mathbf{U}_{(i),l},\mathbf{V}_{(i),l})\right\|_F$$

$$\leq B_1 B_2 \left\|\mathbf{U}'_g - \mathbf{U}_g\right\|_F + B_1^2 \left\|\mathbf{V}'_{(i),g} - \mathbf{V}_{(i),g}\right\|_F$$

$$+ (\hat{\sigma}_{\max} + 3B_1 B_2)\left\|\mathbf{U}'_{(i),l} - \mathbf{U}_{(i),l}\right\|_F + B_1^2 \left\|\mathbf{V}'_{(i),l} - \mathbf{V}_{(i),l}\right\|_F$$

Combining the 4 inequalities, we have:

$$\left\|\nabla\tilde{f}_i(\mathbf{U}'_g,\mathbf{V}'_{(i),g},\mathbf{U}'_{(i),l},\mathbf{V}'_{(i),l}) - \nabla\tilde{f}_i(\mathbf{U}_g,\mathbf{V}_{(i),g},\mathbf{U}_{(i),l},\mathbf{V}_{(i),l})\right\|_F^2$$

$$= \left\|\nabla_{\mathbf{U}_g}\tilde{f}_i(\mathbf{U}'_g,\mathbf{V}'_{(i),g},\mathbf{U}'_{(i),l},\mathbf{V}'_{(i),l}) - \nabla_{\mathbf{U}_g}\tilde{f}_i(\mathbf{U}_g,\mathbf{V}_{(i),g},\mathbf{U}_{(i),l},\mathbf{V}_{(i),l})\right\|_F^2$$

$$+ \left\|\nabla_{\mathbf{V}_{(i),g}}\tilde{f}_i(\mathbf{U}'_g,\mathbf{V}'_{(i),g},\mathbf{U}'_{(i),l},\mathbf{V}'_{(i),l}) - \nabla_{\mathbf{V}_{(i),g}}\tilde{f}_i(\mathbf{U}_g,\mathbf{V}_{(i),g},\mathbf{U}_{(i),l},\mathbf{V}_{(i),l})\right\|_F^2$$

$$+ \left\|\nabla_{\mathbf{U}_{(i),l}}\tilde{f}_i(\mathbf{U}'_g,\mathbf{V}'_{(i),g},\mathbf{U}'_{(i),l},\mathbf{V}'_{(i),l}) - \nabla_{\mathbf{U}_{(i),l}}\tilde{f}_i(\mathbf{U}_g,\mathbf{V}_{(i),g},\mathbf{U}_{(i),l},\mathbf{V}_{(i),l})\right\|_F^2$$

$$+ \left\|\nabla_{\mathbf{V}_{(i),l}}\tilde{f}_i(\mathbf{U}'_g,\mathbf{V}'_{(i),g},\mathbf{U}'_{(i),l},\mathbf{V}'_{(i),l}) - \nabla_{\mathbf{V}_{(i),l}}\tilde{f}_i(\mathbf{U}_g,\mathbf{V}_{(i),g},\mathbf{U}_{(i),l},\mathbf{V}_{(i),l})\right\|_F^2$$

$$\leq L^2\left(\left\|\mathbf{U}'_g - \mathbf{U}_g\right\|_F^2 + \left\|\mathbf{V}'_{(i),g} - \mathbf{V}_{(i),g}\right\|_F^2 + \left\|\mathbf{U}'_{(i),l} - \mathbf{U}_{(i),l}\right\|_F^2 + \left\|\mathbf{V}'_{(i),l} - \mathbf{V}_{(i),l}\right\|_F^2\right)$$

where $L$ is a constant defined as,

$$L = \sqrt{(\hat{\sigma}_{\max} + 3B_1 B_2)^2 + B_1^2 B_2^2 + 2B_1^4 + B_2^4 + (B_2^2 + 2\beta(3B_1^2 + 1))^2}$$

$\square$

The Lipschitz continuity allows us to prove the following sufficient descent lemma.

**Lemma C.3 (Sufficient descent inequality)** *For* $\{\mathbf{U}_{g,\tau},\{\mathbf{V}_{(i),g,\tau}\},\{\mathbf{U}_{(i),l,\tau}\},\{\mathbf{V}_{(i),l,\tau}\}\} \in \mathcal{B}(B_1,B_2)$, *if* $\left\|\mathbf{U}_{g,\tau}^T\mathbf{U}_{g,\tau} - \mathbf{I}\right\| \leq \frac{3}{4}$ *and* $\left\|\mathbf{U}_{(i),l,\tau}^T\mathbf{U}_{(i),l,\tau} - \mathbf{I}\right\|_F \leq \frac{3}{4}$ *for every* $i$, *and the stepsize* $\eta$ *small enough,*

$$\eta \leq \min\left\{\frac{1}{10}\left(2B_1 B_2(2B_1 B_2 + \hat{\sigma}_{\max}) + 2\beta B_1^2(B_1^2 + 1) + \left(B_2(2B_1 B_2 + \hat{\sigma}_{\max}) + 2\beta B_1(B_1^2 + 1)\right)^2\right),\right.$$

$$\left.\frac{1}{2\left(\frac{L}{2} + 272\beta B_1^2 + 400L_g B_1^2\right)}, 1\right\}$$

*where* $L$ *and* $L_g$ *are defined in Lemma C.2, then the iterates of algorithm 1 satisfy the following inequality,*

$$\tilde{f}(\mathbf{U}_{g,\tau+1},\{\mathbf{V}_{(i),g,\tau+1},\mathbf{U}_{(i),l,\tau+1},\mathbf{V}_{(i),l,\tau+1}\}) - \tilde{f}(\mathbf{U}_{g,\tau},\{\mathbf{V}_{(i),g,\tau},\mathbf{U}_{(i),l,\tau},\mathbf{V}_{(i),l,\tau}\})$$

$$\leq -\frac{\eta}{2}\left(\left\|\frac{1}{\sqrt{N}}\nabla_{\mathbf{U}_g}\tilde{f}\right\|_F^2 + \sum_{i=1}^N \left\|\nabla_{\mathbf{V}_{(i),g}}\tilde{f}_i\right\|_F^2 + \left\|\nabla_{\mathbf{U}_{(i),l}}\tilde{f}_i\right\|_F^2 + \left\|\nabla_{\mathbf{V}_{(i),l}}\tilde{f}_i\right\|_F^2\right)$$

**Proof.** We first look at the correction step. Notice that the function value $f_i$ before and after correction is the same:

$$f_i(\mathbf{U}_{g,\tau+1},\mathbf{V}_{(i),g,\tau+1},\mathbf{U}_{(i),l,\tau+1},\mathbf{V}_{(i),l,\tau+1})$$

$$= \left\|\mathbf{M}_{(i)} - \mathbf{U}_{g,\tau+1}\mathbf{V}_{(i),g,\tau+1}^T - \mathbf{U}_{(i),l,\tau+1}\mathbf{V}_{(i),l,\tau+1}^T\right\|_F^2$$

$$= \|\mathbf{M}_{(i)} - \mathbf{U}_{g,\tau+1}\left(\mathbf{V}_{(i),g,\tau+\frac{1}{2}} + \mathbf{V}_{(i),l,\tau+1}\mathbf{U}_{(i),l,\tau+\frac{1}{2}}^T\mathbf{U}_{g,\tau+1}\left(\mathbf{U}_{g,\tau+1}^T\mathbf{U}_{g,\tau+1}\right)^{-1}\right)^T$$

$$- \left(\mathbf{U}_{(i),l,\tau+\frac{1}{2}} - \mathbf{U}_{g,\tau+1}\left(\mathbf{U}_{g,\tau+1}^T\mathbf{U}_{g,\tau+1}\right)^{-1}\mathbf{U}_{g,\tau+1}^T\mathbf{U}_{(i),l,\tau+\frac{1}{2}}\right)\mathbf{V}_{(i),l,\tau+1}^T\|_F^2$$

$$= \left\|\mathbf{M}_{(i),\tau+1} - \mathbf{U}_{g,\tau+1}\mathbf{V}_{(i),g,\tau+\frac{1}{2}}^T - \mathbf{U}_{(i),l,\tau+\frac{1}{2}}\mathbf{V}_{(i),l,\tau+1}^T\right\|_F^2$$

$$= f_i(\mathbf{U}_{g,\tau+1},\mathbf{V}_{(i),g,\tau+\frac{1}{2}},\mathbf{U}_{(i),l,\tau+\frac{1}{2}},\mathbf{V}_{(i),l,\tau+1})$$

For $g_i$, we have:

$$g_i(\mathbf{U}_{g,\tau+1}, \mathbf{U}_{(i),l,\tau+1}) - g_i(\mathbf{U}_{g,\tau+1}, \mathbf{U}_{(i),l,\tau+\frac{1}{2}})$$

$$\leq \left\langle \nabla_{\mathbf{U}_{(i),l}} g_i(\mathbf{U}_{g,\tau+1}, \mathbf{U}_{(i),l,\tau+\frac{1}{2}}), \mathbf{U}_{g,\tau+1} \left(\mathbf{U}_{g,\tau+1}^T \mathbf{U}_{g,\tau+1}\right)^{-1} \mathbf{U}_{g,\tau+1}^T \mathbf{U}_{(i),l,\tau+\frac{1}{2}} \right\rangle$$

$$+ \frac{L_g}{2} \left\| \mathbf{U}_{g,\tau+1} \left(\mathbf{U}_{g,\tau+1}^T \mathbf{U}_{g,\tau+1}\right)^{-1} \mathbf{U}_{g,\tau+1}^T \mathbf{U}_{(i),l,\tau+\frac{1}{2}} \right\|_F^2$$

where $L_g$ is the Lipschitz continuity constant of $g_i$. For the first term, we have,

$$\left\langle \nabla_{\mathbf{U}_{(i),l}} g_i(\mathbf{U}_{g,\tau+1}, \mathbf{U}_{(i),l,\tau+\frac{1}{2}}), \mathbf{U}_{g,\tau+1} \left(\mathbf{U}_{g,\tau+1}^T \mathbf{U}_{g,\tau+1}\right)^{-1} \mathbf{U}_{g,\tau+1}^T \mathbf{U}_{(i),l,\tau+\frac{1}{2}} \right\rangle$$

$$= \mathrm{Tr}\left( 2\beta \mathbf{U}_{(i),l,\tau+\frac{1}{2}}^T (\mathbf{U}_{(i),l,\tau+\frac{1}{2}}^T \mathbf{U}_{(i),l,\tau+\frac{1}{2}} - \mathbf{I}) \mathbf{U}_{(i),l,\tau+\frac{1}{2}}^T \mathbf{U}_{g,\tau+1} \left(\mathbf{U}_{g,\tau+1}^T \mathbf{U}_{g,\tau+1}\right)^{-1} \mathbf{U}_{g,\tau+1}^T \mathbf{U}_{(i),l,\tau+\frac{1}{2}} \right)$$

Since $\left\| \mathbf{U}_{(i),l,\tau}^T \mathbf{U}_{(i),l,\tau} - \mathbf{I} \right\|_F \leq \frac{3}{4}$, we know,

$$\left\| \mathbf{U}_{(i),l,\tau+\frac{1}{2}}^T \mathbf{U}_{(i),l,\tau+\frac{1}{2}} - \mathbf{I} \right\|_F$$

$$= \left\| \mathbf{U}_{(i),l,\tau+\frac{1}{2}}^T \mathbf{U}_{(i),l,\tau+\frac{1}{2}} - \mathbf{I} - \eta \nabla_{\mathbf{U}_{(i),l}} \tilde{f}_i^T \mathbf{U}_{(i),l} - \eta \mathbf{U}_{(i),l}^T \nabla_{\mathbf{U}_{(i),l}} \tilde{f}_i + \eta^2 \nabla_{\mathbf{U}_{(i),l}} \tilde{f}_i^T \nabla_{\mathbf{U}_{(i),l}} \tilde{f}_i \right\|_F$$

$$\leq \left\| \mathbf{U}_{(i),l,\tau}^T \mathbf{U}_{(i),l,\tau} - \mathbf{I} \right\| + \eta \left( \left\| \nabla_{\mathbf{U}_{(i),l}} \tilde{f}_i \right\| B_1 + \left\| \nabla_{\mathbf{U}_{(i),l}} \tilde{f}_i \right\| B_1 + \eta \left\| \nabla_{\mathbf{U}_{(i),l}} \tilde{f}_i \right\|^2 \right)$$

$$\leq \frac{17}{20}$$

where in the last inequality, we applied the condition that $\left\| \mathbf{U}_{(i),l,\tau}^T \mathbf{U}_{(i),l,\tau} - \mathbf{I} \right\| \leq \frac{3}{4}$ and $\eta$ is not too large,

$$\eta \leq \frac{1}{10} \left( 2B_1 B_2(2B_1 B_2 + \hat{\sigma}_{\max}) + 2\beta B_1^2(B_1^2 + 1) + \left(B_2(2B_1 B_2 + \hat{\sigma}_{\max}) + 2\beta B_1(B_1^2 + 1)\right)^2 \right)$$

As,

$$\left\| \mathbf{U}_{g,\tau+1}^T \mathbf{U}_{g,\tau+1} - \mathbf{U}_{g,\tau}^T \mathbf{U}_{g,\tau} \right\|$$

$$= \eta \left\| \mathbf{U}_{g,\tau}^T \nabla_{\mathbf{U}_g} \frac{\tilde{f}}{N} + (\nabla_{\mathbf{U}_g} \frac{\tilde{f}}{N})^T \mathbf{U}_{g,\tau} + (\nabla_{\mathbf{U}_g} \frac{\tilde{f}}{N})^T \nabla_{\mathbf{U}_g} \frac{\tilde{f}}{N} \right\|$$

$$\leq \eta \left( 2B_1 B_2(2B_1 B_2 + \hat{\sigma}_{\max}) + 2\beta B_1^2(B_1^2 + 1) + \eta \left(B_2(2B_1 B_2 + \hat{\sigma}_{\max}) + 2\beta B_1(B_1^2 + 1)\right)^2 \right)$$

Since $\eta$ is sufficiently small

$$\eta \leq \frac{3}{20} \left( 2B_1 B_2(2B_1 B_2 + \hat{\sigma}_{\max}) + 2\beta B_1^2(B_1^2 + 1) + \left(B_2(2B_1 B_2 + \hat{\sigma}_{\max}) + 2\beta B_1(B_1^2 + 1)\right)^2 \right)$$

we know that $\left\| \mathbf{U}_{g,\tau+1}^T \mathbf{U}_{g,\tau+1} - \mathbf{U}_{g,\tau}^T \mathbf{U}_{g,\tau} \right\| \leq \frac{3}{20}$. Then, by Lemma E.4 we have,

$$\left\| \left(\mathbf{U}_{g,\tau+1}^T \mathbf{U}_{g,\tau+1}\right)^{-1} \right\|$$

$$\leq \left\| \left(\mathbf{U}_{g,\tau}^T \mathbf{U}_{g,\tau}\right)^{-1} \right\| + \frac{\left\| \left(\mathbf{U}_{g,\tau}^T \mathbf{U}_{g,\tau}\right)^{-1} \right\|^2 \left\| \mathbf{U}_{g,\tau+1}^T \mathbf{U}_{g,\tau+1} - \mathbf{U}_{g,\tau}^T \mathbf{U}_{g,\tau} \right\|}{1 - \left\| \left(\mathbf{U}_{g,\tau}^T \mathbf{U}_{g,\tau}\right)^{-1} \right\| \left\| \mathbf{U}_{g,\tau+1}^T \mathbf{U}_{g,\tau+1} - \mathbf{U}_{g,\tau}^T \mathbf{U}_{g,\tau} \right\|}$$

Lemma E.1 shows that when $\left\| \mathbf{U}_{g,\tau}^T \mathbf{U}_{g,\tau} - \mathbf{I} \right\|_F \leq \frac{3}{4}$, we have $\left\| \left(\mathbf{U}_{g,\tau}^T \mathbf{U}_{g,\tau}\right)^{-1} \right\| \leq 1 + 3 = 4$. Combining these inequalities, we have $\left\| \left(\mathbf{U}_{g,\tau+1}^T \mathbf{U}_{g,\tau+1}\right)^{-1} \right\| \leq 10$.

Finally, we can look at

$$\left\|\mathbf{U}_{g,\tau+1}^T\mathbf{U}_{(i),l,\tau+\frac{1}{2}}\right\|_F$$

$$= \left\|\mathbf{U}_{g,\tau}^T\mathbf{U}_{(i),l,\tau} - \eta(\nabla_{\mathbf{U}_g}\frac{\tilde{f}}{N})^T\mathbf{U}_{(i),l,\tau} - \eta\mathbf{U}_{g,\tau}^T\nabla_{\mathbf{U}_{(i),l}}\tilde{f}_i + \eta^2\left(\nabla_{\mathbf{U}_g}\frac{\tilde{f}}{N}\right)^T\nabla_{\mathbf{U}_{(i),l}}\tilde{f}_i\right\|_F$$

$$\leq \left\|\eta(\nabla_{\mathbf{U}_g}\frac{\tilde{f}}{N})^T\mathbf{U}_{(i),l,\tau}\right\|_F + \left\|\eta\mathbf{U}_{g,\tau}^T\nabla_{\mathbf{U}_{(i),l}}\tilde{f}_i\right\|_F + \left\|\eta^2\left(\nabla_{\mathbf{U}_g}\frac{\tilde{f}}{N}\right)^T\nabla_{\mathbf{U}_{(i),l}}\tilde{f}_i\right\|_F$$

$$\leq \eta 2B_1\left\|\nabla_{\mathbf{U}_g}\frac{\tilde{f}}{N}\right\|_F + \eta 2B_1\left\|\nabla_{\mathbf{U}_{(i),l}}\tilde{f}_i\right\|_F$$

where we bound the second order terms by first order terms considering the fact that $\eta\left\|\nabla_{\mathbf{U}_g}\frac{\tilde{f}}{N}\right\|_F, \left\|\eta\nabla_{\mathbf{U}_{(i),l}}\tilde{f}_i\right\|_F \leq 1$.

Combining the three, we have,

$$\left\langle\nabla_{\mathbf{U}_{(i),l}}g_i(\mathbf{U}_{g,\tau+1}, \mathbf{U}_{(i),l,\tau+\frac{1}{2}}), \mathbf{U}_{g,\tau+1}\left(\mathbf{U}_{g,\tau+1}^T\mathbf{U}_{g,\tau+1}\right)^{-1}\mathbf{U}_{g,\tau+1}^T\mathbf{U}_{(i),l,\tau+\frac{1}{2}}\right\rangle$$

$$= \mathrm{Tr}\left(2\beta\mathbf{U}_{(i),l,\tau+\frac{1}{2}}^T(\mathbf{U}_{(i),l,\tau+\frac{1}{2}}^T\mathbf{U}_{(i),l,\tau+\frac{1}{2}} - \mathbf{I})\mathbf{U}_{(i),l,\tau+\frac{1}{2}}^T\mathbf{U}_{g,\tau+1}\left(\mathbf{U}_{g,\tau+1}^T\mathbf{U}_{g,\tau+1}\right)^{-1}\mathbf{U}_{g,\tau+1}^T\mathbf{U}_{(i),l,\tau+\frac{1}{2}}\right)$$

$$\leq 2\beta\left\|\mathbf{U}_{(i),l,\tau+\frac{1}{2}}^T(\mathbf{U}_{(i),l,\tau+\frac{1}{2}}^T\mathbf{U}_{(i),l,\tau+\frac{1}{2}} - \mathbf{I})\mathbf{U}_{(i),l,\tau+\frac{1}{2}}^T\mathbf{U}_{g,\tau+1}\left(\mathbf{U}_{g,\tau+1}^T\mathbf{U}_{g,\tau+1}\right)^{-1}\right\|_F\left\|\mathbf{U}_{g,\tau+1}^T\mathbf{U}_{(i),l,\tau+\frac{1}{2}}\right\|_F$$

$$\leq 2\beta\left\|\mathbf{U}_{(i),l,\tau+\frac{1}{2}}\right\|\left\|\mathbf{U}_{(i),l,\tau+\frac{1}{2}}^T\mathbf{U}_{(i),l,\tau+\frac{1}{2}} - \mathbf{I}\right\|_F\left\|\left(\mathbf{U}_{g,\tau+1}^T\mathbf{U}_{g,\tau+1}\right)^{-1}\right\|_F\left\|\mathbf{U}_{g,\tau+1}^T\mathbf{U}_{(i),l,\tau+\frac{1}{2}}\right\|_F^2$$

$$\leq \eta^2\left(\left\|\nabla_{\mathbf{U}_g}\frac{\tilde{f}}{N}\right\|_F^2 + \left\|\nabla_{\mathbf{U}_{(i),l}}\tilde{f}_i\right\|_F^2\right)272\beta B_1^3$$

Similarly,

$$\frac{L_g}{2}\left\|\mathbf{U}_{g,\tau+1}\left(\mathbf{U}_{g,\tau+1}^T\mathbf{U}_{g,\tau+1}\right)^{-1}\mathbf{U}_{g,\tau+1}^T\mathbf{U}_{(i),l,\tau+\frac{1}{2}}\right\|_F^2$$

$$\leq \frac{L_g}{2}\left\|\mathbf{U}_{g,\tau+1}\right\|^2\left\|\left(\mathbf{U}_{g,\tau+1}^T\mathbf{U}_{g,\tau+1}\right)^{-1}\right\|^2\left\|\mathbf{U}_{g,\tau+1}^T\mathbf{U}_{(i),l,\tau+\frac{1}{2}}\right\|_F^2$$

$$\leq \eta^2\left(\left\|\nabla_{\mathbf{U}_g}\frac{\tilde{f}}{N}\right\|_F^2 + \left\|\nabla_{\mathbf{U}_{(i),l}}\tilde{f}_i\right\|_F^2\right)400L_gB_1^6$$

Therefore,

$$g_i(\mathbf{U}_{g,\tau+1}, \mathbf{U}_{(i),l,\tau+1}) - g_i(\mathbf{U}_{g,\tau+1}, \mathbf{U}_{(i),l,\tau+\frac{1}{2}}) \leq \eta^2\left(\left\|\nabla_{\mathbf{U}_g}\frac{\tilde{f}}{N}\right\|_F^2 + \left\|\nabla_{\mathbf{U}_{(i),l}}\tilde{f}_i\right\|_F^2\right)\left(272\beta B_1^2 + 400L_gB_1^6\right)$$

Also, we know

$$\tilde{f}_i(\mathbf{U}_{g,\tau+1}, \mathbf{V}_{(i),g,\tau+\frac{1}{2}}, \mathbf{U}_{(i),l,\tau+\frac{1}{2}}, \mathbf{V}_{(i),l,\tau+1}) - \tilde{f}_i(\mathbf{U}_{g,\tau}, \mathbf{V}_{(i),g,\tau}, \mathbf{U}_{(i),l,\tau}, \mathbf{V}_{(i),l,\tau})$$

$$\leq \left\langle\nabla_{\mathbf{U}_g}\tilde{f}_i, \mathbf{U}_{g,\tau+1} - \mathbf{U}_{g,\tau}\right\rangle + \left\langle\nabla_{\mathbf{V}_{(i),g}}\tilde{f}_i, \mathbf{V}_{(i),g,\tau+\frac{1}{2}} - \mathbf{V}_{(i),g,\tau}\right\rangle$$

$$+ \left\langle\nabla_{\mathbf{U}_{(i),l}}\tilde{f}_i, \mathbf{U}_{(i),l,\tau+\frac{1}{2}} - \mathbf{U}_{(i),l,\tau}\right\rangle + \left\langle\nabla_{\mathbf{V}_{(i),l}}\tilde{f}_i, \mathbf{V}_{(i),l,\tau+1} - \mathbf{V}_{(i),l,\tau}\right\rangle$$

$$+ \frac{L}{2}\left(\|\mathbf{U}_{g,\tau+1} - \mathbf{U}_{g,\tau}\|_F^2 + \left\|\mathbf{V}_{(i),g,\tau+\frac{1}{2}} - \mathbf{V}_{(i),g,\tau}\right\|_F^2 + \left\|\mathbf{U}_{(i),l,\tau+\frac{1}{2}} - \mathbf{U}_{(i),l,\tau}\right\|_F^2 + \|\mathbf{V}_{(i),l,\tau+1} - \mathbf{V}_{(i),l,\tau}\|_F^2\right)$$

$$\leq -\eta\left(\left\langle\nabla_{\mathbf{U}_g}\tilde{f}_i, \nabla_{\mathbf{U}_g}\frac{\tilde{f}}{N}\right\rangle + \left\|\nabla_{\mathbf{U}_{(i),l}}\tilde{f}_i\right\|_F^2 + \left\|\nabla_{\mathbf{V}_{(i),g}}\tilde{f}_i\right\|_F^2 + \left\|\nabla_{\mathbf{V}_{(i),l}}\tilde{f}_i\right\|_F^2\right)$$

$$+ \eta^2\frac{L}{2}\left(\left\|\nabla_{\mathbf{U}_g}\frac{\tilde{f}}{N}\right\|_F^2 + \left\|\nabla_{\mathbf{U}_{(i),l}}\tilde{f}_i\right\|^2 + \left\|\nabla_{\mathbf{V}_{(i),g}}\tilde{f}_i\right\|_F^2 + \left\|\nabla_{\mathbf{V}_{(i),l}}\tilde{f}_i\right\|_F^2\right)$$

Therefore,

$$\tilde{f}_i(\mathbf{U}_{g,\tau+1}, \mathbf{V}_{(i),g,\tau+1}, \mathbf{U}_{(i),l,\tau+1}, \mathbf{V}_{(i),l,\tau+1}) - \tilde{f}_i(\mathbf{U}_{g,\tau}, \mathbf{V}_{(i),g,\tau}, \mathbf{U}_{(i),l,\tau}, \mathbf{V}_{(i),l,\tau})$$
$$= g_i(\mathbf{U}_{g,\tau+1}, \mathbf{U}_{(i),l,\tau+1}) - g_i(\mathbf{U}_{g,\tau+1}, \mathbf{U}_{(i),l,\tau+\frac{1}{2}})$$
$$+ \tilde{f}_i(\mathbf{U}_{g,\tau+1}, \mathbf{V}_{(i),g,\tau+\frac{1}{2}}, \mathbf{U}_{(i),l,\tau+\frac{1}{2}}, \mathbf{V}_{(i),l,\tau+1}) - \tilde{f}_i(\mathbf{U}_{g,\tau}, \mathbf{V}_{(i),g,\tau}, \mathbf{U}_{(i),l,\tau}, \mathbf{V}_{(i),l,\tau})$$
$$\leq -\eta \left( \left\langle \nabla_{\mathbf{U}_g} \tilde{f}_i, \nabla_{\mathbf{U}_g} \frac{\tilde{f}}{N} \right\rangle + \left\| \nabla_{\mathbf{U}_{(i),l}} \tilde{f}_i \right\|_F^2 + \left\| \nabla_{\mathbf{V}_{(i),g}} \tilde{f}_i \right\|_F^2 + \left\| \nabla_{\mathbf{V}_{(i),l}} \tilde{f}_i \right\|_F^2 \right)$$
$$+ \eta^2 \left( \frac{L}{2} + 272\beta B_1^2 + 400 L_g B_1^6 \right) \left( \left\| \nabla_{\mathbf{U}_g} \frac{\tilde{f}}{N} \right\|_F^2 + \left\| \nabla_{\mathbf{U}_{(i),l}} \tilde{f}_i \right\|_F^2 + \left\| \nabla_{\mathbf{V}_{(i),g}} \tilde{f}_i \right\|_F^2 + \left\| \nabla_{\mathbf{V}_{(i),l}} \tilde{f}_i \right\|_F^2 \right)$$

Summing both side for $i$ from 1 to $N$, we have:

$$\tilde{f}(\mathbf{U}_{g,\tau+1}, \{\mathbf{V}_{(i),g,\tau+1}, \mathbf{U}_{(i),l,\tau+1}, \mathbf{V}_{(i),l,\tau+1}\}) - \tilde{f}(\mathbf{U}_{g,\tau}, \{\mathbf{V}_{(i),g,\tau}, \mathbf{U}_{(i),l,\tau}, \mathbf{V}_{(i),l,\tau}\})$$
$$= \sum_{i=1}^{N} \tilde{f}_i(\mathbf{U}_{g,\tau+1}, \mathbf{V}_{(i),g,\tau+1}, \mathbf{U}_{(i),l,\tau+1}, \mathbf{V}_{(i),l,\tau+1}) - \tilde{f}_i(\mathbf{U}_{g,\tau}, \mathbf{V}_{(i),g,\tau}, \mathbf{U}_{(i),l,\tau}, \mathbf{V}_{(i),l,\tau})$$
$$\leq -\eta \left( \left\| \frac{1}{\sqrt{N}} \nabla_{\mathbf{U}_g} \tilde{f} \right\|_F^2 + \sum_{i=1}^{N} \left\| \nabla_{\mathbf{V}_{(i),g}} \tilde{f}_i \right\|_F^2 + \left\| \nabla_{\mathbf{U}_{(i),l}} \tilde{f}_i \right\|_F^2 + \left\| \nabla_{\mathbf{V}_{(i),l}} \tilde{f}_i \right\|_F^2 \right)$$
$$+ \eta^2 \left( \frac{L}{2} + 272\beta B_1^2 + 400 L_g B_1^6 \right) \left( \left\| \frac{1}{\sqrt{N}} \nabla_{\mathbf{U}_g} \tilde{f} \right\|_F^2 + \sum_{i=1}^{N} \left\| \nabla_{\mathbf{V}_{(i),g}} \tilde{f}_i \right\|_F^2 \right.$$
$$+ \left. \left\| \nabla_{\mathbf{U}_{(i),l}} \tilde{f}_i \right\|_F^2 + \left\| \nabla_{\mathbf{V}_{(i),l}} \tilde{f}_i \right\|_F^2 \right)$$

Therefore, when $\eta \leq \frac{1}{\frac{L}{2} + 272\beta B_1^2 + 400 L_g B_1^6}$, we have:

$$\tilde{f}(\mathbf{U}_{g,\tau+1}, \{\mathbf{V}_{(i),g,\tau+1}, \mathbf{U}_{(i),l,\tau+1}, \mathbf{V}_{(i),l,\tau+1}\}) - \tilde{f}(\mathbf{U}_{g,\tau}, \{\mathbf{V}_{(i),g,\tau}, \mathbf{U}_{(i),l,\tau}, \mathbf{V}_{(i),l,\tau}\})$$
$$\leq -\frac{\eta}{2} \left( \left\| \frac{1}{\sqrt{N}} \nabla_{\mathbf{U}_g} \tilde{f} \right\|_F^2 + \sum_{i=1}^{N} \left\| \nabla_{\mathbf{V}_{(i),g}} \tilde{f}_i \right\|_F^2 + \left\| \nabla_{\mathbf{U}_{(i),l}} \tilde{f}_i \right\|_F^2 + \left\| \nabla_{\mathbf{V}_{(i),l}} \tilde{f}_i \right\|_F^2 \right)$$

This completes the proof. □

The above analysis suggests that Algorithm 1 will converge to points where the gradients vanish. In other words, Algorithm 1 converges into stationary points. In the rest of this section, we will analyze the local geometry around the optimal solutions to (4) and show that such geometric properties essentially allow Algorithm 1 to converge into the optimal solutions.

To do so, we will first find one set of optimal solutions that is close to the current iterate. In particular, given an iterate $\{\mathbf{U}_{g,\tau}, \mathbf{V}_{(i),g,\tau}, \mathbf{U}_{(i),l,\tau}, \mathbf{V}_{(i),l,\tau}\}$, define:

$$\widetilde{\mathbf{U}}_{g,\tau} = \mathbf{P}_{\hat{\mathbf{U}}_g} \mathbf{U}_{g,\tau} \left( \mathbf{U}_{g,\tau}^T \mathbf{P}_{\hat{\mathbf{U}}_g} \mathbf{U}_{g,\tau} \right)^{-\frac{1}{2}} \tag{19a}$$

$$\widetilde{\mathbf{U}}_{(i),l,\tau} = \mathbf{P}_{\hat{\mathbf{U}}_{(i),l}} \mathbf{U}_{(i),l,\tau} \left( \mathbf{U}_{(i),l,\tau}^T \mathbf{P}_{\hat{\mathbf{U}}_{(i),l}} \mathbf{U}_{(i),l,\tau} \right)^{-\frac{1}{2}} \tag{19b}$$

$$\widetilde{\mathbf{V}}_{(i),g,\tau} = \mathbf{M}_{(i)}^T \widetilde{\mathbf{U}}_{(i),g,\tau} \tag{19c}$$

$$\widetilde{\mathbf{V}}_{(i),l,\tau} = \mathbf{M}_{(i)}^T \widetilde{\mathbf{U}}_{(i),l,\tau} \tag{19d}$$

Indeed, the defined solutions are optimal since $\widetilde{\mathbf{U}}_{g,\tau}^T \widetilde{\mathbf{U}}_{g,\tau} = \mathbf{I}$, $\widetilde{\mathbf{U}}_{(i),l,\tau}^T \widetilde{\mathbf{U}}_{(i),l,\tau} = \mathbf{I}$, and $\widetilde{\mathbf{U}}_{g,\tau} \widetilde{\mathbf{V}}_{(i),g,\tau}^T + \widetilde{\mathbf{U}}_{(i),l,\tau} \widetilde{\mathbf{V}}_{(i),l,\tau}^T = \hat{\mathbf{U}}_g \hat{\mathbf{V}}_{(i),g}^T + \hat{\mathbf{U}}_{(i),l} \hat{\mathbf{V}}_{(i),l}^T$. Let the difference between iterates and these optimal solutions

be denoted as

$$\begin{cases} \Delta \mathbf{U}_{g,\tau} = \mathbf{U}_{g,\tau} - \widetilde{\mathbf{U}}_{g,\tau} \\ \Delta \mathbf{V}_{(i),g,\tau} = \mathbf{V}_{(i),g,\tau} - \widetilde{\mathbf{V}}_{(i),g,\tau} \\ \Delta \mathbf{U}_{(i),l,\tau} = \mathbf{U}_{(i),l,\tau} - \widetilde{\mathbf{U}}_{(i),l,\tau} \\ \Delta \mathbf{V}_{(i),l,\tau} = \mathbf{V}_{(i),l,\tau} - \widetilde{\mathbf{V}}_{(i),l,\tau} \end{cases} \tag{20}$$

Now we introduce the KKT conditions to (4). The KKT condition relies on the regularity condition that $\mathbf{M}_{(i)}$'s have rank at least $r_1 + r_2$. This assumption is easily satisfied if $\left\| \mathbf{E}_{(i)} \right\|$ is smaller than $\sigma_{\min}$.

**Lemma C.4 (KKT conditions for square error)** *Suppose that $\{\hat{\mathbf{U}}_g, \hat{\mathbf{U}}_{(i),l}, \hat{\mathbf{V}}_{(i),g}, \hat{\mathbf{V}}_{(i),l}\}$ is the optimal solution to problem (4) in the main paper with square error loss $\ell^{se}$, and $\mathbf{M}_{(i)}$ has rank at least $r_1 + r_2$. We have*

$$\sum_{i=1}^{N} \left( \hat{\mathbf{U}}_g \hat{\mathbf{V}}_{(i),g}^T + \hat{\mathbf{U}}_{(i),l} \hat{\mathbf{V}}_{(i),l}^T - \mathbf{M}_{(i)} \right) \hat{\mathbf{V}}_{(i),g} = 0 \tag{21a}$$

$$\left( \hat{\mathbf{U}}_g \hat{\mathbf{V}}_{(i),g}^T + \hat{\mathbf{U}}_{(i),l} \hat{\mathbf{V}}_{(i),l}^T - \mathbf{M}_{(i)} \right) \hat{\mathbf{V}}_{(i),l} = 0 \tag{21b}$$

$$\left( \hat{\mathbf{U}}_g \hat{\mathbf{V}}_{(i),g}^T + \hat{\mathbf{U}}_{(i),l} \hat{\mathbf{V}}_{(i),l}^T - \mathbf{M}_{(i)} \right)^T \hat{\mathbf{U}}_{(i),l} = 0 \tag{21c}$$

$$\left( \hat{\mathbf{U}}_g \hat{\mathbf{V}}_{(i),g}^T + \hat{\mathbf{U}}_{(i),l} \hat{\mathbf{V}}_{(i),l}^T - \mathbf{M}_{(i)} \right)^T \hat{\mathbf{U}}_g = 0 \tag{21d}$$

$$\hat{\mathbf{U}}_{(i),l}^T \hat{\mathbf{U}}_{(i),l} = \mathbf{I}, \hat{\mathbf{U}}_g^T \hat{\mathbf{U}}_g = \mathbf{I}, \hat{\mathbf{U}}_{(i),l}^T \hat{\mathbf{U}}_g = 0. \tag{21e}$$

**Proof.** The proof is presented in two parts.

*LICQ of (4).* We will first show the linear independence constraint qualification (LICQ) of the optimal $\hat{\mathbf{U}}_g$, $\{\hat{\mathbf{V}}_{(i),g}, \hat{\mathbf{U}}_{(i),l}, \hat{\mathbf{V}}_{(i),l}\}$.

We begin by proving $\hat{\mathbf{U}}_g$ has full column rank. Suppose otherwise, $\hat{\mathbf{U}}_g$ has rank $r_1' < r_1$. Since $\mathbf{M}_{(i)}$ has rank at least $r_1 + r_2$, the residual $\mathbf{M}_{(i)} - \hat{\mathbf{U}}_g \hat{\mathbf{V}}_{(i),g}^T - \hat{\mathbf{U}}_{(i),l} \hat{\mathbf{V}}_{(i),l}^T$ has rank at least 1. Therefore we can find $\hat{\mathbf{U}}_g'$ such that $\hat{\mathbf{U}}_g'^T \hat{\mathbf{U}}_g' = \mathbf{I}$ and $f_i(\hat{\mathbf{U}}_g', \hat{\mathbf{V}}_{(i),g}, \hat{\mathbf{U}}_{(i),l}, \hat{\mathbf{V}}_{(i),l}) < f_i(\hat{\mathbf{U}}_g, \hat{\mathbf{V}}_{(i),g}, \hat{\mathbf{U}}_{(i),l}, \hat{\mathbf{V}}_{(i),l})$. This contradicts the fact that $\hat{\mathbf{U}}_g$ is optimal.

Next we will establish the LICQ of constraints. We define $h_{ijk}$ as the inner product between the $j$-th column of $\mathbf{U}_g$ and the $k$-th column of $\mathbf{U}_{(i),l}$, $h_{ijk}(\mathbf{x}) = [\mathbf{U}_g]_{:,j}^T [\mathbf{U}_{(i),l}]_{:,k}$. The constraints in (4) can be rewritten as $h_{ijk}(\hat{\mathbf{x}}) = 0, \forall i \in [r_1], \forall j \in [r_2], \forall k \in [N]$. LICQ requires $\nabla h_{ijk}(\hat{\mathbf{x}})$ to be linearly independent for all $ijk$ (Bertsekas, 1997, Proposition 3.1.1).

Suppose we can find constants $\psi_{ijk}$ such that $\sum_{i=1}^{N} \sum_{j=1}^{r_1} \sum_{k=1}^{r_2} \psi_{ijk} \nabla h_{ijk}(\hat{\mathbf{x}}) = 0$. We consider the partial derivative of $h_{ijk}$ over the $k'$-th column of $\mathbf{U}_{(i'),l}$. It is easy to derive,

$$\frac{\partial}{\partial [\mathbf{U}_{(i'),l}]_{:,k'}} h_{ijk}(\hat{\mathbf{x}}) = \delta_{ii'} \delta_{kk'} [\hat{\mathbf{U}}_g]_{:,j}$$

where $\delta_{ii'}$ is the Kronecker delta function. Then the constants $\psi_{ijk}$ should satisfy,

$$\sum_{j=1}^{r_2} \psi_{i'jk'} [\hat{\mathbf{U}}_g]_{:,j} = 0$$

As the columns of $\hat{\mathbf{U}}_g$ are linearly independent, $\psi_{i'jk'} = 0$ for each $j$. This holds for any $i'$ and $k'$. Therefore $\psi_{ijk} = 0$ for all $i, j, k$. This implies $\nabla h_{ijk}$'s are linearly independent.

*Proof of C.4.* The Lagrangian of the optimization problem (4) can be written as

$$\mathcal{L} = \frac{1}{2}\sum_{i=1}^{N}\left\|\mathbf{U}_g\mathbf{V}_{(i),g}^T + \mathbf{U}_{(i),l}\mathbf{V}_{(i),l}^T - \mathbf{M}_{(i)}\right\|_F^2 + \frac{\beta}{2}\left\|\mathbf{U}_g^T\mathbf{U}_g - \mathbf{I}\right\|_F^2 + \frac{\beta}{2}\left\|\mathbf{U}_{(i),l}^T\mathbf{U}_{(i),l} - \mathbf{I}\right\|_F^2$$
$$+ \mathrm{Tr}\left(\mathbf{\Lambda}_{6,(i)}\mathbf{U}_g^T\mathbf{U}_{(i),l}\right) \tag{22}$$

where $\mathbf{\Lambda}_{6,(i)}$ is the dual variable for the constraint $\mathbf{U}_g^T\mathbf{U}_{(i),l} = 0$.

Under the LICQ, we know that the optimality of $\hat{\mathbf{U}}_g, \{\hat{\mathbf{V}}_{(i),g}, \hat{\mathbf{U}}_{(i),l}, \hat{\mathbf{V}}_{(i),l}\}$ implies the KKT condition. Setting the gradient of $\mathcal{L}$ with respect to $\mathbf{V}_{(i),g}$ and $\mathbf{V}_{(i),l}$ to zero, we can prove equation 21d and equation 21c. Considering the constraint $\hat{\mathbf{U}}_g^T\hat{\mathbf{U}}_{(i),l} = 0$, we can solve them as $\hat{\mathbf{V}}_{(i),g} = \mathbf{M}_{(i)}^T\hat{\mathbf{U}}_g\left(\hat{\mathbf{U}}_g^T\hat{\mathbf{U}}_g\right)^{-1}$ and $\hat{\mathbf{V}}_{(i),l} = \mathbf{M}_{(i)}^T\hat{\mathbf{U}}_{(i),l}\left(\hat{\mathbf{U}}_{(i),l}^T\hat{\mathbf{U}}_{(i),l}\right)^{-1}$. Then we examine the gradient of $\mathcal{L}$ with respect to $\mathbf{U}_{(i),l}$:

$$\frac{\partial}{\partial\mathbf{U}_{(i),l}}\mathcal{L} = \left(\mathbf{U}_g\mathbf{V}_{(i),g}^T + \mathbf{U}_{(i),l}\mathbf{V}_{(i),l}^T - \mathbf{M}_{(i)}\right)\hat{\mathbf{V}}_{(i),l} + 2\beta\mathbf{U}_{(i),l}\left(\mathbf{U}_{(i),l}^T\mathbf{U}_{(i),l} - \mathbf{I}\right) + \mathbf{U}_g\mathbf{\Lambda}_{(6),i}^T$$

Substituting $\hat{\mathbf{V}}_{(i),g} = \mathbf{M}_{(i)}^T\hat{\mathbf{U}}_g\left(\hat{\mathbf{U}}_g^T\hat{\mathbf{U}}_g\right)^{-1}$ and $\hat{\mathbf{V}}_{(i),l} = \mathbf{M}_{(i)}^T\hat{\mathbf{U}}_{(i),l}\left(\hat{\mathbf{U}}_{(i),l}^T\hat{\mathbf{U}}_{(i),l}\right)^{-1}$ in the above gradient and setting it to zero, we have

$$\left(\hat{\mathbf{U}}_g\left(\hat{\mathbf{U}}_g^T\hat{\mathbf{U}}_g\right)^{-1}\hat{\mathbf{U}}_g^T + \hat{\mathbf{U}}_{(i),l}\left(\hat{\mathbf{U}}_{(i),l}^T\hat{\mathbf{U}}_{(i),l}\right)^{-1}\hat{\mathbf{U}}_{(i),l}^T - \mathbf{I}\right)\mathbf{M}_{(i)}\hat{\mathbf{V}}_{(i),l}$$
$$+ 2\beta\hat{\mathbf{U}}_{(i),l}\left(\hat{\mathbf{U}}_{(i),l}^T\hat{\mathbf{U}}_{(i),l} - \mathbf{I}\right) + \hat{\mathbf{U}}_g\mathbf{\Lambda}_{(6),i}^T = 0$$

Left multiplying both sides by $\hat{\mathbf{U}}_g^T$, we have $\mathbf{\Lambda}_{6,(i)} = 0$. Left multiplying both sides by $\hat{\mathbf{U}}_{(i),l}^T$, we have $\hat{\mathbf{U}}_{(i),l}^T\hat{\mathbf{U}}_{(i),l} - \mathbf{I} = 0$. Therefore we also have $\left(\hat{\mathbf{U}}_g\left(\hat{\mathbf{U}}_g^T\hat{\mathbf{U}}_g\right)^{-1}\hat{\mathbf{U}}_g^T + \hat{\mathbf{U}}_{(i),l}\left(\hat{\mathbf{U}}_{(i),l}^T\hat{\mathbf{U}}_{(i),l}\right)^{-1}\hat{\mathbf{U}}_{(i),l}^T - \mathbf{I}\right)\mathbf{M}_{(i)}\mathbf{V}_{(i),l} = 0$. This proves equation equation 21b. Now, setting the derivative of $\mathcal{L}$ with respect to $\mathbf{U}_g$ to zero, we have

$$\frac{\partial}{\partial\mathbf{U}_g}\mathcal{L} = \sum_{i=1}^{N}\left(\hat{\mathbf{U}}_g\hat{\mathbf{V}}_{(i),g}^T + \hat{\mathbf{U}}_{(i),l}\hat{\mathbf{V}}_{(i),l}^T - \mathbf{M}_{(i)}\right)\hat{\mathbf{V}}_{(i),g} + 2N\beta\hat{\mathbf{U}}_g\left(\hat{\mathbf{U}}_g^T\hat{\mathbf{U}}_g - \mathbf{I}\right) = 0$$

Left multiplying both sides by $\hat{\mathbf{U}}_g^T$, we have $\hat{\mathbf{U}}_g^T\hat{\mathbf{U}}_g - \mathbf{I} = 0$. We have thus proven equation 21a. This completes the proof for equation 21.

$\square$

We note that the KKT conditions provide a set of conditions that must be satisfied for *all* stationary points of (4). Since $\widetilde{\mathbf{U}}_{g,\tau}$, $\widetilde{\mathbf{V}}_{(i),g,\tau}$, $\widetilde{\mathbf{U}}_{(i),l,\tau}$, and $\widetilde{\mathbf{V}}_{(i),l,\tau}$ introduced in equation 19aequation 19cequation 19bequation 19d constitute *one* set of optimal solutions, and $\mathbf{R}_{(i)} = \mathbf{M}_{(i)} - \mathbf{U}_{g}^{\star}\mathbf{V}_{(i),g}^{\star T} - \mathbf{U}_{(i),l}^{\star}\mathbf{V}_{(i),l}^{\star T}$, we can rewrite the KKT conditions of the optimal solutions from Lemma C.4 as,

$$\begin{cases} \sum_{i=1}^{N}\mathbf{R}_{(i)}\widetilde{\mathbf{V}}_{(i),g} = 0 \\ \mathbf{R}_{(i)}\widetilde{\mathbf{V}}_{(i),l} = 0 \\ \mathbf{R}_{(i)}^T\widetilde{\mathbf{U}}_{(i),l} = 0, \quad \mathbf{R}_{(i)}^T\widetilde{\mathbf{U}}_g = 0 \\ \widetilde{\mathbf{U}}_{(i),l}^T\widetilde{\mathbf{U}}_{(i),l} = \mathbf{I}, \quad \widetilde{\mathbf{U}}_g^T\widetilde{\mathbf{U}}_g = \mathbf{I}, \widetilde{\mathbf{U}}_{(i),l}^T\widetilde{\mathbf{U}}_g = 0 \end{cases} \tag{23}$$

From KKT conditions, it is also straightforward to verify that,

$$\mathbf{P}_{\hat{\mathbf{U}}_g}\sum_{j=1}^{N}\mathbf{R}_{(j)} = \mathbf{P}_{\hat{\mathbf{U}}_{(i),l}}\mathbf{R}_{(i)} = 0$$

We will use these KKT conditions when analyzing the convergence of Algorithm 1.

We begin by analyzing the subspace spanned by $\mathbf{U}_g$ and $\mathbf{U}_{(i),l}$'s. The following lemma shows that the difference between the subspace spanned by the columns of $\mathbf{U}_g$ and $\mathbf{U}_{(i),l}$ and by the optimal solution cannot be too large.

**Lemma C.5** *For any set of optimal solutions to (4), $\hat{\mathbf{U}}_g, \hat{\mathbf{V}}_{(i),g}, \hat{\mathbf{U}}_{(i),l}, \hat{\mathbf{V}}_{(i),l}$, and $\mathbf{R}_{(i)} = \mathbf{M}_{(i)} - \hat{\mathbf{U}}_g \hat{\mathbf{V}}_{(i),g}^T - \hat{\mathbf{U}}_{(i),l} \hat{\mathbf{V}}_{(i),l}^T$, if there exists $\hat{\sigma}_{gap}^2, \hat{\theta} > 0$, such that (i) $\sigma_{min}^2 \left( \hat{\mathbf{U}}_g \hat{\mathbf{V}}_{(i),g}^T + \hat{\mathbf{U}}_{(i),l} \hat{\mathbf{V}}_{(i),l}^T \right) - \left\| \mathbf{R}_{(i)} \right\|^2 \geq \hat{\sigma}_{gap}^2$, (ii) $\left\| \mathbf{R}_{(i)} \right\| \leq \frac{\hat{\theta} \hat{\sigma}_{gap}^2}{8 \hat{\sigma}_{max}}$ for each $i$, and (iii) $\lambda_{\max} \left( \frac{1}{N} \sum_{i=1}^{N} \mathbf{P}_{\hat{\mathbf{U}}_{(i),l}} \right) \leq 1 - \hat{\theta}$, then we have,*

$$
\begin{aligned}
&\sum_{i=1}^{N} \left\| \left( \mathbf{I} - \mathbf{P}_{\mathbf{U}_{g,\tau}} - \mathbf{P}_{\mathbf{U}_{(i),l,\tau}} \right) \mathbf{M}_{(i)} \right\|_F^2 - \sum_{i=1}^{N} \left\| \mathbf{R}_{(i)} \right\|_F^2 \\
&\geq \frac{\hat{\theta} \hat{\sigma}_{gap}^2}{8} \left( \left\| \mathbf{P}_{\mathbf{U}_{g,\tau}} - \mathbf{P}_{\hat{\mathbf{U}}_g} \right\|_F^2 + \left\| \mathbf{P}_{\mathbf{U}_{(i),l,\tau}} - \mathbf{P}_{\hat{\mathbf{U}}_{(i),l}} \right\|_F^2 \right)
\end{aligned}
\tag{24}
$$

**Proof.** We define $\mathbf{M}_{(i),g} = \hat{\mathbf{U}}_g \hat{\mathbf{V}}_{(i),g}^T$ and $\mathbf{M}_{(i),l} = \hat{\mathbf{U}}_{(i),l} \hat{\mathbf{V}}_{(i),l}^T$, then $\mathbf{M}_{(i)} = \mathbf{M}_{(i),g} + \mathbf{M}_{(i),l} + \mathbf{R}_{(i)}$. We have,

$$
\begin{aligned}
&\left\| \left( \mathbf{I} - \mathbf{P}_{\mathbf{U}_{g,\tau}} - \mathbf{P}_{\mathbf{U}_{(i),l,\tau}} \right) \mathbf{M}_{(i)} \right\|_F^2 \\
&= \operatorname{Tr} \left( \left( \mathbf{I} - \mathbf{P}_{\mathbf{U}_{g,\tau}} - \mathbf{P}_{\mathbf{U}_{(i),l,\tau}} \right) \mathbf{M}_{(i)} \mathbf{M}_{(i)}^T \right) \\
&= \operatorname{Tr} \left( \left( \mathbf{I} - \mathbf{P}_{\mathbf{U}_{g,\tau}} - \mathbf{P}_{\mathbf{U}_{(i),l,\tau}} \right) \left( \mathbf{M}_{(i),g} + \mathbf{M}_{(i),l} \right) \left( \mathbf{M}_{(i),g} + \mathbf{M}_{(i),l} \right)^T \right) \\
&\quad + \operatorname{Tr} \left( \left( \mathbf{I} - \mathbf{P}_{\mathbf{U}_{g,\tau}} - \mathbf{P}_{\mathbf{U}_{(i),l,\tau}} \right) \mathbf{R}_{(i)} \mathbf{R}_{(i)}^T \right) \\
&\quad + \operatorname{Tr} \left( \left( \mathbf{I} - \mathbf{P}_{\mathbf{U}_{g,\tau}} - \mathbf{P}_{\mathbf{U}_{(i),l,\tau}} \right) \left( \mathbf{M}_{(i),g} \mathbf{R}_{(i)}^T + \mathbf{R}_{(i)} \mathbf{M}_{(i),g}^T \right) \right)
\end{aligned}
$$

where we applied the second equation from equation 23 that $\mathbf{R}_{(i)} \mathbf{M}_{(i),l}^T = \mathbf{R}_{(i)} \hat{\mathbf{V}}_{(i),l} \hat{\mathbf{U}}_{(i),l}^T = 0$.

Summing index $i$ from 1 to $N$, and considering the KKT condition $\sum_{i=1}^{N} \mathbf{M}_{(i),g} \mathbf{R}_{(i)}^T = 0$, we have,

$$
\begin{aligned}
&\sum_{i=1}^{N} \left\| \left( \mathbf{I} - \mathbf{P}_{\mathbf{U}_{g,\tau}} - \mathbf{P}_{\mathbf{U}_{(i),l,\tau}} \right) \mathbf{M}_{(i)} \right\|_F^2 \\
&= \sum_{i=1}^{N} \operatorname{Tr} \left( \left( \mathbf{I} - \mathbf{P}_{\mathbf{U}_{g,\tau}} - \mathbf{P}_{\mathbf{U}_{(i),l,\tau}} \right) \mathbf{M}_{(i)} \mathbf{M}_{(i)}^T \right) \\
&= \sum_{i=1}^{N} \operatorname{Tr} \left( \left( \mathbf{I} - \mathbf{P}_{\mathbf{U}_{g,\tau}} - \mathbf{P}_{\mathbf{U}_{(i),l,\tau}} \right) \left( \mathbf{M}_{(i),g} + \mathbf{M}_{(i),l} \right) \left( \mathbf{M}_{(i),g} + \mathbf{M}_{(i),l} \right)^T \right) \\
&\quad + \sum_{i=1}^{N} \operatorname{Tr} \left( \left( \mathbf{I} - \mathbf{P}_{\mathbf{U}_{g,\tau}} - \mathbf{P}_{\mathbf{U}_{(i),l,\tau}} \right) \mathbf{R}_{(i)} \mathbf{R}_{(i)}^T \right) \\
&\quad + \sum_{i=1}^{N} \operatorname{Tr} \left( -\mathbf{P}_{\mathbf{U}_{(i),l,\tau}} \left( \mathbf{M}_{(i),g} \mathbf{R}_{(i)}^T + \mathbf{R}_{(i)} \mathbf{M}_{(i),g}^T \right) \right)
\end{aligned}
$$

We use $\hat{\sigma}_{\min}$ to denote the smallest nonzero singular value of $\mathbf{M}_{(i),g} + \mathbf{M}_{(i),l}$, then $\left( \mathbf{M}_{(i),g} + \mathbf{M}_{(i),l} \right) \left( \mathbf{M}_{(i),g} + \mathbf{M}_{(i),l} \right)^T \succeq \left( \mathbf{P}_{\hat{\mathbf{U}}_g} + \mathbf{P}_{\hat{\mathbf{U}}_{(i),l}} \right) \hat{\sigma}_{\min}^2$. Hence,

$$
\begin{aligned}
&\operatorname{Tr} \left( \left( \mathbf{I} - \mathbf{P}_{\mathbf{U}_{g,\tau}} - \mathbf{P}_{\mathbf{U}_{(i),l,\tau}} \right) \left( \mathbf{M}_{(i),g} + \mathbf{M}_{(i),l} \right) \left( \mathbf{M}_{(i),g} + \mathbf{M}_{(i),l} \right)^T \right) \\
&\geq \operatorname{Tr} \left( \left( \mathbf{I} - \mathbf{P}_{\mathbf{U}_{g,\tau}} - \mathbf{P}_{\mathbf{U}_{(i),l,\tau}} \right) \left( \mathbf{P}_{\hat{\mathbf{U}}_g} + \mathbf{P}_{\hat{\mathbf{U}}_{(i),l}} \right) \right) \hat{\sigma}_{\min}^2 \left( \mathbf{M}_{(i),g} + \mathbf{M}_{(i),l} \right) \\
&= \left( r_1 + r_2 - \left\langle \mathbf{P}_{\mathbf{U}_{g,\tau}} + \mathbf{P}_{\mathbf{U}_{(i),l,\tau}}, \mathbf{P}_{\hat{\mathbf{U}}_g} + \mathbf{P}_{\hat{\mathbf{U}}_{(i),l}} \right\rangle \right) \hat{\sigma}_{\min}^2 \left( \mathbf{M}_{(i),g} + \mathbf{M}_{(i),l} \right)
\end{aligned}
$$

Similarly, since $\mathbf{R}_{(i)}\mathbf{R}_{(i)}^T \preceq \left(\mathbf{I} - \mathbf{P}_{\hat{\mathbf{U}}_g} - \mathbf{P}_{\hat{\mathbf{U}}_{(i),l}}\right)\sigma_{max}^2\left(\mathbf{R}_{(i)}\right)$, we have,

$$
\begin{aligned}
&\mathrm{Tr}\left(\left(\mathbf{I} - \mathbf{P}_{\mathbf{U}_{g,\tau}} - \mathbf{P}_{\mathbf{U}_{(i),l,\tau}}\right)\mathbf{R}_{(i)}\mathbf{R}_{(i)}^T\right) \\
&= \left\|\mathbf{R}_{(i)}\right\|_F^2 - \mathrm{Tr}\left(\left(\mathbf{P}_{\mathbf{U}_{g,\tau}} + \mathbf{P}_{\mathbf{U}_{(i),l,\tau}}\right)\mathbf{R}_{(i)}\mathbf{R}_{(i)}^T\right) \\
&\geq \left\|\mathbf{R}_{(i)}\right\|_F^2 - \mathrm{Tr}\left(\left(\mathbf{P}_{\mathbf{U}_{g,\tau}} + \mathbf{P}_{\mathbf{U}_{(i),l,\tau}}\right)\left(\mathbf{I} - \mathbf{P}_{\hat{\mathbf{U}}_g} - \mathbf{P}_{\hat{\mathbf{U}}_{(i),l}}\right)\right)\sigma_{max}^2\left(\mathbf{R}_{(i)}\right)
\end{aligned}
$$

As a result,

$$
\begin{aligned}
&\sum_{i=1}^N \left\|\left(\mathbf{I} - \mathbf{P}_{\mathbf{U}_{g,\tau}} - \mathbf{P}_{\mathbf{U}_{(i),l,\tau}}\right)\mathbf{M}_{(i)}\right\|_F^2 - \left\|\mathbf{R}_{(i)}\right\|_F^2 \\
&\geq \sum_{i=1}^N \mathrm{Tr}\left(\left(\mathbf{P}_{\mathbf{U}_{g,\tau}} + \mathbf{P}_{\mathbf{U}_{(i),l,\tau}}\right)\left(\mathbf{I} - \mathbf{P}_{\hat{\mathbf{U}}_g} - \mathbf{P}_{\hat{\mathbf{U}}_{(i),l}}\right)\right)\left(\sigma_{min}^2\left(\mathbf{M}_{(i),g} + \mathbf{M}_{(i),l}\right) - \sigma_{max}^2\left(\mathbf{R}_{(i)}\right)\right) \\
&+ \sum_{i=1}^N \mathrm{Tr}\left(-\mathbf{P}_{\mathbf{U}_{(i),l,\tau}}\left(\mathbf{M}_{(i),g}\mathbf{R}_{(i)}^T + \mathbf{R}_{(i)}\mathbf{M}_{(i),g}^T\right)\right) \\
&\geq \sum_{i=1}^N \left\langle\mathbf{P}_{\mathbf{U}_{g,\tau}} + \mathbf{P}_{\mathbf{U}_{(i),l,\tau}}, \mathbf{I} - \mathbf{P}_{\hat{\mathbf{U}}_g} - \mathbf{P}_{\hat{\mathbf{U}}_{(i),l}}\right\rangle\hat{\sigma}_{\mathrm{gap}}^2 - \sum_{i=1}^N 2\left\|\mathbf{P}_{\mathbf{U}_{(i),l,\tau}}\mathbf{M}_{(i),g}\right\|_F\left\|\mathbf{P}_{\mathbf{U}_{(i),l,\tau}}\mathbf{R}_{(i)}\right\|_F \\
&\geq \sum_{i=1}^N \left(r_1 + r_2 - \left\langle\mathbf{P}_{\mathbf{U}_{g,\tau}}, \mathbf{P}_{\hat{\mathbf{U}}_g}\right\rangle - \left\langle\mathbf{P}_{\mathbf{U}_{(i),l,\tau}}, \mathbf{P}_{\hat{\mathbf{U}}_{(i),l}}\right\rangle\right)\frac{\hat{\theta}\hat{\sigma}_{\mathrm{gap}}^2}{2} \\
&- \sum_{i=1}^N 2\left\|\mathbf{P}_{\mathbf{U}_{(i),l,\tau}}\left(\mathbf{I} - \mathbf{P}_{\hat{\mathbf{U}}_{(i),l}}\right)\right\|_F^2\hat{\sigma}_{\max}\left\|\mathbf{R}_i\right\| \\
&\geq \sum_{i=1}^N \left(r_1 + r_2 - \left\langle\mathbf{P}_{\mathbf{U}_{g,\tau}}, \mathbf{P}_{\hat{\mathbf{U}}_g}\right\rangle - \left\langle\mathbf{P}_{\mathbf{U}_{(i),l,\tau}}, \mathbf{P}_{\hat{\mathbf{U}}_{(i),l}}\right\rangle\right)\frac{\hat{\theta}\hat{\sigma}_{\mathrm{gap}}^2}{4}
\end{aligned}
$$

where the second inequality comes from the definition of $\hat{\sigma}_{\mathrm{gap}}^2$ and Cauchy-Schwartz inequality, the third inequality comes from Lemma 2 in Shi & Kontar (2024) and Lemma E.5. Then, since $r_1 - \left\langle\mathbf{P}_{\mathbf{U}_{g,\tau}}, \mathbf{P}_{\hat{\mathbf{U}}_g}\right\rangle = \frac{1}{2}\left\|\mathbf{P}_{\mathbf{U}_{g,\tau}} - \mathbf{P}_{\hat{\mathbf{U}}_g}\right\|_F^2$ and $r_2 - \left\langle\mathbf{P}_{\mathbf{U}_{(i),l,\tau}}, \mathbf{P}_{\hat{\mathbf{U}}_{(i),l}}\right\rangle = \frac{1}{2}\left\|\mathbf{P}_{\mathbf{U}_{(i),l,\tau}} - \mathbf{P}_{\hat{\mathbf{U}}_{(i),l}}\right\|_F^2$, this completes the proof. $\qquad\square$

The next lemma shows that $\left\|\mathbf{U}_{g,\tau}^T\mathbf{U}_{g,\tau} - \mathbf{I}\right\|$ and $\left\|\mathbf{U}_{(i),l,\tau}^T\mathbf{U}_{(i),l,\tau} - \mathbf{I}\right\|$'s are upper bounded by $\phi_\tau$.

**Lemma C.6** *Under the same conditions as Lemma C.5, we have*

$$
\sum_{i=1}^N \left(\left\|\mathbf{U}_{g,\tau}^T\mathbf{U}_{g,\tau} - \mathbf{I}\right\|_F^2 + \left\|\mathbf{U}_{(i),l,\tau}^T\mathbf{U}_{(i),l,\tau} - \mathbf{I}\right\|_F^2\right) \leq \frac{2\phi_\tau}{\beta} \tag{25}
$$

**Proof.** We decompose $\tilde{f}$ as,

$$
\begin{aligned}
2\tilde{f} &= 2\left(\phi_\tau + \frac{1}{2}\sum_{i=1}^{N}\left\|\mathbf{R}_{(i)}\right\|_F^2\right) \\
&= \sum_{i=1}^{N}\left\|\mathbf{U}_{g,\tau}\mathbf{V}_{(i),g,\tau}^T + \mathbf{U}_{(i),l,\tau}\mathbf{V}_{(i),l} - \mathbf{M}_{(i)}\right\|_F^2 + \beta\left(\left\|\mathbf{U}_{g,\tau}^T\mathbf{U}_{g,\tau} - \mathbf{I}\right\|_F^2 + \left\|\mathbf{U}_{(i),l,\tau}^T\mathbf{U}_{(i),l,\tau} - \mathbf{I}\right\|_F^2\right) \\
&= \sum_{i=1}^{N}\left\|\left(\mathbf{P}_{\mathbf{U}_{g,\tau}} + \mathbf{P}_{\mathbf{U}_{(i),l,\tau}}\right)\left(\mathbf{U}_{g,\tau}\mathbf{V}_{(i),g,\tau}^T + \mathbf{U}_{(i),l,\tau}\mathbf{V}_{(i),l} - \mathbf{M}_{(i)}\right) - \left(\mathbf{I} - \mathbf{P}_{\mathbf{U}_{g,\tau}} - \mathbf{P}_{\mathbf{U}_{(i),l,\tau}}\right)\mathbf{M}_{(i)}\right\|_F^2 \\
&\quad + \beta\left(\left\|\mathbf{U}_{g,\tau}^T\mathbf{U}_{g,\tau} - \mathbf{I}\right\|_F^2 + \left\|\mathbf{U}_{(i),l,\tau}^T\mathbf{U}_{(i),l,\tau} - \mathbf{I}\right\|_F^2\right) \\
&= \sum_{i=1}^{N}\left\|\mathbf{U}_{g,\tau}\mathbf{V}_{(i),g,\tau}^T + \mathbf{U}_{(i),l,\tau}\mathbf{V}_{(i),l} - \left(\mathbf{P}_{\mathbf{U}_{g,\tau}} + \mathbf{P}_{\mathbf{U}_{(i),l,\tau}}\right)\mathbf{M}_{(i)}\right\|_F^2 \\
&\quad + \left\|\left(\mathbf{I} - \mathbf{P}_{\mathbf{U}_{g,\tau}} - \mathbf{P}_{\mathbf{U}_{(i),l,\tau}}\right)\mathbf{M}_{(i)}\right\|_F^2 + \beta\left(\left\|\mathbf{U}_{g,\tau}^T\mathbf{U}_{g,\tau} - \mathbf{I}\right\|_F^2 + \left\|\mathbf{U}_{(i),l,\tau}^T\mathbf{U}_{(i),l,\tau} - \mathbf{I}\right\|_F^2\right)
\end{aligned}
\tag{26}
$$

Considering Lemma C.5, we know that

$$
\begin{aligned}
2\phi_\tau &\geq \sum_{i=1}^{N}\frac{\hat{\theta}\hat{\sigma}_{\text{gap}}^2}{8}\left(\left\|\mathbf{P}_{\mathbf{U}_{g,\tau}} - \mathbf{P}_{\hat{\mathbf{U}}_g}\right\|_F^2 + \left\|\mathbf{P}_{\mathbf{U}_{(i),l,\tau}} - \mathbf{P}_{\hat{\mathbf{U}}_{(i),l}}\right\|_F^2\right) + \beta\left\|\mathbf{U}_{g,\tau}^T\mathbf{U}_{g,\tau} - \mathbf{I}\right\|_F^2 \\
&\quad + \beta\left\|\mathbf{U}_{(i),l,\tau}^T\mathbf{U}_{(i),l,\tau} - \mathbf{I}\right\|_F^2 + \left\|\mathbf{U}_{g,\tau}\mathbf{V}_{(i),g,\tau}^T + \mathbf{U}_{(i),l,\tau}\mathbf{V}_{(i),l} - \left(\mathbf{P}_{\mathbf{U}_{g,\tau}} + \mathbf{P}_{\mathbf{U}_{(i),l,\tau}}\right)\mathbf{M}_{(i)}\right\|_F^2
\end{aligned}
\tag{27}
$$

This completes the proof as all terms on the right hand side are nonnegative. $\qquad\square$

Now we can prove that when the objective $\tilde{f}$ is close to the optimal value, the iterates should also be close to the set of optimal solutions defined in equation 19aequation 19cequation 19bequation 19d. We will first examine the norms of $\Delta\mathbf{U}_{g,\tau}$ and $\Delta\mathbf{U}_{(i),l,\tau}$ in Lemma C.7, then calculate the norms of $\Delta\mathbf{V}_{(i),g,\tau}$ and $\Delta\mathbf{V}_{(i),l,\tau}$ in Lemma C.8.

**Lemma C.7** *Under the same conditions of Lemma C.5, if we further have $\phi_\tau \leq \min\{\frac{9\beta}{128}, \frac{9\hat{\theta}\hat{\sigma}_{gap}^2}{1936}\}$, we have the following,*

$$
N\left\|\Delta\mathbf{U}_{g,\tau}\right\|_F^2 + \sum_{i=1}^{N}\left\|\Delta\mathbf{U}_{(i),l,\tau}\right\|_F^2 \leq C_1\phi_\tau
\tag{28}
$$

*where*

$$
C_1 = \left(\frac{64}{9\beta} + 32\left(\frac{4}{3}B_1^2 + B_1\right)^2\frac{1}{\hat{\theta}\hat{\sigma}_{gap}^2}\right)
\tag{29}
$$

*and $\Delta\mathbf{U}_{g,\tau}$ and $\Delta\mathbf{U}_{(i),l,\tau}$ are introduced in equation 20. Also,*

$$
\left\|\mathbf{U}_{g,\tau}\right\|, \left\|\mathbf{U}_{(i),l,\tau}\right\| \leq \sqrt{\frac{11}{8}}, \quad \forall i \in \{1,\cdots,N\}
\tag{30}
$$

**Proof.** We begin by calculating an upper bound of the norm of $\Delta \mathbf{U}_{g,\tau}$,

$$
\|\Delta \mathbf{U}_{g,\tau}\|_F
$$

$$
= \left\| \mathbf{U}_{g,\tau} - \mathbf{P}_{\hat{\mathbf{U}}_g} \mathbf{U}_{g,\tau} \left( \mathbf{U}_{g,\tau}^T \mathbf{P}_{\hat{\mathbf{U}}_g} \mathbf{U}_{g,\tau} \right)^{-\frac{1}{2}} \right\|_F
$$

$$
\leq \left\| \mathbf{U}_{g,\tau} - \mathbf{P}_{\hat{\mathbf{U}}_g} \mathbf{U}_{g,\tau} \right\|_F + \left\| \mathbf{P}_{\hat{\mathbf{U}}_g} \mathbf{U}_{g,\tau} - \mathbf{P}_{\hat{\mathbf{U}}_g} \mathbf{U}_{g,\tau} \left( \mathbf{U}_{g,\tau}^T \mathbf{P}_{\hat{\mathbf{U}}_g} \mathbf{U}_{g,\tau} \right)^{-\frac{1}{2}} \right\|_F
$$

$$
\leq \|\mathbf{U}_{g,\tau}\| \left\| \mathbf{P}_{\mathbf{U}_{g,\tau}} - \mathbf{P}_{\hat{\mathbf{U}}_g} \right\|_F + \left\| \mathbf{P}_{\hat{\mathbf{U}}_g} \mathbf{U}_{g,\tau} \right\| \left\| \mathbf{I} - \left( \mathbf{U}_{g,\tau}^T \mathbf{P}_{\hat{\mathbf{U}}_g} \mathbf{U}_{g,\tau} \right)^{-\frac{1}{2}} \right\|_F
$$

Notice that

$$
\left\| \mathbf{I} - \left( \mathbf{U}_{g,\tau}^T \mathbf{P}_{\hat{\mathbf{U}}_g} \mathbf{U}_{g,\tau} \right)^{-\frac{1}{2}} \right\|_F
$$

$$
\leq \left\| \mathbf{I} - \left( \mathbf{I} + (\mathbf{U}_{g,\tau}^T \mathbf{U}_{g,\tau} - \mathbf{I}) + \mathbf{U}_{g,\tau}^T \left( \mathbf{P}_{\hat{\mathbf{U}}_g} - \mathbf{P}_{\mathbf{U}_{g,\tau}} \right) \mathbf{U}_{g,\tau} \right)^{-\frac{1}{2}} \right\|_F
$$

Since $\phi_\tau \leq \frac{9\beta}{128}$, we know that $\sum_{i=1}^N (\|\mathbf{U}_{g,\tau}^T \mathbf{U}_{g,\tau} - \mathbf{I}\|_F^2 + \|\mathbf{U}_{(i),l,\tau}^T \mathbf{U}_{(i),l,\tau} - \mathbf{I}\|_F^2) \leq \frac{9}{64}$ from Lemma C.6. Therefore, $\|\mathbf{U}_{g,\tau}^T \mathbf{U}_{g,\tau} - \mathbf{I}\|_F \leq \frac{3}{8}$. And by triangle inequality of Frobenius norm, $\|\mathbf{U}_{g,\tau}\| \leq \|\mathbf{U}_{g,\tau}\|_F \leq \sqrt{1 + \frac{3}{8}} \leq \sqrt{\frac{11}{8}}$. Similarly, $\phi_\tau \leq \frac{9\hat{\theta}\hat{\sigma}_{\text{gap}}^2}{1936}$ and Lemma C.5 imply that $\left\| \mathbf{P}_{\hat{\mathbf{U}}_g} - \mathbf{P}_{\mathbf{U}_{g,\tau}} \right\|_F \leq \frac{3}{11}$. As a result, we know $\left\| (\mathbf{U}_{g,\tau}^T \mathbf{U}_{g,\tau} - \mathbf{I}) + \mathbf{U}_{g,\tau}^T \left( \mathbf{P}_{\hat{\mathbf{U}}_g} - \mathbf{P}_{\mathbf{U}_{g,\tau}} \right) \mathbf{U}_{g,\tau} \right\|_F \leq \frac{3}{4}$. Thus by Lemma E.2, we have,

$$
\left\| \mathbf{I} - \left( \mathbf{U}_{g,\tau}^T \mathbf{P}_{\hat{\mathbf{U}}_g} \mathbf{U}_{g,\tau} \right)^{-\frac{1}{2}} \right\|_F
$$

$$
\leq \frac{4}{3} \left\| (\mathbf{U}_{g,\tau}^T \mathbf{U}_{g,\tau} - \mathbf{I}) + \mathbf{U}_{g,\tau}^T \left( \mathbf{P}_{\hat{\mathbf{U}}_g} - \mathbf{P}_{\mathbf{U}_{g,\tau}} \right) \mathbf{U}_{g,\tau} \right\|_F
$$

$$
\leq \frac{4}{3} \left( \|\mathbf{U}_{g,\tau}^T \mathbf{U}_{g,\tau} - \mathbf{I}\| + B_1^2 \left\| \mathbf{P}_{\hat{\mathbf{U}}_g} - \mathbf{P}_{\mathbf{U}_{g,\tau}} \right\|_F \right)
$$

Therefore,

$$
\|\Delta \mathbf{U}_{g,\tau}\|_F^2
$$

$$
\leq \left( \frac{4}{3} B_1 \|\mathbf{U}_{g,\tau}^T \mathbf{U}_{g,\tau} - \mathbf{I}\| + \left( \frac{4}{3} B_1^2 + B_1 \right) \left\| \mathbf{P}_{\hat{\mathbf{U}}_g} - \mathbf{P}_{\mathbf{U}_{g,\tau}} \right\|_F \right)^2
$$

$$
\leq \frac{32}{9} \|\mathbf{U}_{g,\tau}^T \mathbf{U}_{g,\tau} - \mathbf{I}\|^2 + 2 \left( \frac{4}{3} B_1^2 + B_1 \right)^2 \left\| \mathbf{P}_{\hat{\mathbf{U}}_g} - \mathbf{P}_{\mathbf{U}_{g,\tau}} \right\|_F^2
$$

Similar upper bounds hold for $\Delta \mathbf{U}_{(i),l,\tau}$. Summing them up, we have,

$$
\sum_{i=1}^N \|\Delta \mathbf{U}_{g,\tau}\|_F^2 + \|\Delta \mathbf{U}_{(i),l,\tau}\|_F^2
$$

$$
\leq \sum_{i=1}^N \frac{32}{9} \left( \|\mathbf{U}_{g,\tau}^T \mathbf{U}_{g,\tau} - \mathbf{I}\|^2 + \|\mathbf{U}_{(i),l,\tau}^T \mathbf{U}_{(i),l,\tau} - \mathbf{I}\|^2 \right) +
$$

$$
2 \left( \frac{4}{3} B_1^2 + B_1 \right)^2 \left( \left\| \mathbf{P}_{\hat{\mathbf{U}}_g} - \mathbf{P}_{\mathbf{U}_{g,\tau}} \right\|_F^2 + \left\| \mathbf{P}^\star_{(i)} - \mathbf{P}_{\mathbf{U}_{(i),l,\tau}} \right\|_F^2 \right)
$$

$$
\leq \left( \frac{64}{9\beta} + 32 \left( \frac{4}{3} B_1^2 + B_1 \right)^2 \frac{1}{\hat{\theta}\hat{\sigma}_{\text{gap}}^2} \right) \phi_\tau
$$

We applied Lemma C.5, Lemma C.6, and equation 27 in the last inequality. $\qquad\square$

We continue to prove the upper bound of the norm of $\Delta \mathbf{V}_{(i),g,\tau}$ and $\Delta \mathbf{V}_{(i),l,\tau}$ introduced in equation 20.

**Lemma C.8** *Under the same conditions of Lemma C.7, we have the following,*

$$N \|\Delta \mathbf{U}_{g,\tau}\|_F^2 + \sum_{i=1}^N \left[ \|\Delta \mathbf{V}_{(i),g,\tau}\|_F^2 + \|\Delta \mathbf{U}_{(i),l,\tau}\|_F^2 + \|\Delta \mathbf{V}_{(i),l,\tau}\|_F^2 \right] \le C_2 \phi_\tau \tag{31}$$

*where*

$$C_2 = 16 + \frac{704 \hat{\sigma}_{\max}^2}{\beta} + 352 \hat{\sigma}_{\max}^2 C_1 + C_1 \tag{32}$$

This lemma presents a full perturbation analysis of $\tilde{f}$. It shows that if the function value is close to optimal, all iterates must also be close to one set of the optimal solutions.

**Proof.**

We use the decomposition equation 26 to derive upper bounds on the norm of $\Delta \mathbf{V}$'s.

By some algebra, we know that,

$$
\begin{aligned}
&\left( \mathbf{P}_{\mathbf{U}_{g,\tau}} + \mathbf{P}_{\mathbf{U}_{(i),l,\tau}} \right) \mathbf{M}_{(i)} - \mathbf{U}_{g,\tau} \mathbf{V}_{(i),g,\tau}^T - \mathbf{U}_{(i),l,\tau} \mathbf{V}_{(i),l} \\
&= \mathbf{U}_{g,\tau} \Delta \mathbf{V}_{(i),g,\tau}^T + \mathbf{U}_{(i),l,\tau} \Delta \mathbf{V}_{(i),l} + \mathbf{U}_{g,\tau} \left( \left( \mathbf{U}_{g,\tau}^T \mathbf{U}_{g,\tau} \right)^{-1} - \mathbf{I} \right) \widetilde{\mathbf{V}}_{(i),g,\tau}^T \\
&\quad + \mathbf{U}_{g,\tau} \left( \mathbf{U}_{g,\tau}^T \mathbf{U}_{g,\tau} \right)^{-1} \Delta \mathbf{U}_{g,\tau} \mathbf{M}_{(i)} + \mathbf{U}_{(i),l,\tau} \left( \left( \mathbf{U}_{(i),l,\tau}^T \mathbf{U}_{(i),l,\tau} \right)^{-1} - \mathbf{I} \right) \widetilde{\mathbf{V}}_{(i),l,\tau}^T \\
&\quad + \mathbf{U}_{(i),l,\tau} \left( \mathbf{U}_{(i),l,\tau}^T \mathbf{U}_{(i),l,\tau} \right)^{-1} \Delta \mathbf{U}_{(i),l,\tau} \mathbf{M}_{(i)}
\end{aligned}
$$

We will bound each term above. By triangle inequality, we have,

$$
\begin{aligned}
&\left\| \mathbf{U}_{g,\tau} \left( \left( \mathbf{U}_{g,\tau}^T \mathbf{U}_{g,\tau} \right)^{-1} - \mathbf{I} \right) \widetilde{\mathbf{V}}_{(i),g,\tau}^T + \mathbf{U}_{g,\tau} \left( \mathbf{U}_{g,\tau}^T \mathbf{U}_{g,\tau} \right)^{-1} \Delta \mathbf{U}_{g,\tau} \mathbf{M}_{(i)} \right\|_F \\
&\le \|\mathbf{U}_{g,\tau}\| \left\| \left( \mathbf{U}_{g,\tau}^T \mathbf{U}_{g,\tau} \right)^{-1} - \mathbf{I} \right\|_F \left\| \widetilde{\mathbf{V}}_{(i),g,\tau}^T \right\| + \|\mathbf{U}_{g,\tau}\| \left\| \left( \mathbf{U}_{g,\tau}^T \mathbf{U}_{g,\tau} \right)^{-1} \right\| \|\Delta \mathbf{U}_{g,\tau}\|_F \|\mathbf{M}_{(i)}\| \\
&\le 4\sqrt{\frac{11}{8}} \hat{\sigma}_{\max} \left\| \mathbf{U}_{g,\tau}^T \mathbf{U}_{g,\tau} - \mathbf{I} \right\|_F + 4\sqrt{\frac{11}{8}} \hat{\sigma}_{\max} \|\Delta \mathbf{U}_{g,\tau}\|_F
\end{aligned}
$$

where we again used Lemma E.1 to show that when $\left\| \mathbf{U}_{(i),l,\tau}^T \mathbf{U}_{(i),l,\tau} - \mathbf{I} \right\|_F$ and $\left\| \mathbf{U}_{g,\tau}^T \mathbf{U}_{g,\tau} - \mathbf{I} \right\|_F \le \frac{3}{4}$, we have $\left\| \left( \mathbf{U}_{g,\tau}^T \mathbf{U}_{g,\tau} \right)^{-1} \right\| \le 1 + 3 = 4$.

A similar bound holds that,

$$
\begin{aligned}
&\left\| \mathbf{U}_{(i),l,\tau} \left( \left( \mathbf{U}_{(i),l,\tau}^T \mathbf{U}_{(i),l,\tau} \right)^{-1} - \mathbf{I} \right) \widetilde{\mathbf{V}}_{(i),l,\tau}^T + \mathbf{U}_{(i),l,\tau} \left( \mathbf{U}_{(i),l,\tau}^T \mathbf{U}_{(i),l,\tau} \right)^{-1} \Delta \mathbf{U}_{(i),l,\tau} \mathbf{M}_{(i)} \right\|_F \\
&\le 4\sqrt{\frac{11}{8}} \hat{\sigma}_{\max} \left\| \mathbf{U}_{(i),l,\tau}^T \mathbf{U}_{(i),l,\tau} - \mathbf{I} \right\|_F + 4\sqrt{\frac{11}{8}} \hat{\sigma}_{\max} \|\Delta \mathbf{U}_{(i),l,\tau}\|_F
\end{aligned}
$$

Then by Lemma E.3, we have,

$$
\left\| \left(\mathbf{P}_{\mathbf{U}_{g,\tau}} + \mathbf{P}_{\mathbf{U}_{(i),l,\tau}}\right) \mathbf{M}_{(i)} - \mathbf{U}_{g,\tau}\mathbf{V}_{(i),g,\tau}^T - \mathbf{U}_{(i),l,\tau}\mathbf{V}_{(i),l} \right\|^2
$$

$$
\geq \frac{1}{2} \left\| \mathbf{U}_{g,\tau}\Delta\mathbf{V}_{(i),g,\tau}^T + \mathbf{U}_{(i),l,\tau}\Delta\mathbf{V}_{(i),l} \right\|_F^2
$$

$$
- \left(4\sqrt{\frac{11}{8}}\hat{\sigma}_{\max}(\left\|\mathbf{U}_{g,\tau}^T\mathbf{U}_{g,\tau} - \mathbf{I}\right\|_F + \left\|\mathbf{U}_{(i),l,\tau}^T\mathbf{U}_{(i),l,\tau} - \mathbf{I}\right\|_F) + 4\sqrt{\frac{11}{8}}\hat{\sigma}_{\max}(\left\|\Delta\mathbf{U}_{(i),l,\tau}\right\|_F + \left\|\Delta\mathbf{U}_{(i),g,\tau}\right\|_F)\right)^2
$$

$$
\geq \frac{1}{2}\left\|\mathbf{U}_{g,\tau}\Delta\mathbf{V}_{(i),g,\tau}^T\right\|_F^2 + \frac{1}{2}\left\|\mathbf{U}_{(i),l,\tau}\Delta\mathbf{V}_{(i),l}\right\|_F^2
$$

$$
- 44\hat{\sigma}_{\max}^2(\left\|\mathbf{U}_{g,\tau}^T\mathbf{U}_{g,\tau} - \mathbf{I}\right\|_F^2 + \left\|\mathbf{U}_{(i),l,\tau}^T\mathbf{U}_{(i),l,\tau} - \mathbf{I}\right\|_F^2) - 44\hat{\sigma}_{\max}^2(\left\|\Delta\mathbf{U}_{(i),l,\tau}\right\|_F^2 + \left\|\Delta\mathbf{U}_{(i),g,\tau}\right\|_F^2)
$$

$$
\geq \frac{1}{8}\left\|\Delta\mathbf{V}_{(i),g,\tau}^T\right\|^2 + \frac{1}{8}\left\|\Delta\mathbf{V}_{(i),l}\right\|_F^2
$$

$$
- 44\hat{\sigma}_{\max}^2(\left\|\mathbf{U}_{g,\tau}^T\mathbf{U}_{g,\tau} - \mathbf{I}\right\|_F^2 + \left\|\mathbf{U}_{(i),l,\tau}^T\mathbf{U}_{(i),l,\tau} - \mathbf{I}\right\|_F^2) - 44\hat{\sigma}_{\max}^2(\left\|\Delta\mathbf{U}_{(i),l,\tau}\right\|_F^2 + \left\|\Delta\mathbf{U}_{(i),g,\tau}\right\|_F^2)
$$

where we used the condition $\left\|\mathbf{U}_{g,\tau}^T\mathbf{U}_{g,\tau} - \mathbf{I}\right\| \leq \frac{3}{4}$ and $\left\|\mathbf{U}_{(i),l,\tau}^T\mathbf{U}_{(i),l,\tau} - \mathbf{I}\right\| \leq \frac{3}{4}$ in the third inequality.

Summing both sides for $i$ from 1 to $N$, and considering Lemma C.5, Lemma C.7, and inequality equation 27, we have,

$$
\sum_{i=1}^{N}\left\|\Delta\mathbf{V}_{(i),g,\tau}\right\|_F^2 + \left\|\Delta\mathbf{V}_{(i),l,\tau}\right\|_F^2 \leq \left(16 + \frac{704\hat{\sigma}_{\max}^2}{\beta} + 352\hat{\sigma}_{\max}^2 C_1\right)\phi_\tau
$$

We can complete the proof by adding $\sum_{i=1}^{N}\left\|\Delta\mathbf{U}_{g,\tau}\right\|_F^2 + \left\|\Delta\mathbf{U}_{(i),l,\tau}\right\|_F^2$ on both sides. □

Now we can look at the following lemma which shows the gradient is aligned with the direction pointing from current iterate to one set of the optimal solution.

**Lemma C.9 (Gradient points to an optimal solution)** *Under the same conditions as Lemma C.8, if additionally $\phi_\tau \leq \frac{1}{\left(2\sqrt{2\max\{\hat{\sigma}_{\max}^2,1\}+4\beta}\right)^2 C_2^3}$, the following inequality would hold,*

$$
\left\langle\nabla_{\mathbf{U}_g}\tilde{f}, \Delta\mathbf{U}_{g,\tau}\right\rangle + \sum_{i=1}^{N}\left[\left\langle\nabla_{\mathbf{U}_{(i),l}}\tilde{f}, \Delta\mathbf{U}_{(i),l,\tau}\right\rangle + \left\langle\nabla_{\mathbf{V}_{(i),g}}\tilde{f}, \Delta\mathbf{V}_{(i),g,\tau}\right\rangle\right.
$$

$$
\left. + \left\langle\nabla_{\mathbf{V}_{(i),l}}\tilde{f}, \Delta\mathbf{V}_{(i),l}\right\rangle\right] \geq \phi_\tau
$$

$$(33)$$

This lemma shows that the geometry of the problem is benign.

**Proof.** Similar to Sun & Luo (2016), we define two notations $a_{i,\tau}$, $b_{i,\tau}$ as,

$$
a_{i,\tau} = \widetilde{\mathbf{U}}_{g,\tau}\Delta\mathbf{V}_{(i),g,\tau}^T + \Delta\mathbf{U}_{g,\tau}\widetilde{\mathbf{V}}_{(i),g,\tau}^T
$$

$$
+ \widetilde{\mathbf{U}}_{(i),l,\tau}\Delta\mathbf{V}_{(i),l,\tau}^T + \Delta\mathbf{U}_{(i),l,\tau}\widetilde{\mathbf{V}}_{(i),l,\tau}^T
$$

$$(34)$$

and

$$
b_{i,\tau} = \Delta\mathbf{U}_{g,\tau}\Delta\mathbf{V}_{(i),g,\tau}^T + \Delta\mathbf{U}_{(i),l,\tau}\Delta\mathbf{V}_{(i),l,\tau}^T
$$

$$(35)$$

Then we can calculate the inner product between the derivatives of $f_i$ and the difference between iterates and optimal values,

$$
\begin{aligned}
&\left\langle \nabla_{\mathbf{U}_g} f_i, \Delta\mathbf{U}_{g,\tau} \right\rangle + \left\langle \nabla_{\mathbf{V}_{(i),g}} f_i, \Delta\mathbf{V}_{(i),g,\tau} \right\rangle \\
&+ \left\langle \nabla_{\mathbf{U}_{(i),l}} f_i, \Delta\mathbf{U}_{(i),l,\tau} \right\rangle + \left\langle \nabla_{\mathbf{V}_{(i),l,\tau}} f_i, \Delta\mathbf{V}_{(i),l,\tau} \right\rangle \\
&= \left\langle \mathbf{U}_{g,\tau}\mathbf{V}_{(i),g,\tau}^T + \mathbf{U}_{(i),l,\tau}\mathbf{V}_{(i),l,\tau}^T - \mathbf{M}_{(i)}, a_{i,\tau} + 2b_{i,\tau} \right\rangle \\
&= \left\langle a_{i,\tau} + b_{i,\tau} + \mathbf{R}_{(i)}, a_{i,\tau} + 2b_{i,\tau} \right\rangle \\
&= \|a_{i,\tau}\|_F^2 + 2\|b_{i,\tau}\|_F^2 + 3\left\langle a_{i,\tau}, b_{i,\tau} \right\rangle + \left\langle \mathbf{R}_{(i)}, a_{i,\tau} \right\rangle + 2\left\langle \mathbf{R}_{(i)}, b_{i,\tau} \right\rangle
\end{aligned}
$$

As

$$
\begin{aligned}
2f_i - \left\|\mathbf{R}_{(i)}\right\|_F^2 &= \left\|\mathbf{U}_{g,\tau}\mathbf{V}_{(i),g,\tau}^T + \mathbf{U}_{(i),l,\tau}\mathbf{V}_{(i),l,\tau}^T - \mathbf{M}_{(i)}\right\|_F^2 - \left\|\mathbf{R}_{(i)}\right\|_F^2 \\
&= \left\|a_{i,\tau} + b_{i,\tau} + \mathbf{R}_{(i)}\right\|_F^2 - \left\|\mathbf{R}_{(i)}\right\|_F^2 \\
&= \|a_{i,\tau}\|_F^2 + \|b_{i,\tau}\|_F^2 + 2\left\langle a_{i,\tau}, b_{i,\tau} \right\rangle + 2\left\langle \mathbf{R}_{(i)}, a_{i,\tau} \right\rangle + 2\left\langle \mathbf{R}_{(i)}, b_{i,\tau} \right\rangle
\end{aligned}
$$

We know that,

$$
\begin{aligned}
&\left\langle \nabla_{\mathbf{U}_g} f_i, \Delta\mathbf{U}_{g,\tau} \right\rangle + \left\langle \nabla_{\mathbf{V}_{(i),g}} f_i, \Delta\mathbf{V}_{(i),g,\tau} \right\rangle \\
&+ \left\langle \nabla_{\mathbf{U}_{(i),l}} f_i, \Delta\mathbf{U}_{(i),l,\tau} \right\rangle + \left\langle \nabla_{\mathbf{V}_{(i),l,\tau}} f_i, \Delta\mathbf{V}_{(i),l,\tau} \right\rangle \\
&= 2f_i + \|b_{i,\tau}\|_F^2 + \left\langle a_{i,\tau}, b_{i,\tau} \right\rangle + \left\langle \mathbf{R}_{(i)}, a_{i,\tau} \right\rangle - \left\|\mathbf{R}_{(i)}\right\|_F^2
\end{aligned}
$$

Similarly, we can calculate the inner product

$$
\begin{aligned}
&\left\langle \nabla_{\mathbf{U}_g} g_i, \Delta\mathbf{U}_{g,\tau} \right\rangle \\
&= 2\beta\,\mathrm{Tr}\left( \widetilde{\mathbf{U}}_{g,\tau}^T \Delta\mathbf{U}_{g,\tau} \widetilde{\mathbf{U}}_{g,\tau}^T \Delta\mathbf{U}_{g,\tau} + \widetilde{\mathbf{U}}_{g,\tau}^T \Delta\mathbf{U}_{g,\tau} \Delta\mathbf{U}_{g,\tau}^T \widetilde{\mathbf{U}}_{g,\tau} \right) \\
&+ 2\beta\,\mathrm{Tr}\left( 3\Delta\mathbf{U}_{g,\tau}^T \Delta\mathbf{U}_{g,\tau} \widetilde{\mathbf{U}}_{g,\tau}^T \Delta\mathbf{U}_{g,\tau} \right) + 2\beta\,\mathrm{Tr}\left( \Delta\mathbf{U}_{g,\tau}^T \Delta\mathbf{U}_{g,\tau} \Delta\mathbf{U}_{g,\tau}^T \Delta\mathbf{U}_{g,\tau} \right)
\end{aligned}
$$

Since

$$
\begin{aligned}
&\left\|\mathbf{U}_{g,\tau}^T \mathbf{U}_{g,\tau} - \mathbf{I}\right\|_F^2 \\
&= \left\|\widetilde{\mathbf{U}}_{g,\tau}^T \Delta\mathbf{U}_{g,\tau} + \Delta\mathbf{U}_{g,\tau}^T \widetilde{\mathbf{U}}_{g,\tau} + \Delta\mathbf{U}_{g,\tau}^T \Delta\mathbf{U}_{g,\tau}\right\|_F^2 \\
&= 2\left\|\widetilde{\mathbf{U}}_{g,\tau}^T \Delta\mathbf{U}_{g,\tau}\right\|_F^2 + 2\,\mathrm{Tr}\left( \Delta\mathbf{U}_{g,\tau}^T \widetilde{\mathbf{U}}_{g,\tau} \Delta\mathbf{U}_{g,\tau}^T \widetilde{\mathbf{U}}_{g,\tau} \right) \\
&+ 2\,\mathrm{Tr}\left( \widetilde{\mathbf{U}}_{g,\tau}^T \Delta\mathbf{U}_{g,\tau} \Delta\mathbf{U}_{g,\tau}^T \Delta\mathbf{U}_{g,\tau} \right) + 2\,\mathrm{Tr}\left( \Delta\mathbf{U}_{g,\tau}^T \widetilde{\mathbf{U}}_{g,\tau} \Delta\mathbf{U}_{g,\tau}^T \Delta\mathbf{U}_{g,\tau} \right) \\
&+ \left\|\Delta\mathbf{U}_{g,\tau}^T \Delta\mathbf{U}_{g,\tau}\right\|_F^2
\end{aligned}
$$

We know that

$$
\begin{aligned}
&\left\langle \nabla_{\mathbf{U}_g} g_i, \Delta\mathbf{U}_{g,\tau} \right\rangle \\
&= \beta\left\|\mathbf{U}_{g,\tau}^T \mathbf{U}_{g,\tau} - \mathbf{I}\right\|_F^2 + 2\beta\,\mathrm{Tr}\left( \widetilde{\mathbf{U}}_{g,\tau}^T \Delta\mathbf{U}_{g,\tau} \Delta\mathbf{U}_{g,\tau}^T \Delta\mathbf{U}_{g,\tau} \right) + \beta\left\|\Delta\mathbf{U}_{g,\tau}^T \Delta\mathbf{U}_{g,\tau}\right\|_F^2
\end{aligned}
$$

As a result,

$$
\begin{aligned}
&\left\langle \nabla_{\mathbf{U}_g} g_i, \Delta\mathbf{U}_{g,\tau} \right\rangle + \left\langle \nabla_{\mathbf{U}_{(i),l}} g_i, \Delta\mathbf{U}_{(i),l,\tau} \right\rangle \\
&= 2g_i + 2\beta\,\mathrm{Tr}\left( \widetilde{\mathbf{U}}_{g,\tau}^T \Delta\mathbf{U}_{g,\tau} \Delta\mathbf{U}_{g,\tau}^T \Delta\mathbf{U}_{g,\tau} \right) + 2\beta\,\mathrm{Tr}\left( \widetilde{\mathbf{U}}_{(i),l,\tau}^T \Delta\mathbf{U}_{(i),l,\tau} \Delta\mathbf{U}_{(i),l,\tau}^T \Delta\mathbf{U}_{(i),l,\tau} \right) \\
&+ \beta\left\|\Delta\mathbf{U}_{g,\tau}\right\|_F^4 + \beta\left\|\Delta\mathbf{U}_{(i),l,\tau}\right\|_F^4 \\
&\geq 2g_i - 2\beta\left\|\Delta\mathbf{U}_{g,\tau}\right\|_F^3 - 2\beta\left\|\Delta\mathbf{U}_{(i),l,\tau}\right\|_F^3 + \beta\left\|\Delta\mathbf{U}_{g,\tau}\right\|_F^4 + \beta\left\|\Delta\mathbf{U}_{(i),l,\tau}\right\|_F^4
\end{aligned}
$$

Therefore, we can sum up the two inequalities and derive,

$$
\begin{aligned}
&\left\langle \nabla_{\mathbf{U}_g} \tilde{f}_i, \mathbf{U}_{g,\tau} - \widetilde{\mathbf{U}}_{g,\tau} \right\rangle + \left\langle \nabla_{\mathbf{V}_{(i),g}} \tilde{f}_i, \mathbf{V}_{(i),g,\tau} - \widetilde{\mathbf{V}}_{(i),g,\tau} \right\rangle \\
&+ \left\langle \nabla_{\mathbf{U}_{(i),l}} \tilde{f}_i, \mathbf{U}_{(i),l,\tau} - \widetilde{\mathbf{U}}_{(i),l,\tau} \right\rangle + \left\langle \nabla_{\mathbf{V}_{(i),l,\tau}} \tilde{f}_i, \mathbf{V}_{(i),l,\tau} - \widetilde{\mathbf{V}}_{(i),l,\tau} \right\rangle \\
&\geq 2\tilde{f}_i - \left\| \mathbf{R}_{(i)} \right\|_F^2 \\
&+ \| b_{i,\tau} \|_F^2 + \langle a_{i,\tau}, b_{i,\tau} \rangle + \langle \mathbf{R}_{(i)}, a_{i,\tau} \rangle \\
&- 2\beta \| \Delta \mathbf{U}_{g,\tau} \|_F^3 - 2\beta \left\| \Delta \mathbf{U}_{(i),l,\tau} \right\|_F^3 + \beta \| \Delta \mathbf{U}_{g,\tau} \|_F^4 + \beta \left\| \Delta \mathbf{U}_{(i),l,\tau} \right\|_F^4
\end{aligned}
$$

We can sum up both sides for $i$ from 1 to $N$, considering the KKT condition that $\sum_{i=1}^N \langle \mathbf{R}_{(i)}, a_{i,\tau} \rangle = 0$, and the definition of $\phi_\tau$ that $\sum_{i=1}^N 2\tilde{f}_i - \left\| \mathbf{R}_{(i)} \right\|_F^2 = 2\phi_\tau$, we have,

$$
\begin{aligned}
&\sum_{i=1}^N \Big[ \left\langle \nabla_{\mathbf{U}_g} \tilde{f}_i, \mathbf{U}_{g,\tau} - \widetilde{\mathbf{U}}_{g,\tau} \right\rangle + \left\langle \nabla_{\mathbf{V}_{(i),g}} \tilde{f}_i, \mathbf{V}_{(i),g,\tau} - \widetilde{\mathbf{V}}_{(i),g,\tau} \right\rangle \\
&+ \left\langle \nabla_{\mathbf{U}_{(i),l}} \tilde{f}_i, \mathbf{U}_{(i),l,\tau} - \widetilde{\mathbf{U}}_{(i),l,\tau} \right\rangle + \left\langle \nabla_{\mathbf{V}_{(i),l,\tau}} \tilde{f}_i, \mathbf{V}_{(i),l,\tau} - \widetilde{\mathbf{V}}_{(i),l,\tau} \right\rangle \Big] \\
&\geq 2\phi_\tau \\
&+ \sum_{i=1}^N \Big[ \| b_{i,\tau} \|_F^2 + \langle a_{i,\tau}, b_{i,\tau} \rangle - 2\beta \| \Delta \mathbf{U}_{g,\tau} \|_F^3 - 2\beta \left\| \Delta \mathbf{U}_{(i),l,\tau} \right\|_F^3 + \beta \| \Delta \mathbf{U}_{g,\tau} \|_F^4 + \beta \left\| \Delta \mathbf{U}_{(i),l,\tau} \right\|_F^4 \Big] \\
&\geq 2\phi_\tau + \sum_{i=1}^N \Big[ - \| a_{i,\tau} \|_F \| b_{i,\tau} \|_F - 2\beta \| \Delta \mathbf{U}_{g,\tau} \|_F^3 - 2\beta \left\| \Delta \mathbf{U}_{(i),l,\tau} \right\|_F^3 \Big]
\end{aligned}
\tag{36}
$$

For the higher order terms,

$$
\begin{aligned}
&\sum_{i=1}^N \| a_{i,\tau} \|_F \| b_{i,\tau} \|_F \\
&\leq \sqrt{\sum_{i=1}^N \| a_{i,\tau} \|_F^2} \sqrt{\sum_{i=1}^N \| b_{i,\tau} \|_F^2} \\
&\leq \sqrt{\sum_{i=1}^N \left( \hat{\sigma}_{\max}(\| \Delta \mathbf{U}_{g,\tau} \|_F + \left\| \Delta \mathbf{U}_{(i),l,\tau} \right\|_F) + \left\| \Delta \mathbf{V}_{(i),g,\tau} \right\|_F + \left\| \Delta \mathbf{V}_{(i),l,\tau} \right\|_F \right)^2} \\
&\qquad \sqrt{\sum_{i=1}^N \left( \| \Delta \mathbf{U}_{g,\tau} \|_F \left\| \Delta \mathbf{V}_{(i),g,\tau} \right\|_F + \left\| \Delta \mathbf{U}_{(i),l,\tau} \right\|_F \left\| \Delta \mathbf{V}_{(i),l,\tau} \right\|_F \right)^2} \\
&\leq \sqrt{2 \sum_{i=1}^N \left( \hat{\sigma}_{\max}^2 (\| \Delta \mathbf{U}_{g,\tau} \|_F^2 + \left\| \Delta \mathbf{U}_{(i),l,\tau} \right\|_F^2) + \left\| \Delta \mathbf{V}_{(i),g,\tau} \right\|_F^2 + \left\| \Delta \mathbf{V}_{(i),l,\tau} \right\|_F^2 \right)} \\
&\qquad \sqrt{2 \sum_{i=1}^N \left( \| \Delta \mathbf{U}_{g,\tau} \|_F^2 \left\| \Delta \mathbf{V}_{(i),g,\tau} \right\|_F^2 + \left\| \Delta \mathbf{U}_{(i),l,\tau} \right\|_F^2 \left\| \Delta \mathbf{V}_{(i),l,\tau} \right\|_F^2 \right)} \\
&\leq \sqrt{2 \max\{\hat{\sigma}_{\max}^2, 1\}} \sqrt{\sum_{i=1}^N \left( \| \Delta \mathbf{U}_{g,\tau} \|_F^2 + \left\| \Delta \mathbf{U}_{(i),l,\tau} \right\|_F^2 + \left\| \Delta \mathbf{V}_{(i),g,\tau} \right\|_F^2 + \left\| \Delta \mathbf{V}_{(i),l,\tau} \right\|_F^2 \right)} \\
&\qquad \sqrt{2} \sqrt{\left( \sum_{i=1}^N \| \Delta \mathbf{U}_{g,\tau} \|_F^2 \right) \left( \sum_{i=1}^N \left\| \Delta \mathbf{V}_{(i),g,\tau} \right\|_F^2 \right) + \left( \sum_{i=1}^N \left\| \Delta \mathbf{U}_{(i),l,\tau} \right\|_F^2 \right) \left( \sum_{i=1}^N \left\| \Delta \mathbf{V}_{(i),l,\tau} \right\|_F^2 \right)}
\end{aligned}
$$

$$\leq 2\sqrt{2\max\{\hat{\sigma}_{\max}^2,1\}}\sqrt{\sum_{i=1}^N\left(\|\Delta\mathbf{U}_{g,\tau}\|_F^2+\|\Delta\mathbf{U}_{(i),l,\tau}\|_F^2+\|\Delta\mathbf{V}_{(i),g,\tau}\|_F^2+\|\Delta\mathbf{V}_{(i),l,\tau}\|_F^2\right)}^3$$

$$\leq 2\sqrt{2\max\{\hat{\sigma}_{\max}^2,1\}}\left(\phi_\tau C_2\right)^{3/2}$$

where we applied Lemma C.8 in the last inequality.

And

$$2\beta\sum_{i=1}^N\left(\|\Delta\mathbf{U}_{g,\tau}\|_F^3+\|\Delta\mathbf{U}_{(i),l,\tau}\|_F^3\right)$$

$$\leq 4\beta\sqrt{\sum_{i=1}^N\left(\|\Delta\mathbf{U}_{g,\tau}\|_F^2+\|\Delta\mathbf{U}_{(i),l,\tau}\|_F^2+\|\Delta\mathbf{V}_{(i),g,\tau}\|_F^2+\|\Delta\mathbf{V}_{(i),l,\tau}\|_F^2\right)}^3$$

$$\leq 4\beta\left(\phi_\tau C_2\right)^{3/2}$$

where we also applied Lemma C.8 in the last inequality.

Thus when $\phi_\tau\leq\frac{1}{\left(2\sqrt{2\max\{\hat{\sigma}_{\max}^2,1\}}+4\beta\right)^2 C_2^3}$, the higher order $O(\phi_\tau^{3/2})$ terms in equation 36 are dominated by the leading order term $2\phi_\tau$. As a result, we have,

$$\sum_{i=1}^N\Big[\left\langle\nabla_{\mathbf{U}_g}\tilde{f}_i,\mathbf{U}_{g,\tau}-\widetilde{\mathbf{U}}_{g,\tau}\right\rangle+\left\langle\nabla_{\mathbf{V}_{(i),g}}\tilde{f}_i,\mathbf{V}_{(i),g,\tau}-\widetilde{\mathbf{V}}_{(i),g,\tau}\right\rangle$$
$$+\left\langle\nabla_{\mathbf{U}_{(i),l}}\tilde{f}_i,\mathbf{U}_{(i),l,\tau}-\widetilde{\mathbf{U}}_{(i),l,\tau}\right\rangle+\left\langle\nabla_{\mathbf{V}_{(i),l,\tau}}\tilde{f}_i,\mathbf{V}_{(i),l,\tau}-\widetilde{\mathbf{V}}_{(i),l,\tau}\right\rangle\Big]$$
$$\geq\phi_\tau$$

This completes the proof. $\square$

Finally, we are able to prove the PL inequality,

**Lemma C.10 (PL inequality)** *Under the same conditions as Lemma C.9, we have the following PL inequality,*

$$\left\|\frac{1}{\sqrt{N}}\nabla_{\mathbf{U}_g}\tilde{f}\right\|_F^2+\sum_{i=1}^N\|\nabla_{\mathbf{V}_{(i),g}}\tilde{f}_i\|_F^2+\|\nabla_{\mathbf{U}_{(i),l}}\tilde{f}_i\|_F^2+\|\nabla_{\mathbf{V}_{(i),l}}\tilde{f}_i\|_F^2\geq\frac{1}{C_2}\phi_\tau \tag{37}$$

**Proof.** The proof is straightforward. We first combine Lemma C.9 with Cauchy-Schwartz inequality,

$$\sqrt{\left\|\frac{1}{\sqrt{N}}\nabla_{\mathbf{U}_g}\tilde{f}\right\|_F^2+\sum_{i=1}^N\|\nabla_{\mathbf{V}_{(i),g}}\tilde{f}_i\|_F^2+\|\nabla_{\mathbf{U}_{(i),l}}\tilde{f}_i\|_F^2+\|\nabla_{\mathbf{V}_{(i),l}}\tilde{f}_i\|_F^2}$$

$$\sqrt{N\|\Delta\mathbf{U}_{g,\tau}\|_F^2+\sum_{i=1}^N\|\Delta\mathbf{V}_{(i),g,\tau}\|_F^2+\|\Delta\mathbf{U}_{(i),l,\tau}\|_F^2+\|\nabla_{\mathbf{V}_{(i),l}}\tilde{f}_i\|_F^2}$$

$$\geq\left\|\frac{1}{\sqrt{N}}\nabla_{\mathbf{U}_g}\tilde{f}\right\|_F\left\|\sqrt{N}\Delta\mathbf{U}_{g,\tau}\right\|_F+\sum_{i=1}^N\Big[\|\nabla_{\mathbf{V}_{(i),g}}\tilde{f}_i\|_F\|\Delta\mathbf{V}_{(i),g,\tau}\|_F$$

$$\|\nabla_{\mathbf{U}_{(i),l}}\tilde{f}_i\|_F\|\Delta\mathbf{U}_{(i),l,\tau}\|_F+\|\nabla_{\mathbf{V}_{(i),l}}\tilde{f}_i\|_F\|\Delta\mathbf{V}_{(i),l,\tau}\|_F\Big]$$

$$\geq\sum_{i=1}^N\Big[\left\langle\nabla_{\mathbf{U}_g}\tilde{f}_i,\mathbf{U}_{g,\tau}-\widetilde{\mathbf{U}}_{g,\tau}\right\rangle+\left\langle\nabla_{\mathbf{V}_{(i),g}}\tilde{f}_i,\mathbf{V}_{(i),g,\tau}-\widetilde{\mathbf{V}}_{(i),g,\tau}\right\rangle$$

$$+\left\langle\nabla_{\mathbf{U}_{(i),l}}\tilde{f}_i,\mathbf{U}_{(i),l,\tau}-\widetilde{\mathbf{U}}_{(i),l,\tau}\right\rangle+\left\langle\nabla_{\mathbf{V}_{(i),l,\tau}}\tilde{f}_i,\mathbf{V}_{(i),l,\tau}-\widetilde{\mathbf{V}}_{(i),l,\tau}\right\rangle\Big]$$

$$\geq\phi_\tau$$

Dividing both sides by $\sqrt{N \left\| \Delta \mathbf{U}_{g,\tau} \right\|_F^2 + \sum_{i=1}^N \left\| \Delta \mathbf{V}_{(i),g,\tau} \right\|_F^2 + \left\| \Delta \mathbf{U}_{(i),l,\tau} \right\|_F^2 + \left\| \nabla_{\mathbf{V}_{(i),l}} \tilde{f}_i \right\|_F^2}$ and applying Lemma C.8 will give us the desired result. $\qquad\square$

Finally, we will combine the derived results and show the linear convergence of Algorithm 1. Theorem 4 is a formal statement of the convergence guarantee.

**Theorem 4 (Formal version of Theorem 2 in the main paper)** *Under the following conditions,*

1. *The largest singular values of $\mathbf{M}_{(i)}$'s are upper bounded by $\hat{\sigma}_{\max}$.*

2. *There exists constants $\hat{\theta}$, $\hat{\sigma}_{gap}^2 > 0$ such that $\left\| \frac{1}{N} \sum_{i=1}^N \mathbf{P}_{\hat{\mathbf{U}}_{(i),l}} \right\| \leq 1 - \hat{\theta}$,*
   $\sigma_{min}^2 \left( \hat{\mathbf{U}}_g \hat{\mathbf{V}}_{(i),g}^T + \hat{\mathbf{U}}_{(i),l} \hat{\mathbf{V}}_{(i),l}^T \right) - \left\| \mathbf{R}_{(i)} \right\|^2 \geq \hat{\sigma}_{gap}^2$, and $\left\| \mathbf{R}_{(i)} \right\| \leq \frac{\hat{\theta} \hat{\sigma}_{gap}^2}{8 \hat{\sigma}_{\max}}$ *for every $i$.*

3. *There exist constants $B_1 > 1$ and $B_2 > \hat{\sigma}_{\max}$ such that $(\mathbf{U}_{g,1}, \{\mathbf{U}_{(i),l,1}\}, \{\mathbf{V}_{(i),g,1}\}, \{\mathbf{V}_{(i),l,1}\}) \in \mathcal{B}(B_1, B_2)$ and the constant stepsize $\eta$ is upper bounded by $\eta \leq \min\{ \frac{1}{10} \left( 2B_1 B_2 (2B_1 B_2 + \sigma_{\max}) + 2\beta B_1^2 (B_1^2 + 1) + \left( B_2 (2B_1 B_2 + \sigma_{\max}) + 2\beta B_1 (B_1^2 + 1) \right)^2 \right), \frac{1}{2\left( \frac{L}{2} + 272\beta B_1^2 + 400 L_g B_1^2 \right)}, 1 \}$.*

4. *The iterates are initialized properly $\phi_1 \leq \min\{ \frac{9\beta}{128}, \frac{9\hat{\theta} \hat{\sigma}_{gap}^2}{1936}, \frac{1}{\left( 2\sqrt{2 \max\{\sigma_{\max}^2, 1\} + 4\beta} \right)^2 C_2^3}, \frac{(B_1 - 1)^2}{C_1}, \frac{(B_2 - \sigma_{\max})^2}{C_2} \}$.*

*then all iterates reside in $\mathcal{B}(B_1, B_2)$ and the following holds,*

$$\phi_\tau \leq (1 - \eta B_3)^{\tau-1} \phi_1 \tag{38}$$

*where $B_3$ is a constant*

$$B_3 = \frac{1}{2C_2}$$

Notice that by combining Theorem 4 with Lemma C.1, we can immediately prove Theorem 2.

We shall emphasize that the conditions in Theorem 4 are more general than that in Theorem 2, as we do not make assumptions on the input noise structure. Most works on nonconvex matrix factorization (e.g., Sun & Luo (2016); Chen et al. (2020); Ye & Du (2021)) assume the input data are exactly low-rank, i.e., $\mathbf{R}_{(i)} = 0$. Theorem 4 relaxes such assumption by allowing $\mathbf{R}_{(i)}$ to be small nonzero matrices.

**Proof.** We will prove that the following claims hold for every $\tau \geq 1$ by induction.

1. $\left\| \mathbf{U}_{g,\tau} \right\|, \left\| \mathbf{U}_{(i),l,\tau} \right\| \leq B_1$ and $\left\| \mathbf{V}_{(i),g,\tau} \right\|, \left\| \mathbf{V}_{(i),l,\tau} \right\| \leq B_2$.

2. $\phi_\tau \leq (1 - \frac{\eta}{2C_2})^\tau \phi_0$

*Base case* At initialization $\tau = 1$, Claim 2 is simply true. Claim 1 is true because we assume at initialization, the norms of $\mathbf{U}_{g,1}$, $\mathbf{U}_{(i),l,1}$'s, $\mathbf{V}_{(i),g,1}$'s, $\mathbf{V}_{(i),l,1}$'s are upper bounded.

*Induction step* Now we assume Claim 1 and 2 are true for $\tau = 1, \cdots, t$ and prove that they still hold for $\tau = t + 1$.

Since Claim 2 is true for $\tau = t$, we know $\phi_t = (1 - \frac{\eta}{2C_2})^{t-1} \phi_1 \leq \phi_1$. As $\phi_1 \leq \frac{1}{(2\sqrt{2 \max\{\hat{\sigma}_{\max}^2, 1\} + 4\beta})^2 C_3^3}$ and $\phi_1 \leq \min\{ \frac{9\beta}{128}, \frac{9\hat{\theta} \hat{\sigma}_{\text{gap}}^2}{1936} \}$, we know that the result of Lemma C.10 holds.

Also from Lemma C.6 and the fact that $\phi_\tau \leq \frac{9\beta}{128}$, we know $\left\| \mathbf{U}_{g,\tau}^T \mathbf{U}_{g,\tau} - \mathbf{I} \right\|_F \leq \frac{3}{4}$ and $\left\| \mathbf{U}_{(i),l,\tau}^T \mathbf{U}_{(i),l,\tau} - \mathbf{I} \right\|_F \leq \frac{3}{4}$.

Since Claim 1 is true for $\tau = t$, and the stepsize is upper bounded by $\eta \leq \min\{\frac{1}{10}\Big(2B_1B_2(2B_1B_2 + \sigma_{\max}) +$

$2\beta B_1^2(B_1^2 + 1) + \big(B_2(2B_1B_2 + \sigma_{\max}) + 2\beta B_1(B_1^2 + 1)\big)^2\Big), \frac{1}{2\big(\frac{L}{2} + 272\beta B_1^2 + 400L_gB_1^2\big)}, 1\}$, all the conditions of Lemma C.3 are satisfied. Thus the result of Lemma C.3 holds for $\tau = t$.

Therefore we can combine the results of Lemma C.10 and Lemma C.3 to derive,

$$\phi_{t+1} \leq \phi_t - \frac{\eta}{2}\left(\left\|\frac{1}{\sqrt{N}}\nabla_{\mathbf{U}_g}\tilde{f}\right\|_F^2 + \sum_{i=1}^N \left\|\nabla_{\mathbf{V}_{(i),g}}\tilde{f}_i\right\|_F^2 + \left\|\nabla_{\mathbf{U}_{(i),l}}\tilde{f}_i\right\|_F^2 + \left\|\nabla_{\mathbf{V}_{(i),l}}\tilde{f}_i\right\|_F^2\right)$$

$$\leq \left(1 - \frac{\eta}{2}\frac{1}{C_2}\right)\phi_t$$

$$\leq (1 - \frac{\eta}{2C_2})^{t+1}\phi_0$$

. We thus prove Claim 2 for $\tau = t + 1$.

We then show that Claim 1 is true for $t + 1$. As $\phi_{t+1} < \phi_1 \leq \frac{(B_1 - 1)^2}{C_1}$, by Lemma C.7, we have,

$$\|\Delta\mathbf{U}_{g,t+1}\|_F \leq \sqrt{C_1\phi_{t+1}} \leq \sqrt{C_1\phi_0} \leq B_1 - 1$$

Thus by triangle inequality,

$$\|\mathbf{U}_{g,t+1}\| \leq \left\|\widetilde{\mathbf{U}}_{g,t+1}\right\| + \|\Delta\mathbf{U}_{g,t+1}\| \leq B_1$$

Similar bounds hold for $\mathbf{U}_{(i),l,t+1}$'s. Also, as $\phi_{t+1} < \phi_1 \leq \frac{(B_2 - \hat{\sigma}_{\max})^2}{C_1}$, by Lemma C.7, we have,

$$\left\|\Delta\mathbf{V}_{(i),g,t+1}\right\|_F \leq \sqrt{C_2\phi_{t+1}} \leq \sqrt{C_2\phi_0} \leq B_2 - \hat{\sigma}_{\max}$$

Also, by triangle inequality,

$$\left\|\mathbf{V}_{(i),g,t+1}\right\| \leq \left\|\widetilde{\mathbf{V}}_{(i),g,t+1}\right\| + \left\|\Delta\mathbf{U}_{(i),g,t+1}\right\| \leq B_2$$

Similar bounds hold for $\mathbf{V}_{(i),l,t+1}$'s.

This completes the proof. $\square$

## D   Proof of Theorem 3

The procedures to prove Theorem 3 are close to the first stage of the proof of Theorem 2. We first establish the KKT condition, then derive the sufficient decrease inequality for HMF.

The following lemma presents the KKT conditions to problem (4) in the main paper.

**Lemma D.1 (KKT conditions for general loss)** *Suppose that* $\{\hat{\mathbf{U}}_g, \hat{\mathbf{U}}_{(i),l}, \hat{\mathbf{V}}_{(i),g}, \hat{\mathbf{V}}_{(i),l}\}$ *is the optimal solution to problem (4) in the main paper with general loss metric* $\ell$, $\hat{\mathbf{U}}_g$ *and* $\hat{\mathbf{U}}_{(i),l}$*'s are non-singular, and* $\mathbf{M}_{(i)}$ *has rank at least* $r_1 + r_2$*. We have*

$$\sum_{i=1}^N \ell'\left(\mathbf{M}_{(i)}, \hat{\mathbf{U}}_g\hat{\mathbf{V}}_{(i),g}^T + \hat{\mathbf{U}}_{(i),l}\hat{\mathbf{V}}_{(i),l}^T\right)\hat{\mathbf{V}}_{(i),g} = 0 \tag{39a}$$

$$\ell'\left(\mathbf{M}_{(i)}, \hat{\mathbf{U}}_g\hat{\mathbf{V}}_{(i),g}^T + \hat{\mathbf{U}}_{(i),l}\hat{\mathbf{V}}_{(i),l}^T\right)\hat{\mathbf{V}}_{(i),l} = 0 \tag{39b}$$

$$\ell'\left(\mathbf{M}_{(i)}, \hat{\mathbf{U}}_g\hat{\mathbf{V}}_{(i),g}^T + \hat{\mathbf{U}}_{(i),l}\hat{\mathbf{V}}_{(i),l}^T\right)^T\hat{\mathbf{U}}_{(i),l} = 0 \tag{39c}$$

$$\ell'\left(\mathbf{M}_{(i)}, \hat{\mathbf{U}}_g\hat{\mathbf{V}}_{(i),g}^T + \hat{\mathbf{U}}_{(i),l}\hat{\mathbf{V}}_{(i),l}^T\right)^T\hat{\mathbf{U}}_g = 0 \tag{39d}$$

$$\hat{\mathbf{U}}_{(i),l}^T\hat{\mathbf{U}}_{(i),l} = \mathbf{I}, \hat{\mathbf{U}}_g^T\hat{\mathbf{U}}_g = \mathbf{I}, \hat{\mathbf{U}}_{(i),l}^T\hat{\mathbf{U}}_g = 0. \tag{39e}$$

**Proof.** The proof of the above lemma is very similar to the proof of Lemma C.4. The Lagrangian of the optimization problem (4) in the main paper can be written as

$$
\mathcal{L} = \sum_{i=1}^{N} \ell\left(\mathbf{M}_{(i)}, \mathbf{U}_g \mathbf{V}_{(i),g}^T + \mathbf{U}_{(i),l} \mathbf{V}_{(i),l}^T\right) + \frac{\beta}{2}\left\|\mathbf{U}_g^T \mathbf{U}_g - \mathbf{I}\right\|_F^2 + \frac{\beta}{2}\left\|\mathbf{U}_{(i),l}^T \mathbf{U}_{(i),l} - \mathbf{I}\right\|_F^2 \tag{40}
$$
$$
+ \operatorname{Tr}\left(\mathbf{\Lambda}_{7,(i)} \mathbf{U}_g^T \mathbf{U}_{(i),l}\right)
$$

where $\mathbf{\Lambda}_{7,(i)}$ is the dual variable for the constraint $\mathbf{U}_g^T \mathbf{U}_{(i),l} = 0$.

Similar to Lemma C.4, under the LICQ, we know that the optimality of $\hat{\mathbf{U}}_g, \{\hat{\mathbf{V}}_{(i),g}, \hat{\mathbf{U}}_{(i),l}, \hat{\mathbf{V}}_{(i),l}\}$ implies the KKT condition. Setting the gradient of $\mathcal{L}$ with respect to $\mathbf{V}_{(i),g}$ and $\mathbf{V}_{(i),l}$ to zero, we can prove equation 39d and equation 39c. Then we examine the gradient of $\mathcal{L}$ with respect to $\mathbf{U}_{(i),l}$:

$$
\frac{\partial}{\partial \mathbf{U}_{(i),l}}\mathcal{L} = \ell'\left(\mathbf{M}_{(i)}, \mathbf{U}_g \mathbf{V}_{(i),g}^T + \mathbf{U}_{(i),l} \mathbf{V}_{(i),l}^T\right)\hat{\mathbf{V}}_{(i),l} + 2\beta \mathbf{U}_{(i),l}\left(\mathbf{U}_{(i),l}^T \mathbf{U}_{(i),l} - \mathbf{I}\right) + \mathbf{U}_g \mathbf{\Lambda}_{(7),i}^T
$$

Left multiplying both sides by $\hat{\mathbf{U}}_g^T$, we have $\mathbf{\Lambda}_{7,(i)} = 0$. Left multiplying both sides by $\hat{\mathbf{U}}_{(i),l}^T$, we have $\hat{\mathbf{U}}_{(i),l}^T\hat{\mathbf{U}}_{(i),l} - \mathbf{I} = 0$. Therefore we have $\ell'\left(\mathbf{M}_{(i)}, \mathbf{U}_g \mathbf{V}_{(i),g}^T + \mathbf{U}_{(i),l} \mathbf{V}_{(i),l}^T\right)\hat{\mathbf{V}}_{(i),l} = 0$. This proves equation equation 39b.

Now, setting the derivative of $\mathcal{L}$ with respect to $\mathbf{U}_g$ to zero, we have

$$
\frac{\partial}{\partial \mathbf{U}_g}\mathcal{L} = \sum_{i=1}^{N} \ell'\left(\mathbf{M}_{(i)}, \hat{\mathbf{U}}_g \hat{\mathbf{V}}_{(i),g}^T + \hat{\mathbf{U}}_{(i),l} \hat{\mathbf{V}}_{(i),l}^T\right)\hat{\mathbf{V}}_{(i),g} + 2N\beta \hat{\mathbf{U}}_g\left(\hat{\mathbf{U}}_g^T \hat{\mathbf{U}}_g - \mathbf{I}\right) = 0
$$

Left multiplying both sides by $\hat{\mathbf{U}}_g^T$, we have $\hat{\mathbf{U}}_g^T\hat{\mathbf{U}}_g - \mathbf{I} = 0$. We have thus proven equation 39a. This completes the proof for equation 39. □

The following lemma gives an upper bound on the Lipschitz constant when all the norm of iterates and gradients are bounded.

**Lemma D.2 (Lipschitz continuity for general loss metrics)** *In region $\mathcal{B}(B_1, B_2)$ as defined in equation 16, if there exists a constant $B_4 > 0$ such that $\left\|\ell'(\mathbf{M}_{(i)}, \mathbf{U}_g \mathbf{V}_{(i),g}^T + \mathbf{U}_{(i),l} \mathbf{V}_{(i),l}^T)\right\| \leq B_4$, and loss metric $\ell$ is $L_\ell$ Lipschitz continuous in the region, then the objectives $\tilde{f}_i$ with $\beta = 0$ are Lipschitz continuous in the region,*

$$
\left\|\nabla \tilde{f}_i(\mathbf{U}_g', \mathbf{V}_{(i),g}', \mathbf{U}_{(i),l}', \mathbf{V}_{(i),l}') - \nabla \tilde{f}_i(\mathbf{U}_g, \mathbf{V}_{(i),g}, \mathbf{U}_{(i),l}, \mathbf{V}_{(i),l})\right\|_F
$$
$$
\leq L_{gen}\sqrt{\left\|\mathbf{U}_g' - \mathbf{U}_g\right\|_F^2 + \left\|\mathbf{V}_{(i),g}' - \mathbf{V}_{(i),g}\right\|_F^2 + \left\|\mathbf{U}_{(i),l}' - \mathbf{U}_{(i),l}\right\|_F^2 + \left\|\mathbf{V}_{(i),l}' - \mathbf{V}_{(i),l}\right\|_F^2} \tag{41}
$$

*for $\{\mathbf{U}_g', \{\mathbf{V}_{(i),g}'\}, \{\mathbf{U}_{(i),l}'\}, \{\mathbf{V}_{(i),l}'\}\}, \{\mathbf{U}_g, \{\mathbf{V}_{(i),g}\}, \{\mathbf{U}_{(i),l}\}, \{\mathbf{V}_{(i),l}\}\} \in \mathcal{B}(B_1, B_2)$, where $L_{gen}$ is a constant dependent on $B_1$, $B_2$, $L_\ell$, and $B_4$,*

$$
L_{gen} = \sqrt{8\left(B_2^2 + B_1^2\right)L_\ell^2 \max\{B_1^2, B_2^2\} + 2B_4^2}
$$

**Proof.** The proof is similar to the proof of Lemma C.2. We will calculate the gradient of $\tilde{f}_i$ over each variable, and bound the norm of the difference of the gradients. For simplicity, we use $\Delta\mathbf{M}_{(i)}'$ to denote $\Delta\mathbf{M}_{(i)}' = \mathbf{U}_g'(\mathbf{V}_{(i),g}')^T + \mathbf{U}_{(i),l}'(\mathbf{V}_{(i),l}')^T - \mathbf{U}_g \mathbf{V}_{(i),g}^T - \mathbf{U}_{(i),l} \mathbf{V}_{(i),l}^T$

$$
\nabla_{\mathbf{U}_g}\tilde{f}_i(\mathbf{U}_g', \mathbf{V}_{(i),g}', \mathbf{U}_{(i),l}', \mathbf{V}_{(i),l}') - \nabla_{\mathbf{U}_g}\tilde{f}_i(\mathbf{U}_g, \mathbf{V}_{(i),g}, \mathbf{U}_{(i),l}, \mathbf{V}_{(i),l})
$$
$$
= \ell'\left(\mathbf{M}_{(i)}, \mathbf{U}_g'(\mathbf{V}_{(i),g}')^T + \mathbf{U}_{(i),l}'(\mathbf{V}_{(i),l}')^T\right)\mathbf{V}_{(i),g}'
$$
$$
- \ell'\left(\mathbf{M}_{(i)}, \mathbf{U}_g(\mathbf{V}_{(i),g})^T + \mathbf{U}_{(i),l}(\mathbf{V}_{(i),l})^T\right)\mathbf{V}_{(i),g}
$$

Also by triangle inequalities, when $(\mathbf{U}'_g, \mathbf{V}'_{(i),g}, \mathbf{U}'_{(i),l}, \mathbf{V}'_{(i),l})$ and $(\mathbf{U}_g, \mathbf{V}_{(i),g}, \mathbf{U}_{(i),l}, \mathbf{V}_{(i),l})$ are in $\mathcal{B}(B_1, B_2)$, we have:

$$\left\| \nabla_{\mathbf{U}_g} \tilde{f}_i(\mathbf{U}'_g, \mathbf{V}'_{(i),g}, \mathbf{U}'_{(i),l}, \mathbf{V}'_{(i),l}) - \nabla_{\mathbf{U}_g} \tilde{f}_i(\mathbf{U}_g, \mathbf{V}_{(i),g}, \mathbf{U}_{(i),l}, \mathbf{V}_{(i),l}) \right\|_F$$
$$\leq L_\ell B_2 \left\| \Delta \mathbf{M}'_{(i)} \right\|_F + B_4 \left\| \mathbf{V}'_{(i),g} - \mathbf{V}_{(i),g} \right\|_F$$

Then on the derivative over $\mathbf{V}_{(i),g}$,

$$\nabla_{\mathbf{V}_{(i),g}} \tilde{f}_i(\mathbf{U}'_g, \mathbf{V}'_{(i),g}, \mathbf{U}'_{(i),l}, \mathbf{V}'_{(i),l}) - \nabla_{\mathbf{V}_{(i),g}} \tilde{f}_i(\mathbf{U}_g, \mathbf{V}_{(i),g}, \mathbf{U}_{(i),l}, \mathbf{V}_{(i),l})$$
$$= \ell'\left(\mathbf{M}_{(i)}, \mathbf{U}'_g(\mathbf{V}'_{(i),g})^T + \mathbf{U}'_{(i),l}(\mathbf{V}'_{(i),l})^T\right)^T \mathbf{U}'_g - \ell'\left(\mathbf{M}_{(i)}, \mathbf{U}_g(\mathbf{V}_{(i),g})^T + \mathbf{U}_{(i),l}(\mathbf{V}_{(i),l})^T\right)^T \mathbf{U}_g$$

Thus by similar calculations, we have,

$$\left\| \nabla_{\mathbf{V}_{(i),g}} \tilde{f}_i(\mathbf{U}'_g, \mathbf{V}'_{(i),g}, \mathbf{U}'_{(i),l}, \mathbf{V}'_{(i),l}) - \nabla_{\mathbf{V}_{(i),g}} \tilde{f}_i(\mathbf{U}_g, \mathbf{V}_{(i),g}, \mathbf{U}_{(i),l}, \mathbf{V}_{(i),l}) \right\|_F$$
$$\leq L_\ell B_1 \left\| \Delta \mathbf{M}'_{(i)} \right\|_F + B_4 \left\| \mathbf{U}'_{(i),g} - \mathbf{U}_{(i),g} \right\|_F$$

And,

$$\left\| \nabla_{\mathbf{U}_{(i),l}} \tilde{f}_i(\mathbf{U}'_g, \mathbf{V}'_{(i),g}, \mathbf{U}'_{(i),l}, \mathbf{V}'_{(i),l}) - \nabla_{\mathbf{U}_{(i),l}} \tilde{f}_i(\mathbf{U}_g, \mathbf{V}_{(i),g}, \mathbf{U}_{(i),l}, \mathbf{V}_{(i),l}) \right\|_F$$
$$\leq L_\ell B_2 \left\| \Delta \mathbf{M}'_{(i)} \right\|_F + B_4 \left\| \mathbf{V}'_{(i),l} - \mathbf{V}_{(i),l} \right\|_F$$

Also,

$$\left\| \nabla_{\mathbf{V}_{(i),l}} \tilde{f}_i(\mathbf{U}'_g, \mathbf{V}'_{(i),g}, \mathbf{U}'_{(i),l}, \mathbf{V}'_{(i),l}) - \nabla_{\mathbf{V}_{(i),l}} \tilde{f}_i(\mathbf{U}_g, \mathbf{V}_{(i),g}, \mathbf{U}_{(i),l}, \mathbf{V}_{(i),l}) \right\|_F$$
$$\leq L_\ell B_1 \left\| \Delta \mathbf{M}'_{(i)} \right\|_F + B_4 \left\| \mathbf{U}'_{(i),l} - \mathbf{U}_{(i),l} \right\|_F$$

Combining the 4 inequalities and the fact that $\left\| \Delta \mathbf{M}'_{(i)} \right\|_F \leq B_2 \left\| \mathbf{U}_g - \mathbf{U}'_g \right\|_F + B_1 \left\| \mathbf{V}_{(i),g} - \mathbf{V}'_{(i),g} \right\|_F + B_2 \left\| \mathbf{U}_{(i),l} - \mathbf{U}'_{(i),l} \right\|_F + B_1 \left\| \mathbf{V}_{(i),l} - \mathbf{V}'_{(i),l} \right\|_F$, we have:

$$\left\| \nabla \tilde{f}_i(\mathbf{U}'_g, \mathbf{V}'_{(i),g}, \mathbf{U}'_{(i),l}, \mathbf{V}'_{(i),l}) - \nabla \tilde{f}_i(\mathbf{U}_g, \mathbf{V}_{(i),g}, \mathbf{U}_{(i),l}, \mathbf{V}_{(i),l}) \right\|_F^2$$
$$= \left\| \nabla_{\mathbf{U}_g} \tilde{f}_i(\mathbf{U}'_g, \mathbf{V}'_{(i),g}, \mathbf{U}'_{(i),l}, \mathbf{V}'_{(i),l}) - \nabla_{\mathbf{U}_g} \tilde{f}_i(\mathbf{U}_g, \mathbf{V}_{(i),g}, \mathbf{U}_{(i),l}, \mathbf{V}_{(i),l}) \right\|_F^2$$
$$+ \left\| \nabla_{\mathbf{V}_{(i),g}} \tilde{f}_i(\mathbf{U}'_g, \mathbf{V}'_{(i),g}, \mathbf{U}'_{(i),l}, \mathbf{V}'_{(i),l}) - \nabla_{\mathbf{V}_{(i),g}} \tilde{f}_i(\mathbf{U}_g, \mathbf{V}_{(i),g}, \mathbf{U}_{(i),l}, \mathbf{V}_{(i),l}) \right\|_F^2$$
$$+ \left\| \nabla_{\mathbf{U}_{(i),l}} \tilde{f}_i(\mathbf{U}'_g, \mathbf{V}'_{(i),g}, \mathbf{U}'_{(i),l}, \mathbf{V}'_{(i),l}) - \nabla_{\mathbf{U}_{(i),l}} \tilde{f}_i(\mathbf{U}_g, \mathbf{V}_{(i),g}, \mathbf{U}_{(i),l}, \mathbf{V}_{(i),l}) \right\|_F^2$$
$$+ \left\| \nabla_{\mathbf{V}_{(i),l}} \tilde{f}_i(\mathbf{U}'_g, \mathbf{V}'_{(i),g}, \mathbf{U}'_{(i),l}, \mathbf{V}'_{(i),l}) - \nabla_{\mathbf{V}_{(i),l}} \tilde{f}_i(\mathbf{U}_g, \mathbf{V}_{(i),g}, \mathbf{U}_{(i),l}, \mathbf{V}_{(i),l}) \right\|_F^2$$
$$\leq L_{gen}^2 \left( \left\| \mathbf{U}'_g - \mathbf{U}_g \right\|_F^2 + \left\| \mathbf{V}'_{(i),g} - \mathbf{V}_{(i),g} \right\|_F^2 + \left\| \mathbf{U}'_{(i),l} - \mathbf{U}_{(i),l} \right\|_F^2 + \left\| \mathbf{V}'_{(i),l} - \mathbf{V}_{(i),l} \right\|_F^2 \right)$$

where $L_{gen}$ is a constant defined as,

$$L_{gen} = \sqrt{8\left(B_2^2 + B_1^2\right) L_\ell^2 \max\{B_1^2, B_2^2\} + 2B_4^2} = O(L_\ell)$$

$\square$

With the established Lipschitz continuity of the objective, we can prove the convergence of Algorithm 1. Theorem 5 is a formal statement of such convergence guarantee.

**Theorem 5 (Formal version of Theorem 3 in the main paper)** *Under the following conditions,*

1. *There exist conctants $B_1, B_2, B_4 > 0$ such that the iterates are bounded, $(\mathbf{U}_g\{\mathbf{V}_{(i),g}, \mathbf{U}_{(i),l}, \mathbf{V}_{(i),l}\} \in \mathcal{B}(B_1, B_2)$ and the gradient norm is also upper bounded $\ell' \le B_4$.*

2. *$\ell$ is $L_\ell$-Lipschitz continuous and lower bounded by a constant.*

3. *The constant stepsize $\eta$ is upper bounded by $\eta \le \frac{1}{L_{gen}} = O(\frac{1}{L_\ell})$.*

*then for Algorithm* **with $\beta = 0$, the following holds,**

$$\min_{\tau \in [1,\cdots,T]} \left\| \nabla \tilde{f} \left( \mathbf{U}_{g,\tau}, \{\mathbf{V}_{(i),g,\tau}, \mathbf{U}_{(i),l,\tau}, \mathbf{V}_{(i),l,\tau}\} \right) \right\|_F^2 = O\left(\frac{1}{T}\right) \tag{42}$$

**Proof.** The proof is straightforward given Lemma D.2. As $\beta = 0$, we have,

$$\tilde{f}_i(\mathbf{U}_{g,\tau+1}, \mathbf{V}_{(i),g,\tau+1}, \mathbf{U}_{(i),l,\tau+1}, \mathbf{V}_{(i),l,\tau+1})$$
$$= \ell\left(\mathbf{M}_{(i)}, \mathbf{U}_{g,\tau+1}\mathbf{V}_{(i),g,\tau+1}^T + \mathbf{U}_{(i),l,\tau+1}\mathbf{V}_{(i),l,\tau+1}^T\right)$$
$$= \ell\Big(\mathbf{M}_{(i)}, \mathbf{U}_{g,\tau+1}\left(\mathbf{V}_{(i),g,\tau+\frac{1}{2}} + \mathbf{V}_{(i),l,\tau+1}\mathbf{U}_{(i),l,\tau+\frac{1}{2}}^T\mathbf{U}_{g,\tau+1}\left(\mathbf{U}_{g,\tau+1}^T\mathbf{U}_{g,\tau+1}\right)^{-1}\right)^T$$
$$+ \left(\mathbf{U}_{(i),l,\tau+\frac{1}{2}} - \mathbf{U}_{g,\tau+1}\left(\mathbf{U}_{g,\tau+1}^T\mathbf{U}_{g,\tau+1}\right)^{-1}\mathbf{U}_{g,\tau+1}^T\mathbf{U}_{(i),l,\tau+\frac{1}{2}}\right)\mathbf{V}_{(i),l,\tau+1}^T\Big)$$
$$= \ell\left(\mathbf{M}_{(i)}, \mathbf{U}_{g,\tau+1}\mathbf{V}_{(i),g,\tau+\frac{1}{2}}^T + \mathbf{U}_{(i),l,\tau+\frac{1}{2}}\mathbf{V}_{(i),l,\tau+1}^T\right)$$
$$= \tilde{f}_i(\mathbf{U}_{g,\tau+1}, \mathbf{V}_{(i),g,\tau+\frac{1}{2}}, \mathbf{U}_{(i),l,\tau+\frac{1}{2}}, \mathbf{V}_{(i),l,\tau+1})$$

Combining this and Lemma D.2, we have

$$\tilde{f}_i(\mathbf{U}_{g,\tau+1}, \mathbf{V}_{(i),g,\tau+1}, \mathbf{U}_{(i),l,\tau+1}, \mathbf{V}_{(i),l,\tau+1}) - \tilde{f}_i(\mathbf{U}_{g,\tau}, \mathbf{V}_{(i),g,\tau}, \mathbf{U}_{(i),l,\tau}, \mathbf{V}_{(i),l,\tau})$$
$$= \tilde{f}_i(\mathbf{U}_{g,\tau+1}, \mathbf{V}_{(i),g,\tau+\frac{1}{2}}, \mathbf{U}_{(i),l,\tau+\frac{1}{2}}, \mathbf{V}_{(i),l,\tau+1}) - \tilde{f}_i(\mathbf{U}_{g,\tau}, \mathbf{V}_{(i),g,\tau}, \mathbf{U}_{(i),l,\tau}, \mathbf{V}_{(i),l,\tau})$$
$$\le \left\langle \nabla_{\mathbf{U}_g}\tilde{f}_i, \mathbf{U}_{g,\tau+1} - \mathbf{U}_{g,\tau} \right\rangle + \left\langle \nabla_{\mathbf{V}_{(i),g}}\tilde{f}_i, \mathbf{V}_{(i),g,\tau+\frac{1}{2}} - \mathbf{V}_{(i),g,\tau} \right\rangle$$
$$+ \left\langle \nabla_{\mathbf{U}_{(i),l}}\tilde{f}_i, \mathbf{U}_{(i),l,\tau+\frac{1}{2}} - \mathbf{U}_{(i),l,\tau} \right\rangle + \left\langle \nabla_{\mathbf{V}_{(i),l}}\tilde{f}_i, \mathbf{V}_{(i),l,\tau+1} - \mathbf{V}_{(i),l,\tau} \right\rangle$$
$$+ \frac{L_{gen}}{2}\left( \|\mathbf{U}_{g,\tau+1} - \mathbf{U}_{g,\tau}\|_F^2 + \left\|\mathbf{V}_{(i),g,\tau+\frac{1}{2}} - \mathbf{V}_{(i),g,\tau}\right\|_F^2 + \left\|\mathbf{U}_{(i),l,\tau+\frac{1}{2}} - \mathbf{U}_{(i),l,\tau}\right\|_F^2 + \|\mathbf{V}_{(i),l,\tau+1} - \mathbf{V}_{(i),l,\tau}\|_F^2 \right)$$
$$\le -\eta\left( \left\langle \nabla_{\mathbf{U}_g}\tilde{f}_i, \nabla_{\mathbf{U}_g}\frac{\tilde{f}}{N} \right\rangle + \|\nabla_{\mathbf{U}_{(i),l}}\tilde{f}_i\|_F^2 + \|\nabla_{\mathbf{V}_{(i),g}}\tilde{f}_i\|_F^2 + \|\nabla_{\mathbf{V}_{(i),l}}\tilde{f}_i\|_F^2 \right)$$
$$+ \eta^2\frac{L_{gen}}{2}\left( \left\|\nabla_{\mathbf{U}_g}\frac{\tilde{f}}{N}\right\|_F^2 + \|\nabla_{\mathbf{U}_{(i),l}}\tilde{f}_i\|_F^2 + \|\nabla_{\mathbf{V}_{(i),g}}\tilde{f}_i\|_F^2 + \|\nabla_{\mathbf{V}_{(i),l}}\tilde{f}_i\|_F^2 \right)$$

Summing both side for $i$ from 1 to $N$, we have:

$$\tilde{f}(\mathbf{U}_{g,\tau+1}, \{\mathbf{V}_{(i),g,\tau+1}, \mathbf{U}_{(i),l,\tau+1}, \mathbf{V}_{(i),l,\tau+1}\}) - \tilde{f}(\mathbf{U}_{g,\tau}, \{\mathbf{V}_{(i),g,\tau}, \mathbf{U}_{(i),l,\tau}, \mathbf{V}_{(i),l,\tau}\})$$
$$= \sum_{i=1}^{N} \tilde{f}_i(\mathbf{U}_{g,\tau+1}, \mathbf{V}_{(i),g,\tau+1}, \mathbf{U}_{(i),l,\tau+1}, \mathbf{V}_{(i),l,\tau+1}) - \tilde{f}_i(\mathbf{U}_{g,\tau}, \mathbf{V}_{(i),g,\tau}, \mathbf{U}_{(i),l,\tau}, \mathbf{V}_{(i),l,\tau})$$
$$\le -\eta\left( \left\|\frac{1}{\sqrt{N}}\nabla_{\mathbf{U}_g}\tilde{f}\right\|_F^2 + \sum_{i=1}^{N} \|\nabla_{\mathbf{V}_{(i),g}}\tilde{f}_i\|_F^2 + \|\nabla_{\mathbf{U}_{(i),l}}\tilde{f}_i\|_F^2 + \|\nabla_{\mathbf{V}_{(i),l}}\tilde{f}_i\|_F^2 \right)$$
$$+ \eta^2\frac{L_{gen}}{2}\Big( \left\|\frac{1}{\sqrt{N}}\nabla_{\mathbf{U}_g}\tilde{f}\right\|_F^2 + \sum_{i=1}^{N} \|\nabla_{\mathbf{V}_{(i),g}}\tilde{f}_i\|_F^2$$
$$+ \|\nabla_{\mathbf{U}_{(i),l}}\tilde{f}_i\|_F^2 + \|\nabla_{\mathbf{V}_{(i),l}}\tilde{f}_i\|_F^2 \Big)$$

Therefore, when $\eta \leq \frac{1}{L_{gen}}$, we have:

$$\tilde{f}(\mathbf{U}_{g,\tau+1}, \{\mathbf{V}_{(i),g,\tau+1}, \mathbf{U}_{(i),l,\tau+1}, \mathbf{V}_{(i),l,\tau+1}\}) - \tilde{f}(\mathbf{U}_{g,\tau}, \{\mathbf{V}_{(i),g,\tau}, \mathbf{U}_{(i),l,\tau}, \mathbf{V}_{(i),l,\tau}\})$$

$$\leq -\frac{\eta}{2} \left( \left\| \frac{1}{\sqrt{N}} \nabla_{\mathbf{U}_g} \tilde{f} \right\|_F^2 + \sum_{i=1}^N \left\| \nabla_{\mathbf{V}_{(i),g}} \tilde{f}_i \right\|_F^2 + \left\| \nabla_{\mathbf{U}_{(i),l}} \tilde{f}_i \right\|_F^2 + \left\| \nabla_{\mathbf{V}_{(i),l}} \tilde{f}_i \right\|_F^2 \right)$$

Summing up both sides for $\tau$ from 1 to $T$, we have:

$$\sum_{\tau=1}^T \left( \left\| \frac{1}{\sqrt{N}} \nabla_{\mathbf{U}_g} \tilde{f} \right\|_F^2 + \sum_{i=1}^N \left\| \nabla_{\mathbf{V}_{(i),g}} \tilde{f}_i \right\|_F^2 + \left\| \nabla_{\mathbf{U}_{(i),l}} \tilde{f}_i \right\|_F^2 + \left\| \nabla_{\mathbf{V}_{(i),l}} \tilde{f}_i \right\|_F^2 \right)$$

$$\leq \frac{2}{\eta} \left( \tilde{f}(\mathbf{U}_{g,1}, \{\mathbf{V}_{(i),g,1}, \mathbf{U}_{(i),l,1}, \mathbf{V}_{(i),l,1}\}) - \tilde{f}(\mathbf{U}_{g,T+1}, \{\mathbf{V}_{(i),g,T+1}, \mathbf{U}_{(i),l,T+1}, \mathbf{V}_{(i),l,T+1}\}) \right)$$

As $\tilde{f}(\mathbf{U}_{g,T+1}, \{\mathbf{V}_{(i),g,T+1}, \mathbf{U}_{(i),l,T+1}, \mathbf{V}_{(i),l,T+1}\})$ is lower bounded by a constant, the right hand $\frac{2}{\eta} \left( \tilde{f}(\mathbf{U}_{g,1}, \{\mathbf{V}_{(i),g,1}, \mathbf{U}_{(i),l,1}, \mathbf{V}_{(i),l,1}\}) - \tilde{f}(\mathbf{U}_{g,T+1}, \{\mathbf{V}_{(i),g,T+1}, \mathbf{U}_{(i),l,T+1}, \mathbf{V}_{(i),l,T+1}\}) \right)$ is upper bounded by a constant. Hence,

$$\min_{\tau \in \{1,\cdots,T\}} \left( \left\| \frac{1}{\sqrt{N}} \nabla_{\mathbf{U}_g} \tilde{f} \right\|_F^2 + \sum_{i=1}^N \left\| \nabla_{\mathbf{V}_{(i),g}} \tilde{f}_i \right\|_F^2 + \left\| \nabla_{\mathbf{U}_{(i),l}} \tilde{f}_i \right\|_F^2 + \left\| \nabla_{\mathbf{V}_{(i),l}} \tilde{f}_i \right\|_F^2 \right) = O\left( \frac{1}{T} \right)$$

This completes the proof. $\qquad\square$

# E  Auxiliary Lemmas

This section discusses some helper lemmas useful for our main proofs. These lemmas are mostly derived from basic linear algebra.

**Lemma E.1** *For a symmetric matrix $\mathbf{A} \in \mathbb{R}^{r \times r}$, if $\|\mathbf{A}\|_F \leq \frac{3}{4}$, we have,*

$$\left\| \mathbf{I} - (\mathbf{I} + \mathbf{A})^{-1} \right\|_F \leq 4 \|\mathbf{A}\|_F$$

**Proof.** We have

$$\left\| \mathbf{I} - (\mathbf{I} + \mathbf{A})^{-1} \right\|_F = \left\| (\mathbf{I} + \mathbf{A})^{-1} (-\mathbf{A}) \right\|_F \leq (1 - \|\mathbf{A}\|_F)^{-1} \|\mathbf{A}\|_F \leq 4 \|\mathbf{A}\|_F$$

$\qquad\square$

The next lemma presents a similar result.

**Lemma E.2** *For a symmetric matrix $\mathbf{A} \in \mathbb{R}^{r \times r}$, if $\|\mathbf{A}\|_F \leq \frac{3}{4}$, we have,*

$$\left\| \mathbf{I} - (\mathbf{I} + \mathbf{A})^{-\frac{1}{2}} \right\|_F \leq \frac{4 \|\mathbf{A}\|_F}{3}$$

**Proof.** Since $\|\mathbf{A}\|_F \leq \frac{3}{4} < 1$, we can use the series

$$\left\| \mathbf{I} - (\mathbf{I} + \mathbf{A})^{-\frac{1}{2}} \right\|_F = \left\| \sum_{n=1}^\infty \frac{(2n-1)!!(-1)^n}{2^n n!} \mathbf{A}^n \right\|_F$$

$$\leq \sum_{n=1}^\infty \frac{(2n-1)!!(-1)^n}{2^n n!} \|\mathbf{A}\|_F^n = (1 - \|\mathbf{A}\|_F)^{-\frac{1}{2}} - 1$$

$$\leq \frac{\|\mathbf{A}\|_F}{\sqrt{1 - \|\mathbf{A}\|_F} \left( \sqrt{1 - \|\mathbf{A}\|_F} + 1 \right)} \leq \frac{4 \|\mathbf{A}\|_F}{3}$$

$\qquad\square$

**Lemma E.3** *For two matrices* $\mathbf{A}, \mathbf{B} \in \mathbb{R}^{r \times r}$, *we have,*

$$\|\mathbf{A} - \mathbf{B}\|_F^2 \geq \frac{1}{2} \|\mathbf{A}\|_F^2 - \|\mathbf{B}\|_F^2$$

**Proof.** We have

$$\|\mathbf{A} - \mathbf{B}\|_F^2 = \|\mathbf{A}\|_F^2 + \|\mathbf{B}\|_F^2 + 2 \langle \mathbf{A}, \mathbf{B} \rangle \geq \|\mathbf{A}\|_F^2 + \|\mathbf{B}\|_F^2 - \left( \frac{1}{2} \|\mathbf{A}\|_F^2 + 2 \|\mathbf{B}\|_F^2 \right) \geq \frac{1}{2} \|\mathbf{A}\|_F^2 - \|\mathbf{B}\|_F^2$$

$\square$

**Lemma E.4** *For two matrices* $\mathbf{A}, \mathbf{B} \in \mathbb{R}^{r \times r}$, *if* $\mathbf{A}$ *is invertable and* $\|\mathbf{A}^{-1}\| \|\mathbf{B}\| < 1$, *we have,*

$$\left\| (\mathbf{A} + \mathbf{B})^{-1} \right\| \leq \|\mathbf{A}^{-1}\| + \frac{\|\mathbf{A}^{-1}\|^2 \|\mathbf{B}\|}{1 - \|\mathbf{A}^{-1}\| \|\mathbf{B}\|}$$

**Proof.** We have

$$\left\| (\mathbf{A} + \mathbf{B})^{-1} - \mathbf{A}^{-1} \right\| = \left\| \left( \left( \mathbf{I} + \mathbf{B}\mathbf{A}^{-1} \right)^{-1} - \mathbf{I} \right) \mathbf{A}^{-1} \right\| \leq \|\mathbf{A}^{-1}\| \frac{\|\mathbf{A}^{-1}\| \|\mathbf{B}\|}{1 - \|\mathbf{A}^{-1}\| \|\mathbf{B}\|}.$$

The proof is completed by invoking triangle inequality. $\square$

The following lemma is a well-known result and provides an upper bound on the norm of product matrices.

**Lemma E.5** *For two matrices* $\mathbf{A} \in \mathbb{R}^{m \times n}$ *and* $\mathbf{B} \in \mathbb{R}^{n \times p}$, *we have,*

$$\|\mathbf{A}\mathbf{B}\|_F \leq \|\mathbf{A}\|_2 \|\mathbf{B}\|_F$$

*and*

$$\|\mathbf{A}\mathbf{B}\|_2 \leq \|\mathbf{A}\|_2 \|\mathbf{B}\|_2$$

**Proof.**

The proof can be found in Sun & Luo (2016). $\square$

