# OpenReview forum: "Heterogeneous Matrix Factorization: When Features Differ by Datasets"
_TMLR — Accepted by TMLR_

### Review · Reviewer_3bmd · 2026-02-02

**Summary Of Contributions:**

The authors propose a Heterogeneous Matrix Factorization (HMF), a scalable method for separating shared and source-specific factors in heterogeneous data while maintaining orthogonality. In addition, the authors establish theoretical convergence guarantees for HMF under squared error loss, showing convergence to solutions close to the ground truth. Finally, the authors demonstrate HMF’s practical effectiveness and flexibility, including integration with auto-encoders, across applications in video, time series (stock market), and recommender systems.

Strength:
- The paper addresses an interesting and timely problem.
- The manuscript is generally well written and easy to follow.
- The inclusion of theoretical guarantees is a notable strength, as such results are relatively uncommon in matrix factorization work.
- The presented use cases effectively motivate the applicability of the proposed algorithm and demonstrate its effectiveness, including improvements over several baselines.

Weaknesses:
- The proofs supporting the theoretical guarantees are lengthy and cannot be accommodated in the main body of the paper. Given TMLR’s space constraints, this venue may not be ideal for presenting the full technical development.
- The orthogonality assumption appears to be central to the proposed approach and would benefit from stronger justification, potentially including quantitative evidence from the empirical studies.
- The experimental results are only loosely connected to the theoretical claims, and the paper would benefit from clearer empirical validation of the stated guarantees.
- Several concepts (e.g., KKT conditions) are introduced without explanation or appropriate citations, which may hinder accessibility for a broader audience.

**Additional Comments:**

I skimmed through the proofs in the appendix, but I cannot guarantee their correctness.

**Audience:**

Yes

**Audience Explanation:**

At least some members of the TMLR audience would be interested in this work because it addresses a fundamental and recurring challenge in modern machine learning: learning shared structure across heterogeneous data sources while preserving source-specific information. The proposed HMF framework is appealing in that it combines scalability, theoretical convergence guarantees, and practical relevance, bridging a gap that between principled matrix factorization (or matrix completion) theory and real-world applications such as time series, video analysis, and recommender systems.

**Claims And Evidence:**

No

**Claims Explanation:**

- The orthogonality assumption appears to be central to the proposed approach and would benefit from stronger justification, potentially including quantitative evidence from the empirical studies.
- The theoretical claims would benefit from being explicitly connected to, and corroborated by, the experimental results for at least one of the use cases.

**Requested Changes:**

- The orthogonality assumption appears to be central to the proposed approach and would benefit from stronger justification, potentially including quantitative evidence from the empirical studies.
- The experimental results are only loosely connected to the theoretical claims, and the paper would benefit from clearer empirical validation of the stated guarantees.

---

> ### Author Response · Authors · 2026-06-03
> **Thank you for your encouraging comments!**
>
> Please see our detailed response below.
>
> ---
>
> ## Proof length and presentation
>
> > The proofs supporting the theoretical guarantees are lengthy.
>
> We appreciate this concern. The appendix is long because the analysis combines statistical perturbation analysis, feasibility-preserving correction steps, and local convergence analysis.
>
> That said, we agree that the current main text does not provide enough guidance for readers who do not want to follow every algebraic step. We will revise the theory section in three ways:
>
> 1. **Add a proof sketch in the main text.** Before the detailed theorem statements, we will add a short paragraph explaining the proof structure. The proof first establishes the sufficient descrease of the HMF algorithm in Lemma C.3, then uses a series of lemmas to analyze the landscape of the objective function. Eventually, we establish the geometry analysis between the gradient and optimality gap in Lemma C.8.
> 2. **Add a theorem-to-lemma dependency guide.** At the start of the appendix, we will include a concise map showing which lemmas support Theorems 1, 2, and 3.
> 3. **Separate intuition from technical verification.** The main paper will contain the high-level proof sketches and the role of each assumption, while the appendix will contain the complete proof details.
>
> This revision should make the theory easier to assess.
>
> ---
>
> ## Orthogonality assumption justification
>
> Intuitively, the constraint is natural because the goal is to separate shared and unique features: what is shared should not also be unique, and vice versa. A natural way to encode this separation is to assume that the shared and unique vectors span different subspaces. Orthogonality is a standard and interpretable way to enforce this distinction.
>
> Theoretically, orthogonality condition makes shared/local decomposition identifiable and also regularized the optimization trajectory. Without a constraint separating the shared subspace from each source-specific subspace, the same reconstruction can often be represented by shifting components between the shared and local terms.
>
> We will add this identifiability explanation to the model section and add a new synthetic ablation comparing HMF with and without the orthogonality correction step.
>
> **Experimental setup.** We use the synthetic ground-truth dataset and run both variants for 1000 epochs with the same random initialization.
>
> - **HMF with orthogonality:** the standard algorithm 1 discussed in the original manuscript.
> - **HMF without orthogonality:** the same optimization procedure but with the orthogonality correction removed.
>
> Results will be reported in the revised manuscript. At the final epoch, HMF with orthogonality achieves essentially exact recovery. Without orthogonality, the reconstruction loss decreases but the recovered subspaces remain inaccurate, especially for the local factors. The final global and local subspace errors are also much larger without orthogonality.
>
> This ablation directly supports the theoretical motivation. The model without orthogonal constraint converges slower as it does not respect the data structure during the optimization trajectory. The local-global separation is especially important because the purpose of HMF is to distinguish these factors. We will include this experiment as a new figure in the synthetic-data section.
>
> ---
>
> ## Connect experiments to theoretical claims
>
> The current synthetic section reports decreasing shared/local factor error, which validates the theoretical claim of linear convergence. To directly verify the theoretical claims, we add an optimality-gap experiment on the same synthetic dataset.
>
>  Specifically, we record the training loss $f(x_t)$ at every epoch and use $f(x_t)-f^*$ as the empirical optimality gap.This connects directly to Theorem 2, which states that the HMF optimality gap satisfies
>
> $\phi_t \leq (1-C\eta)^{t-1}\phi_1$
>
> under the theorem’s assumptions. We will add an additional convergence verification plot in our revised manuscript.
>
> ---
>
> ## KKT conditions
>
> We agree and will add a short explanation the first time KKT conditions are mentioned. The revised text will read approximately as follows:
>
> > The Karush-Kuhn-Tucker (KKT) conditions are first-order necessary optimality conditions for constrained optimization problems. For a problem with equality constraints $h(x)=0$ and inequality constraints $g(x)\le 0$, they consist of primal feasibility, stationarity of the Lagrangian, dual feasibility for inequality multipliers, and complementary slackness. In our setting, the relevant constraints include the shared/local orthogonality constraints $U_g^\top U_{i,l}=0$. Thus, convergence to a KKT point means convergence to a first-order stationary point of the constrained HMF objective.
>
> We will cite standard references such as Boyd and Vandenberghe, *Convex Optimization* (2004), and Bertsekas, *Nonlinear Programming* (1999). We will also add explanations about Theorem 3.

---

### Review · Reviewer_oSYi · 2026-04-19

**Summary Of Contributions:**

The paper addresses the task of analyzing multiple matrices that are assumed to be generated from a linear mixture of common factors, unique factors, and additive noise. The goal is to identify the shared and unique factors under orthogonality constraints.

The paper introduces a novel iterative algorithms that combines a corrective step and gradient descent. A key contribution is the thorough theoretical analysis. The paper provides theorems that show that the introduced algorithm will converge to optimal solutions that are close to the ground truth. The theorems provide meaningful characterization of the convergence and the statistical error.

The paper shows how the technique can be integrated into autoencoders to learn nonlinear feature mappings. The paper presents several experimental case studies.

**Audience:**

Yes

**Audience Explanation:**

The paper presents an interesting technique and the theoretical analysis is thorough. Matrix Factorization is an important problem that is of interest to many practitioners in diverse application domains. The extension to the heterogeneous setting with some common and some unique features seems natural and of interest.

**Broader Impact Concerns:**

None.

**Claims And Evidence:**

No

**Claims Explanation:**

The paper makes the following claims:

(C1) The paper introduces a novel algorithm to identify shared and unique factors for linear models with additive noise. The algorithm is easy to implement and intrinsically distributed.

(C2) Theoretical results show that the algorithm converges to optimal solutions which are close to the ground truth.

(C3) The algorithm can be integrated into an auto-encoder to learn nonlinear feature mappings.

(C4) Case studies showcase the algorithm’s benefits in a variety of applications.

Claims C1 and C2 are supported by the methodology and a very thorough theoretical analysis, which is a key contribution. I followed the proofs in the appendix and I did not detect any issues with the technical arguments.

Claim C3 is supported by a methodological description and experiments in a movie rating prediction setting. The ground truth nonlinear feature mappings are not known in this setting, but the method does learn nonlinear feature mappings as claimed.

Claim C4 is the most problematic. The presented case studies include video segmentation, but this is performed on a synthetic dataset with very artificial manipulation to create a setting that matches the model adopted in the paper. It is very likely that basic video segmentation techniques on real data would perform as effectively or better than the proposed method. The financial data analysis does not provide any insight beyond what could be achieved using numerous simple time-series segmentation techniques and calculating residuals of stock returns relative to the market index. The rating prediction setting falls far short of what would be remotely interesting for the recommender systems research community. The baseline methods do not include a meaningful representation of the state-of-the-art (a single 2022 method in an active research field), but more importantly the only investigated dataset is very outdated (very small and much denser than practical data). Given its distributed nature, perhaps the technique could be of interest in a distributed setting, but there is no analysis of communication overhead or privacy, and no comparison to alternative methods. With the presented experimental results, this work seems more like an algorithm in search of a problem.

**Requested Changes:**

The primary requested change concerns Claim C4. The paper already makes a strong methodological contribution with thorough theoretical analysis. There is no real need for it to contain compelling experimental results; these can be considered more as illustrative (motivational) results showing how the technique might be applied in various settings.

The simplest solution is to modify the claim. The current claims read: “Through a variety of case studies, we showcase HMF’s benefits and applicability in video segmentation, time-series feature extraction, and recommender systems.” (Abstract) and
“We use a wide range of numerical experiments to demonstrate the effectiveness of the proposed HMF. The case studies on video segmentation, temporal signal analysis, and movie recommendation showcase the benefits of extracting shared and unique factors.” (Introduction).

The experiments in their current form do not demonstrate the "benefits and applicability" or the "effectiveness" of the proposed technique in these application domains. The experiments are very far from practical settings and do not compare to state-of-the-art methods (or even simple baseline techniques). The claims should be amended to include phrases like "analysis of synthetic datasets", "small-scale preliminary studies", "illustrate how the technique could potentially be applied in several application domains". The experiments do indicate how the method could potentially be applied, but I don't they do anything beyond that in terms of establishing genuine effectiveness or benefits.

The alternative, more involved option is to perform much more thorough experiments in at least one domain. For example, the authors could examine recent papers in the recommender systems conferences and conduct experiments similar to those reported there, adopting the expected datasets, baselines, and performance metrics. While the submitted paper is not a recommender systems paper, so there is not an expectation to have the same level of detailed experimental analysis, the goal is to have experiments that are sufficiently meaningful that a paper presenting them wouldn't be automatically rejected by a reviewer who conducts recommender systems research.

---

> ### Author Response · Authors · 2026-06-03
> **Thank you for the careful and constructive review!**
>
> We are glad that the methodology (Claim C1), the theoretical analysis (Claim C2), and the nonlinear extension (Claim C3) are acknowledged. We address the concerns about Claim C4 below.
>
> ---
>
> ## Experimental language
>
> > The claims should be amended to include phrases like "illustrate how the technique could potentially be applied."
>
> We agree that "benefits" overstate what the case studies are intended to show, and we will revise the abstract and introduction accordingly.
>
> This revision is intended to align the claims more precisely with the role of the experiments: they demonstrate HMF’s applicability across several settings while avoiding overstatement relative to domain-specific methods. As we explain below, the video and financial experiments provide head-to-head comparisons against linear factorization baselines.
>
> ---
>
> ## Video segmentation
>
> > Video segmentation is performed on a synthetic dataset with very artificial manipulation...
>
> We will not claim that we achieve state-of-the-art video segmentation performance. Our goal in this experiment is more focused: to evaluate HMF against related shared/individual low-rank decomposition methods under incomplete observations. The *linear* methods designed for this problem are JIVE, Robust JIVE (RJIVE), and perPCA.
>
> In the revision, we have further strengthened the comparison by adding linear models, including AJIVE, ProJIVE, MOFA+, BIDIFAC, Stacked RPCA, and SoftImpute+JIVE across three experimental scenarios (fully observed, 40% missing with zero-fill, 40% missing with spatial NN imputation). The results highlight the robust performance of HMF across all three scenarios.
>
> ---
>
> ## Financial data analysis
>
> Thank you for your comments. The S&P 500 return dataset is used in the original manuscript because the global/local structure is economically meaningful.
>
> The contribution relative to "calculating residuals from the market index" is threefold: (i) HMF recovers both the global and sector-specific factors simultaneously and jointly, rather than sequentially; (ii) it does so under a wide range of observation patterns (missing trading days, partial sensor coverage) with no imputation step; and (iii) it provides convergence and recovery guarantees. We will make these distinctions more explicit in the revision.
>
> In the revision, we have also added a quantitative baseline comparison on the stock return dataset, reporting held-out reconstruction MSE and Pearson correlation under 20% random masking. HMF achieves the best held-out Pearson correlation (0.829 vs. 0.738 for the next-best baseline, AJIVE), confirming that the global+local decomposition improves predictive accuracy on held-out returns.
>
> ---
>
> ## Recommender systems
>
> > The rating prediction setting falls far short of what would be remotely interesting.
>
> We agree that he MovieLens-100k dataset is a historical benchmark. We will reframe this experiment as an illustrative case study. State-of-the-art recommender systems are built on graph-based collaborative filtering, self-supervised learning, or generative models. HMF is not designed to compete with these methods, and we will say so explicitly in the revised manuscript.
>
> The MovieLens experiment is intended to illustrate that the HMF framework extends naturally to sparse and non-uniformly sampled data, and that the autoencoder variant can learn nonlinear embeddings in this setting. We will tone down the language about performance in the recommender context and describe the experiment as a proof-of-concept illustration.
>
> ---
>
> ## Distributed setting
>
> > Given its distributed nature, perhaps the technique could be of interest in a distributed setting.
>
> Thank you for the observation. The distributed structure of HMF is a direct consequence of the factorization: each client $i$ maintains its private local factors $(U_{l,i}, V_{l,i})$ and participates in updates to the shared global factor $U_g$ only. Per round, each client communicates a gradient of size $O(n_1 \times r_g)$ to the server, and receives an updated $U_g$ of the same size back. The local data matrix $M_i$ and local factors never leave the client.
>
> A few competing methods in the paper, including JIVE, RJIVE, AJIVE, and BIDIFAC, are not naturally distributed. JIVE and BIDIFAC both require forming the concatenated matrix $[M_1 | \cdots | M_N]$ on a single server for the joint SVD step, so the full data from all clients must be transmitted. AJIVE similarly requires a global sketch that involves all blocks.
>
> We will also clarify the privacy statement. HMF avoids sending raw local data and local factors to the server, but it does not by itself provide a formal privacy guarantee, such as differential privacy or cryptographic security. However, since HMF resembles standard distributed learning, popular privacy-preserving techniques, including differential privacy or secure aggregation, are directly applicable. In comparison, applying secure aggregation to many baseline algorithms could be more challenging.

---

### Review · Reviewer_TJPz · 2026-04-20

**Summary Of Contributions:**

The paper proposes the Heterogeneous Matrix Factorization (HMF) designed to characterize common and unique factors based on matrix factorizations imposing orthogonality constraints between shared and individual subspaces. The approach relies on N observed matrices with the same number of rows in which the common space provides a subspace shared across the N observed matrices whereas a different orthogonal subspace specific also in rank to each observed matrices are used to characterize the variability unique to each observed matrices. Whereas this has been studied previously, the authors’ approach presently provides a first-order method with convergence guarantees that is also able to naturally handle incomplete (missing) data. Specifically, the orthogonality between common and individual spaces are imposed using a reparameterization in which the individual spaces are optimized using invariant parameters ensuring the optimized space is orthogonal to the common space. Theoretical properties in terms of error bound on (Theorem 1) and convergence (Theorem 2) is derived. The approach is subsequently evaluated on synthetic datasets as well as three real datasets.

**Audience:**

Yes

**Audience Explanation:**

The approach of performing joint factorization with common and individual subspaces is a well studied problem of wider interest that is of good relevance to the TMLR community.

**Claims And Evidence:**

No

**Claims Explanation:**

**Strengths:**

* The approach is sound and the reparameterization useful.

* The theoretical properties derived is a strongpoint.

* The approach appears to perform well and can notably handle missing data

**Weaknesses:**

* The positioning of the paper with respect to the larger body of literature on multi-block and multi-view learning needs to be substantially improved.

* Furthermore, the method needs to be compared more extensively and systematically to the current stat-of-the-art and in particular the experimentation expanded to include more baselines.

* The missing data handling is not as I see it a unique feature of this approach as indicated in Table 1, and I consider this incorrect as the approach is not sufficiently grounded and compared to the related literature.

**The above weaknesses are detailed in the below:**

Several approaches also handles missing data, see also the approaches reviewed here:
Flores, Javier E., et al. "Missing data in multi-omics integration: Recent advances through artificial intelligence." Frontiers in artificial intelligence 6 (2023): 1098308.

This survey notably includes several imputation and masking approaches that could be compared against.

For imputation missing data handling is for instance included in:
**MOFA+** that uses sparsity to account for shared and individual view specific factors
Argelaguet R, Arnol D, Bredikhin D, Deloro Y, Velten B, Marioni JC, Stegle O. MOFA+: a statistical framework for comprehensive integration of multi-modal single-cell data. Genome Biol. 2020 May 11;21(1):111. doi: 10.1186/s13059-020-02015-1. PMID: 32393329; PMCID: PMC7212577.
As well as the more recent **BIDIFAC+** as described in (i.e., an extension of BIDIFAC that is referenced in the paper):
F Lock, J Park and KA Hoadley. Bidimensional linked matrix factorization for pan-omics pan-cancer analysis, Annals of Applied Statistics 2022

There are further additional common and individual subspace learning procedures that the paper needs to compare against. This includes:
**aJIVE:**
Feng, Qing, et al. "Angle-based joint and individual variation explained." Journal of multivariate analysis 166 (2018): 241-265.
That also proposes an efficient reparameterization framework for the common and individual subspace learning problem. Please relate and explain how your approach here differs.
The paper would further benefit from a comparative analysis to
**ProJIVE:**
Murden, R. J., Tian, G., Qiu, D., & Risk, B. B. (2026). Probabilistic Joint and Individual Variation Explained (ProJIVE) for Data Integration. Journal of Computational and Graphical Statistics, 1–17. https://doi.org/10.1080/10618600.2026.2639081
That uses EM based inference procedures and although not explicitly accommodating missing data could naturally be extended to also handle this.

My main concern is therefore the lack of comparison to the existing state-of-the-art within multi-block and multi-view learning for which this paper contributes to the literature. Specifically, there are more advanced procedures that needs to be discussed and the current approach related to.

I also find the current experimentation limited and I do not think it a fair comparison to compare the proposed method handling missing data to methods not handling missing data and treating missing entries as 0. This automatically produces inferior comparisons and the method proposed therefore need to be compared to these methods when everything is observed in the video experiments for a fair comparison and compared to the above approaches handling missing data properly for the missing data experimentation. Furthermore, I expect the existing baselines could trivially use imputation based on the model estimates for missing data handling – this should be discussed.
Finally, the paper does not investigate or discuss the critical issue of determining the rank. The manuscript should provide such discussion and assessment.

It is also unclear to me why the different baseline procedures are not also compared against (including the above related approaches) in the stocks market analysis. This comparison needs to also here be included. As a result, I find the experimentation to presently be insufficient.


**Minor comments:**

Please provide description of abbreviation when first time used, i.e. SE -> squared error (SE) etc.

the problem equation 3 is nonconvex. -> the problem in equation 3 is nonconvex.

avarage-> average

why is lemma D1 and all following text in bold in the appendix?

**Requested Changes:**

See the above, specifically:

Please ensure the presentation in Table 1 is correct in light of approaches also handling missing data.

Compare in the video experiment and synthetic data to the other approaches when everything is observed and there is no missing data. Consider further to include imputation based on modeling estimates as opposed to setting missing data to zero.
Include comparison to methods that handle missing data, see also
Flores, Javier E., et al. "Missing data in multi-omics integration: Recent advances through artificial intelligence." Frontiers in artificial intelligence 6 (2023): 1098308

Include all baselines in the analyses of the stock market dataset.

Clarify how you specify the rank and discuss how to set the rank of the individual and common subspaces, preferably also include analyses of the impact of rank choices.

---

> ### Author Response · Authors · 2026-06-03
> **Thanks for your comment!**
>
> Thank you for the encouraging and careful feedback. Below we describe the revisions and additional experiments we will add to address each point.
>
> ---
>
> ## Missing-data and multi-omics baselines
>
> Thank you for pointing this out. We will expand the related-work section to include the missing-data multi-omics literature and explicitly discuss Flores et al. (2023), MOFA+ (Argelaguet et al., 2020), and BIDIFAC+ (Lock et al., 2022). We will clarify that these methods and HMF share the goal of separating common and source-specific structure, but differ in several important ways:
>
> - **Orthogonality and identifiability.** HMF explicitly maintains orthogonality between shared and source-specific factors through the invariance-based correction step. This is important for convergence analysis and also helps numerical convergence.
> - **Scalability and distribution.** HMF is naturally distributed: each client updates local factors and sends only the shared factor update to the server.
> - **Theory.** HMF provides convergence guarantees for the proposed algorithm and a statistical error bound for the recovered shared and individual components under square loss.
>
> For the empirical revision, we will include **MOFA+** and **BIDIFAC** as additional quantitative baselines in both the synthetic and video experiments. Our BIDIFAC implementation uses rank-constrained alternating SVD with EM imputation for missing entries, which gives a fair comparison with all other methods. Results will be reported in the revised manuscript.
>
> ---
>
> ## Relationship to AJIVE and ProJIVE
>
> We agree and will add both methods to the discussion and experiments.
>
> **AJIVE.** AJIVE identifies joint and individual variation by first extracting signal spaces for each data block and then using principal angles to determine the joint subspace.
>
> **ProJIVE.** ProJIVE provides a probabilistic formulation for joint and individual variation and uses EM-style inference. Since ProJIVE does not directly optimize the same masked matrix-completion objective as HMF, we implement ProJIVE both under zero-fill and under the same nearest-neighbor imputation used for other complete-data baselines.
>
> We will add a paragraph in the related-work section emphasizing that AJIVE/ProJIVE and HMF are complementary.
>
> ---
>
> ## Missing entries imputation
>
> We agree with your comments on the imputation. In the revision, we will report three experimental regimes for the video segmentation experiment:
>
> | Scenario | Input to complete-data baselines | Input to HMF |
> | --- | --- | --- |
> | **A. Fully observed** | Original frames, no missing data | Original frames |
> | **B. 40% missing, zero-fill** | Zero-filled frames | Observation mask, frames |
> | **C. 40% missing, NN imputation** | Spatial nearest-neighbor-imputed frames | Observation mask, NN-imputed frames |
>
>
> ---
>
> ## Rank determination
>
> We agree that rank selection is an important practical issue and will add a dedicated paragraph and appendix subsection on rank determination. The revised manuscript will use the following protocol.
>
> For synthetic experiments, the true ranks are known by construction, so all methods use the true ranks in the original manuscrit as this makes the comparison among algorithms fair.
>
> For real-data experiments, we will select the shared rank and local rank using a held-out validation set of observed entries. Specifically, for each candidate pair $(r_g, r_l)$, we fit the model on the training entries and compute validation reconstruction error or PSNR on held-out observed entries. Then, we choose the ranks that maximize the validation metrics. For methods that require a total rank rather than separate shared/local ranks, we use the same total rank budget $r_g+r_l$ to keep the comparison fair.
>
> We now provide additional numerical results for both the synthetic dataset and the video segmentation task in the revised manuscript. Both results suggest that the cross-validation rank selection mechanism is reasonable and that HMF can withstand mild rank overparametrization.
>
> ---
>
> ## Stock-market analysis
>
> The stock market analysis in the original manuscript was used to demonstrate the interpretability of the factors learned by HMF. Based on your suggestions, we further explore the use of HMF on an extended dataset. Specifically, we download daily adjusted closing prices for 50 S&P 500 stocks spanning five sectors (Technology, Healthcare, Financials, Energy, Utilities), with 10 stocks per sector, covering January 2020 to December 2023.
>
> To perform numerical comparisons, we frame the problem as matrix completion by randomly sampling 20% of entries as the held out entries. We report Pearson correlation and MSE between the true and reconstructed held-out returns. The results are reported in the revised manuscript. HMF achieves the highest held-out Pearson correlation (0.829) and lowest MSE (1.165). AJIVE is the best baseline, followed by proJIVE and JIVE. The results confirm the superior data imputation performance provided by HMF.

---

> > ### Comment · Reviewer_TJPz · 2026-06-10
> > **I thank the authors for the added experimentation which has improved positioning the proposed framework.**
> >
> > I thank the authors for their added experimentation which has helped position the paper.
> >
> > I at this point have minor corrections to the manuscript that needs a bit of proof reading from the revision.
> > Abstract: HMF can be integrated auto-encoders -> HMF can be integrated into auto-encoders
> >
> > Table 1:  convergence guarantee -> convergence guarantee
> >
> > In Table 3 it is observed that “HMF and BIDIFAC achieve near-perfect global subspace recovery, while all other methods incur substantially larger errors.” In the missing data experimentation. It should here be discussed that these methods can explicitly handle missing data directly or by imputation which I assume is also why these two approaches here are performant.
> >
> > Please be consistent and define BIDIFAC in Table 3 as BIDIFAC (EM)
> >
> > Please also make reference in main text to Table 3.

---

### Decision · Action_Editor_nN82 · 2026-06-26

**Recommendation:** Accept as is

**Audience:**

Yes

**Audience Explanation:**

Matrix factorization and latent variable modeling is a core part of machine learning research, so the proposed research is of interest to the TMLR audience.

**Claims And Evidence:**

Yes

**Claims Explanation:**

The authors have introduced a matrix factorization method which assumes the shared and unique factors are orthogonal. The expanded experimental section has giving sufficient empirical evidence to the utility of the method, with supporting theoretical results.